



# Dynamical and chemical processes contributing to ozone loss in exceptional Arctic stratosphere winter-spring of 2020

Sergei P. Smyshlyaev[1], Pavel N. Vargin[2], Alexander N. Lukyanov[2], Natalia D. Tsvetkova[2], Maxim A. Motsakov[1]

[1]Russian State Hydrometeorological University, St. Petersburg, Russia

[2]Central Aerological Observatory, Dolgoprudny, Moscow region, Russia

*Correspondence to*: Sergei P. Smyshlyaev (smyshl@rshu.ru)

**Abstract.** The features of dynamical processes and changes in the ozone layer in the Arctic stratosphere during the winter-spring season 2019-2020 are analyzed using ozonesondes, reanalysis data and numerical experiments with a chemistry-

transport model (CTM). Using the trajectory model of the Central Aerological Observatory (TRACAO) and the ERA5 reanalysis ozone mixing ratio data, a comparative analysis of the evolution of stratospheric ozone averaged along the trajectories in the winter-spring seasons of 2010-2011, 2015-2016, and 2019-2020 was carried out, which demonstrated that the largest ozone loss at altitudes of 18-20 km within stratospheric polar vortex in the Arctic in winter-spring 2019-2020 exceeded the corresponding values of the other two winter-spring seasons 2010-2011 and 2015-2016 with the largest

decrease in ozone content in recent year. The total decrease in the column ozone inside the stratospheric polar vortex, calculated using the vertical ozone profiles obtained based on the ozonesondes data, in the 2019-2020 winter-spring season was more than 150 Dobson Units, which repeated the record depletion for the 2010-2011 winter-spring season. At the same time, the maximum ozone loss in winter 2019-2020 was observed at lower levels than in 2010-2011, which is consistent with the results of trajectory analysis and the results of other authors. The results of numerical calculations with the CTM with

dynamical parameters specified from the MERRA-2 reanalysis data, carried out according to several scenarios of accounting for the chemical destruction of ozone, indicated that both dynamical and chemical processes make contributions to ozone loss inside the polar vortex. In this case, dynamical processes predominate in the western hemisphere, while in the eastern hemisphere chemical processes make an almost equal contribution with dynamical factors, and the chemical depletion of ozone is determined not only by heterogeneous processes on the surface of the polar stratospheric clouds, but by the gas-

phase destruction in nitrogen catalytic cycles as well.

## 1 Introduction

The circulation of the Arctic stratosphere in the winter-spring season (hereinafter winter season) is characterized by strong interannual and seasonal variability, which can affect the tropospheric circulation and weather conditions (Kidston et al., 2015; Nath et al., 2016; Peters et al., 2018; King et al., 2019; Matthias et al., 2020), temperature and chemical composition



of the stratosphere and upper atmosphere (Funke et al., 2016; Pedatella et al., 2018) and ozone layer (Smyshlyaev et al., 2016; WMO, 2018). Due to the stronger wave activity and the frequent occurrence of Sudden Stratospheric Warming (SSW), and, consequently, the less stable and warm stratospheric polar vortex, significant ozone anomalies are observed in the Arctic less often than in the Antarctic, where the main SSW was observed only in September 2002 (Solomon, 2014). However, Antarctica has also experienced strong interannual variability in recent years. Thus, in 2015, the ozone anomaly in

Antarctica was one of the most significant for all observation years (Vargin et al., 2020a), while in 2019 it was one of the weakest due to the SSW that occurred in September (Millevsky et al., 2020; Safieddine et al., 2020).

Over the past 40 years, the largest decrease in the Arctic ozone was observed in the winter seasons of 1995-1996, 1996-1997, 2004-2005, 2010-2011, 2015-2016, as well as in 2019-2020. Prior to the 2019-2020 winter season, the maximum

ozone depletion in the Arctic was observed in 2011 (Manney et al., 2011; WMO 2014). As a result, increased levels of UV radiation on the surface were recorded in many regions of the Northern Hemisphere (UNEP, 2014), including Moscow in April (Belikov, 2012). Strong spring ozone depletion in the Arctic (as in 2011) may lead to an increase in UV radiation on the surface by 20-40% during the next summer season (Karpechko et al., 2013). Since 1979, in the spring and summer seasons, a positive trend of erythemic UV radiation has been observed over Eastern Europe and some regions of Siberia and

North-East Asia, amounting to 6-8% over 10 years, the main reasons for which are the reduction of the ozone layer and cloudiness (Chubarova et al., 2020). According to model estimations, in addition to the effect on UV radiation on the surface, a significant ozone depletion in the Arctic stratosphere can also affect the troposphere, for example, the surface temperature in April-May (Calvo et al., 2015), the spring precipitation in China (Ma et al., 2018) and Northwest United States (Xie et al., 2018). On the other hand, it was suggested that ozone anomalies do not have a statically significant effect

on weather conditions on the scale observed over the past 30 years (Smith and Polvani, 2014).

A possibly even greater ozone reduction in the Arctic stratosphere during the 2015-2016 winter season compared to 2010-2011 was prevented by a SSW in March 2016 (Khosrawi et al., 2017). In the meanwhile, in January-February 2016, lower values of the total column ozone (up to ~ 200 Dobson Units (DU)) were observed in the northern part of Russia, which is

50% less than the average climatic values (Nikiforova et al., 2019). Analysis of the model simulation revealed that these anomalies were mainly due to dynamical reasons (Timofeyev et al, 2018).

The 2019-2020 winter season in the Arctic stratosphere was characterized by an extremely strong and stable stratospheric polar vortex, which was caused by reduced wave activity propagation from the troposphere into stratosphere and its

downward reflection from the upper stratosphere, which strengthened the polar vortex through residual circulation changes (Lawrence et al., 2020). In late February - early March 2020, record low temperatures were observed in the Arctic lower stratosphere, and, as a result, a record volume of Polar Stratospheric Clouds (PSCs) was expected. As a result of the prevailing meteorological conditions favourable for ozone depletion, during five weeks in March and April 2020, the values



of the total column ozone were less than 220 DU, which makes it possible to speak about the first such long-term ozone
anomaly in the Arctic (Dameris et al., 2020). For the first time in all the years of observations, the analysis of data from a
number of certain ozonesondes revealed an extremely strong decrease in the ozone content in the Arctic stratosphere in the
spring of 2020, amounting to up to 90% (Wohltmann et al., 2020). MLS satellite data also indicate record low ozone
concentrations in the polar stratosphere during Arctic 2020 spring, which began to be recorded earlier than in all other years
and ended later than in all other years, with the exception of the winter season 2010-2011 (Manney et al., 2020). According
to data produced by the Copernicus Atmosphere Monitoring service (CAMS reanalysis) the monthly mean ozone columns in
Arctic in March 2020 were up to 180 DU or 40% lower than mean values over 2003–2019 (CAMS climatology) while
values for 2011 and 1997 were lower by 31% and 35%, respectively (Innes et al., 2020). Severe ozone depletion led to large
increase of solar ultraviolet radiation according to measurements performed at 10 Arctic and subarctic locations between
early March and mid-April 2020 (Bernhard et al., 2020). According to OMPS LP satellite observations, the area of the PSCs
during the winter 2019-2020 in the Arctic has reached the values typical to Antarctica (deLand et al., 2020). The general
conclusion of all studies: the reason for such record strong ozone depletion in the Arctic during spring 2020 is an unusually
cold and isolated stratospheric polar vortex.

Model experiments indicated, that one of the reasons for the low wave activity propagation to stratosphere could be the
positive phase of the dipole in the Indian Ocean (IOD), defined as the SST gradient between the western and eastern
equatorial Pacific Ocean. The positive phase of IOD through the propagation of wave chains to the northeast led to a
weakening of the Aleutian low (Hardiman et al., 2020), which, by analogy with the El Niño-South Oscillation effect,
weakens the propagation of planetary waves from the troposphere to the stratosphere. The frequency of positive IOD events
doubled in the 20th century, and their intensity also increased, and according to model projections, this trend is expected to
continue (Abram et al., 2020).

The 2019-20 winter was very different from the previous one, with the main SSW and splitting of the stratospheric polar
vortex in early January 2019, after which it recovered in the middle stratosphere, in contrast to the lower stratosphere, where
above zero temperature anomalies persisted until the final warming in April (Lee et al., 2020; Rao et al., 2020). The recovery
of the polar vortex after SSW in the middle and upper stratosphere could be due to the weak propagation of wave activity
from the upper troposphere - lower stratosphere from early January to mid-March and its reflection downward in the first
half of January 2019 (Vargin et al., 2020). According to model estimations that take into account the decrease in the content
of ozone-depleting substances in the atmosphere (revealed from satellite data since the early 2000s and is a consequence of
the implementation of the Montreal Protocol and its amendments) and the continued growth of greenhouse gases, leading to
decrease in the stratospheric temperature, by the middle of this century, significant ozone layer anomalies comparable to
record depletion in spring 2011 (Langematz et al., 2014). It has been suggested that due to changes in stratospheric climate
the cold northern winters are getting colder in coming decades and it will influence the winter loss of Arctic ozone (Rex et



al., 2004; Wohltmann et al., 2020). The cooling of the stratosphere could delay the recovery of the ozone layer (Pommereau et al., 2018). However, the other results of model simulations supposed that climate change will accelerate stratospheric ozone recovery instead of delaying it (e.g. Langematz et al., 2014).

Assessing the role of climate change in the occurrence of winter seasons with significant depletion of the ozone layer in the stratosphere, both in the Arctic and in the Antarctic, remains an urgent task, which determines the timing of ozone layer recovery (Pommereau et al., 2018; WMO, 2018). By the end of the twentieth century, the increasing anthropogenic impact on the ozone layer led to the formation of a tendency towards a decrease in the thickness of the ozone layer on a global scale and the regular appearance of spring ozone holes in Antarctica and the episodic appearance of a large ozone holes in the Arctic. The unprecedented measures taken by the joint efforts in the framework of Montreal protocol to reduce emissions of ozone-depleting substances containing chlorine and bromine components have led to a decrease in the tendency for an increase in the content of ozone-depleting substances in the stratosphere, and in recent years there have been signs of recovery of the ozone content (WMO 2014). However for most datasets and regions the total ozone trends since the stratospheric halogen reached its maximum (in 1996-2000) are mostly not significantly different from zero (Weber et al., 2018). Moreover multiple satellite measurements showed a decline of lower stratospheric ozone between 60°S and 60°N continuously since 1985 (Ball et al., 2018).

Signs of recovery in ozone levels began to be noted in the polar regions, in particular, a decrease in the depth of the ozone hole and its size in Antarctica. The 2019 ozone hole in Antarctica was one of the lowest in decades. Nevertheless, despite the decrease in the content of halogen-containing ozone-depleting substances in the atmosphere, episodes of low ozone content have been formed in the Arctic in recent years. In addition, the strong interannual variability of the ozone content that has always existed in the Arctic has been increasingly manifested in the Antarctic in recent years. In particular, after the 2019 low Antarctic hole, the 2020 hole is again one of the deepest in recent years (Butler et al., 2020; Wargan et al., 2020). Taking into account the fact that in the Arctic in 2020 one of the deepest spring ozone anomalies was recorded, understanding the processes affecting the variability of the stratospheric ozone in the polar regions under the conditions of the decrease in the concentration of ozone-depleting substances in the atmosphere controlled by the Montreal Protocol and its amendments requires further clarification.

The aim of this work is to study the dynamical, chemical processes and ozone layer changes and their possible causes in the Arctic stratosphere during the winter 2019-2020 using numerical modeling, data analysis and to identify the relative role of dynamic and chemical factors in the observed near record reduction in ozone content. The paper is organized in the following manner. The data used and the methodology of applied diagnoses are described in Section 2. Diagnostic results of the ozone layer and dynamical evolution for the Northern Hemisphere stratosphere winter of 2019–2020 are provided and an estimate of the polar chemical ozone loss using trajectory modeling and balloon measurements of ozone are given in Section



3. Results of chemistry-transport modeling experiments performed to reveal the relative role of dynamical and chemical processes in the formation of the Arctic ozone anomaly in the spring 2020 are presented in Section 4. The discussion and conclusion are given in Section 5.

**2. Data and methods of analysis**

The present study of the dynamical and chemical processes features and variability of the ozone layer in the Arctic stratosphere is carried out using numerical modeling (chemical transport model of the RSHU, trajectory model TRACAO), reanalysis data ERA5 (Hersbach and Dee, 2016), NCEP (Kalnay et al., 1996), JRA (Kobayashi et al., 2015), MERRA2 (Gelaro et al., 2017) and observational data. A brief description of applied methods of analysis is present below.

**2.1 Arctic stratospheric dynamics analysis**

The evolution of the temperature of the Arctic lower stratosphere, where the largest reduction of the ozone layer takes place, in the winter 2019-2020 was analyzed by comparing the minimum temperatures at 70 hPa in the region of 70-90° N with other winters with the severe depletion of the ozone layer: 1995-1996, 1996-1997, 2004-2005, 2010-2011, 2015-2016. Temperature anomalies of stratosphere-troposphere are calculated relative to climate means over 1981-2010. The

propagation of wave activity was analyzed by using the zonal mean meridional heat flux and three-dimensional Plumb flux (Plumb 1985). It is known as an extended Eliassen-Palm (EP) flux due to the fact that its zonal average is equal to the well-known EP flux. The Plumb flux vectors are proportional to the group velocity of a planetary wave packet indicating the direction of propagation of the wave activity and are useful to localize regions of wave activity sources and sinks.

The influence of the circulation of the Arctic stratosphere on the troposphere is analyzed through the propagation from the stratosphere to the troposphere of geopotential height anomalies from climatic values in the region of 60-90°N, normalized to the standard deviation. When multiplied by -1, this parameter corresponds to the North Annular Mode (NAM) index. Following to (Runde et al., 2016) the periods with an uninterrupted downward propagation of NAM index with values above +/- 1.5 σ from the middle stratosphere to troposphere were defined as events with a consistent downward propagation of

anomalies to the troposphere.

**2.2 Trajectory analysis**

To estimate the average behavior of ozone inside the polar vortex at the levels (18-22 km) in the Arctic winters 2010-2011, 2015-2016, and 2019-2020 the Lagrangian approach was applied similar to an analysis of Arctic stratospheric polar vortex in the winter of 2018-2019 (Vargin et al., 2020). An ensemble of forward 120-day trajectories were calculated inside the polar

vortex using TRACAO trajectory model (Lukyanov et. al., 2003) and ERA5 reanalysis data. ERA5 ozone mixing ratio data



were interpolated into the points of each trajectory and were ensemble averaged, providing the estimates of the mean ozone loss in descending air masses inside the polar vortex.

### 2.3 Ozonesonde analysis

Ozonesonde data obtained from several Arctic stations during the winter 2019-2020 has been used to estimate vortex-averaged chemical ozone loss inside the polar vortex. It is assumed that inside the well isolated polar vortex the temporal evolution of vortex-averaged ozone volume mixing ratio on an isentropic surface is driven by two processes: 1) the net chemical ozone change due to gas phase and heterogeneous reactions and 2) the dynamical ozone change due to diabatic descent of air mass inside the polar vortex (Braaten, et.al., 1994). Thus, chemical ozone loss rate on any isentropic surface can be derived as the difference between the observed resulting ozone change rate on this surface and the calculated dynamical ozone change due to the diabatic descent. The mean rate of temporal ozone change on different fixed isentropic levels from 350 K to 675 K has been calculated from the ozonesonde volume mixing ratio time series by linear regression method. The diabatic cooling/heating rates were calculated by the radiation transfer model (Chou et al., 1999) using daily temperature profiles from JRA reanalysis. A decrease of total column ozone due to chemical ozone loss can be deduced by integrating of cumulative ozone loss profile (Fig. 7) between 350 K and 675 K potential temperature levels. Detailed description of the calculation method is given in (Braathen, et. al., 1994, Tsvetkova, et al., 2004).

### 2.3 Chemistry-transport modeling

To assess the relative role of chemical and dynamical processes in the ozone anomalies formation, numerical experiments with a chemical transport model were carried out. The temporal evolution of the Arctic stratospheric gases was simulated with the Russian State Hydrometeorological University chemistry-transport model (RSHU CTM) using meteorological fields directly from the reanalysis data (Smyshlyaev et al., 2017). This version of the global RSHU CTM is based on the Institute of Numerical Mathematics and RSHU chemistry-climate model (INM RAS – RSHU CCM) (Galin et al., 2007; Smyshlyaev et al., 2020), but meteorological fields are not calculated but specified from Modern-Era Retrospective Analysis for Research and Applications version 2 (MERRA-2) reanalysis (Gelaro et al., 2017). The RSHU CTM has a 5°×4° horizontal resolution in longitude by latitude and 31 vertical sigma levels from the surface up to approximately 60 km. The distributions of the 74 oxygen, hydrogen, nitrogen, chlorine, bromine, carbon and sulfate gases are calculated in the manner described by Smyshlyaev et al. (1998). Polar Stratospheric Clouds (PSCs) formation and evolution is treated as a super cooled ternary solution ($H_2O/HNO_3/H_2SO_4$) based on the Carslaw (1995) parameterization. PSCs surface variability, depending on temperature, pressure and partial pressures of the relevant gases, including denitrification and dehydration through sedimentation is taken into account according to Sovde et al., (2007) and Smyshlyaev et al. (2010).

For a more detailed study of the influence of dynamical and chemical factors on the local variability of the ozone content, additional numerical experiments with the RSHU CTM for two additional scenarios to the baseline (PSC) scenario have




been done. The first scenario (noPSCaer) does not take into account the formation of PSCs during the Arctic winter and spring (from November 1 to April 30 north to 64°N), while the second scenario (noCHEMall) does not take into account any

chemical ozone destruction in this area during the same period. Comparison of the baseline scenario with these two additional scenarios makes it possible to estimate the periods when the chemical destruction of ozone is most effective after heterogeneous activation on the PSC surface, and when the gas-phase destruction of ozone in nitrogen catalytic cycles is more significant. In addition, based on a comparison of these calculation scenarios, it is possible to assess the comparative role of dynamical and chemical processes of ozone reduction.

**3. Results**

**3.1 Main features of Arctic stratosphere winter 2019-2020**

The evolution of the total column ozone (TCO) in the Arctic during March-April 2020 as measured by the Ozone Monitoring Instrument (OMI) instrument on the board of AURA satellite is presented in Fig. 1. Measurements indicate that the ozone anomaly at the early March (Fig. 1a) covered mainly the Eastern part of the Arctic with minimum values less than

220 DU in the Northern part of the European territory of Russia. In the Western Arctic, values below 300 DU were observed North to Alaska and Western Canada. A significant part of the Arctic was still in the polar night region at the beginning of March and was not covered by the measurements of the OMI instrument, which uses solar radiation. On the other hand, in the absence of the Sun, the chemical destruction of ozone does not yet reach high values associated with the previous halogen activation on the surface of polar stratospheric clouds, therefore, it can be assumed, that there should not be

extremely low values of TCO in the region of absence of observations by the OMI instrument.

In mid-March (Fig. 1b), the ozone anomaly turned around the pole to the East, covering most of the Arctic region of the Western hemisphere with minimum values less than 220 DU in the area of Northern Greenland and the Canadian Arctic Archipelago North to Canadian mainland. Again, the minimum values are detected along the border of the region of absence

of observations - the zone of polar night. In the Eastern Hemisphere, values below 250 DU are found north to the mainland of Eastern Siberia. By the beginning of April (Fig. 1c), the ozone anomaly turned further eastward, shifting towards Greenland, the Norwegian Sea, Svalbard and Franz Josef Land. The minimum values less than 200 DU were observed in the Northern part of Greenland. By mid-April, the ozone anomaly continued to move eastwards, covering most of the Arctic region of the Eastern Hemisphere with minimum values less than 220 DU in the Franz Josef Land archipelago. At the same

time, the territory with values below 300 DU also covered a significant part of the North-West Russia.





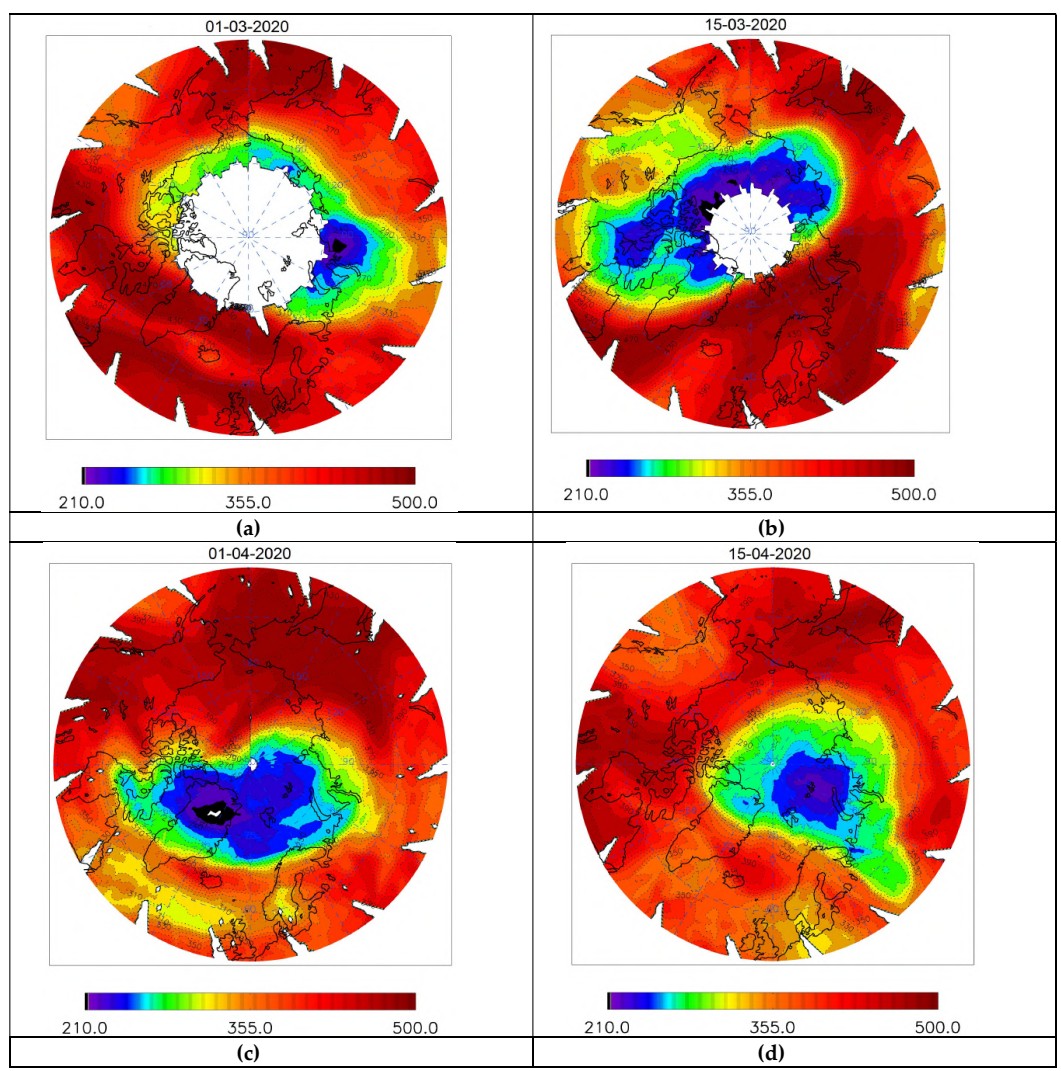

**Figure 1: AURA Arctic column ozone (Dobson Units) during 2020 spring: a- March 1, 2020; b- March 15, 2020; c – April 1, 2020; d - April 15, 2020**


The winter season 2019-2020 in the Arctic stratosphere was one of the coldest in the last 40 years. The minimum temperatures of the lower stratosphere since mid-February were either the lowest, or close to the record lowest observed in





the winter season 1996-1997 and 2010-2011, and sufficient (with temperatures below -78 C) for the formation of PSC type 1

(Fig . 2a). Negative temperature anomalies, relative to climatic values over the period of 1981 - 2010, were observed in the polar lower stratosphere during almost the entire winter season of 2019-2020 with maximum values in the range from -15 K to -10 K from mid-February to mid-April (Fig. 2b). Two main causes of so cold and stable Arctic polar vortex were suggested: reduced upward wave activity propagation from troposphere and downward refraction of wave activity from upper stratosphere (Lawrence, 2020). The evolution of wave activity propagation during the winter of 2019-2020 is

illustrated by variability of zonal mean meridional heat flux (Fig.3 a). Reduced wave activity was observed mainly over two periods: January 5- 20 and from early February to early March. The latter period was followed by enhanced wave activity propagation over about 10 days in the middle of March.

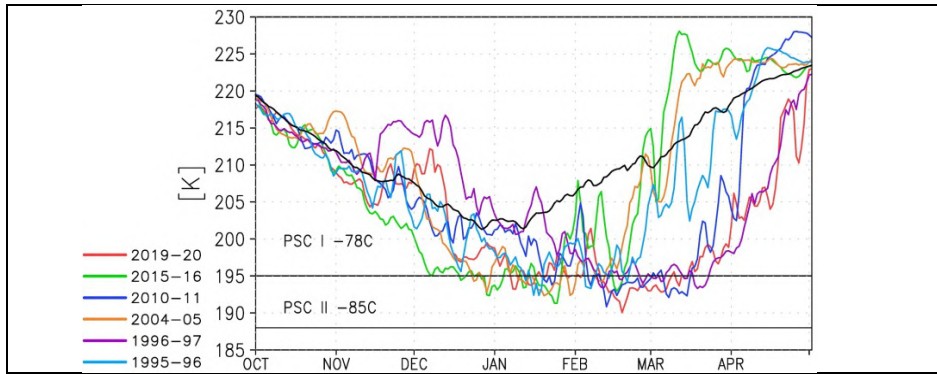

**Figure 2. Minimum temperature in the region 70-90° N at 70 hPa in October-April 1995-1996, 1996-97, 2004-05, 2010-**
**2011, 2015-16 и 2019-20. Black line corresponds to climate mean over 1981-2010.**

Further we compare the daily integrated zonal mean heat flux in the lower stratosphere at 70 hPa in the winter season of 2019-2020 over three time periods and normalized by the number of days in the winter of 2019-2020 with the five other winters with strong and cold stratospheric polar vortex and severe ozone destruction: 1995-1996, 1996-1997, 2004-2005,

2010-2011, 2015-2016 (Fig.3 b). The first period (February 7 - March 7) corresponds to strongest weakening of wave activity propagation in 2020, the second: January - February, the third: January - March. It is seen that heat fluxes over all periods are lowest in 2020 with the exception of the third period when it is slightly stronger than in 1997.



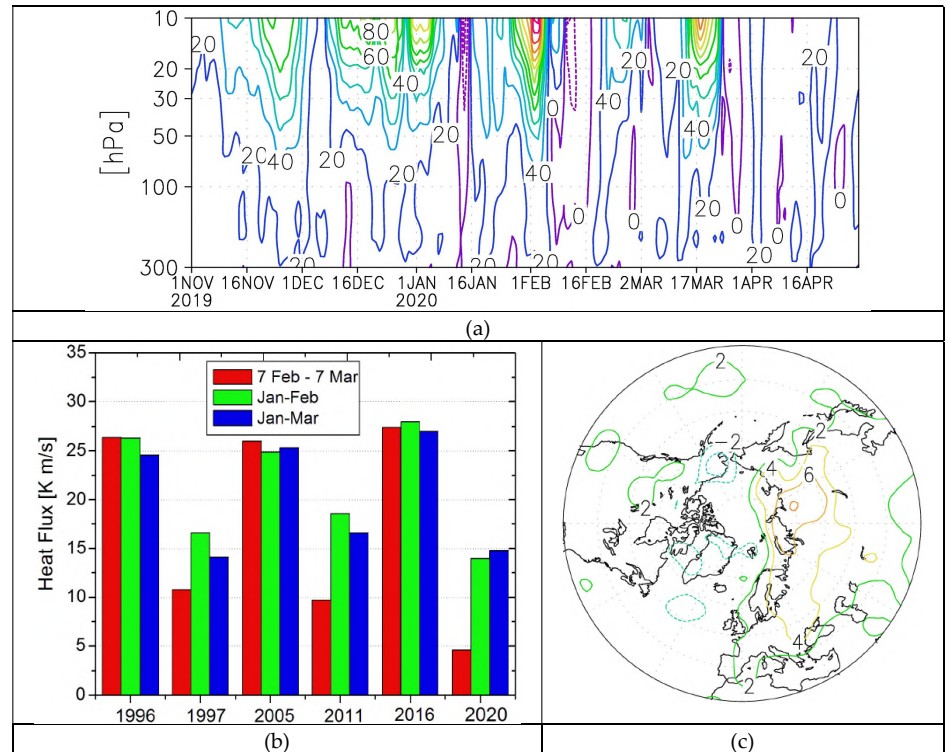

**Figure 3. Zonal mean meridional heat flux (K̄ m/s) averaged over 45-75° N from 1 November 2019 to 30 April 2020 (a).**
        **Zonal mean meridional heat flux at 70 hPa averaged over 45-75° N and the following periods: from 7 February to 7**
        **March, from 1 January to 28 February, from 1 January to 31 March of 1996, 1997, 2005, 2011, 2016, and 2020 (b).**
        **Temperature anomalies at 925 hPa averaged over the period from 7 February to 7 March 2020. Latitudes are from**
        **30°N (c).**


        At the same time, the winter season of 2019-2020 in most of Northern Eurasia was characterized by anomalously warm
        weather due to increased westerly winds. The positive temperature anomalies were observed in most of Northern Eurasia,
        including Siberia, where the absolute maximum of the average temperature in February was reached. The period with
        strongly reduced wave activity was also characterized by positive anomalies of temperature over the north-eastern Eurasia
(Fig 3c). Positive temperature anomalies over the Northern Eurasia of more than 5 K are retained when averaged over
        January-March 2020 (Lawrence et al., 2020). The positive anomalies up to 6 K were observed over the north-eastern Eurasia





during the period of strongly reduced upward wave activity propagation: February 7 - March 7 (Fig. 3c). Notably that described positive temperature anomalies were observed not only near surface but at higher levels in troposphere.

The dominant mode of circulation in the troposphere of extratropical latitudes of the Northern Hemisphere in winter is the Arctic Oscillation (AO). The AO positive phase is characterized by a reduced pressure at the pole and increased in the region of 40-50°N, and the negative one, on the contrary (Thompson and Wallace, 1998). With a positive AO phase, a stronger western zonal transport leads to milder winters, but with more precipitation in Southern Europe. In the negative AO phase, this transfer is weaker; as a result, cold air masses from the Arctic spread more strongly to the territory of Europe, including

the European territory of Russia.

AO is the result of interaction between the dynamics of the stratosphere and the troposphere. The positive AO phase is associated with a positive anomaly in the intensity of the stratospheric polar vortex, which is facilitated by an increase in the temperature gradient between the heated by the sun and shaded parts of the atmosphere, an increase in the zonal mean flux,

and smaller amplitude of planetary waves. The stratospheric polar vortex becomes less sensitive to the effects of waves, due to their refraction towards the equator. Also the strong stratospheric polar vortex is often accompanied by a positive AO phase (Thompson and Wallace, 1998).

In the first half of January and February-March 2020, a positive AO phase was observed with index values of more than 4 in

early January and then in February-March. Average monthly values of the AO index were 3.4 in February and 2.6 in March. Thus, the positive AO phase with a reduced propagation of wave activity into the stratosphere could contribute to the enhancement of the Arctic stratospheric polar vortex (Lawrence et al., 2020).

In the second half of February and during most of March 2020, areas with NAM index values above 1.5 σ spread

continuously from the middle stratosphere to the lower troposphere, which indicates the influence of the Arctic stratosphere on the troposphere (Fig. 4a). After the SSW event in the mid-March, this spread ended nearly in two weeks.

We assume that the anomalously warm winter 2019-2020 (and especially in February- early March with high positive AO index) contributed to the strengthening of the stratospheric polar vortex due to a decrease in the propagation of wave activity

into the stratosphere from the troposphere over the north-eastern Eurasia. It is known that the main source of wave activity propagation into the stratosphere, characterized by the maximum of the vertical component Fz of Plumb's fluxes, (e.g. Jadin 2011) is located over this region.  Comparison of two diagrams with Plumb vertical component Fz at 100 hPa averaged over February 7 - March 7, 2020 and corresponding climate mean over 1981-2010 shows that weakening of upward wave activity propagation in the first period was observed over the north-eastern Eurasia (Fig. 4 b, c). In the same time, a strong

stratospheric polar vortex in February-March 2020 influenced the troposphere, enhancing the positive AO phase.





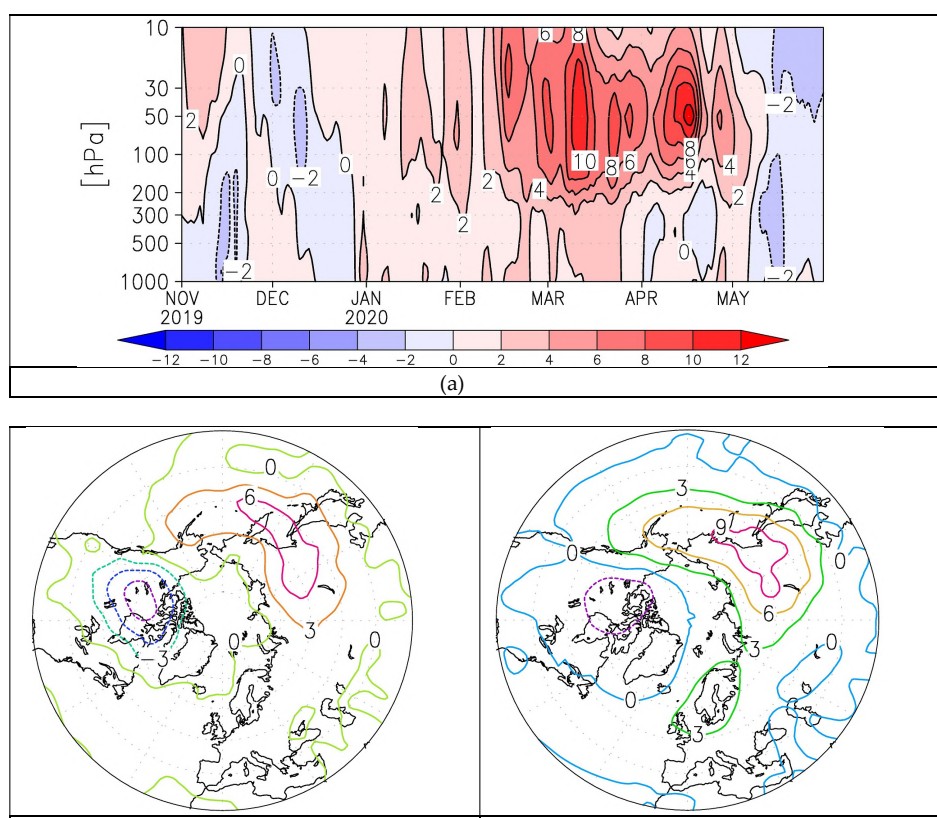

**Figure 4. NAM index in November 2019 - May 2020. Green contour corresponds to +/- 1.5 σ (a). Vertical component of**
**Plumb fluxes Fz (m²/s² · 10⁻²) at 100 hPa averaged over the period of February 7 - March 7 2020 (b) the same but for**
**climate mean from 1981 to 2010 (c).**

After the period with a strongly weakened propagation of wave activity from the troposphere to the stratosphere (since early

February till early March), a sharp increase in such propagation was observed in the upper troposphere - lower stratosphere

over northwestern Canada in the middle of March over about 10 days. This is confirmed by the diagram with the vertical

component of the Plumb fluxes in the lower stratosphere at 100 hPa during the SSW event onset on March 14-16 (Fig. 5a).

The enhanced propagation of wave activity, which led to the development of SSW event, is apparently associated with the

Rossby wave breaking event in the troposphere over the Gulf of Alaska region (150°W - 120°W), and accompanied by a poleward transport of wet and warm air masses with low potential vorticity and formation of anticyclone (Fig. 5 b). Such a

poleward transport, leading through a change in the zonal current and redirection of wave activity upward to the stratosphere (instead to the equator), can lead to the development of SSW event, for example, as in January 2006 (Coy et al., 2009).

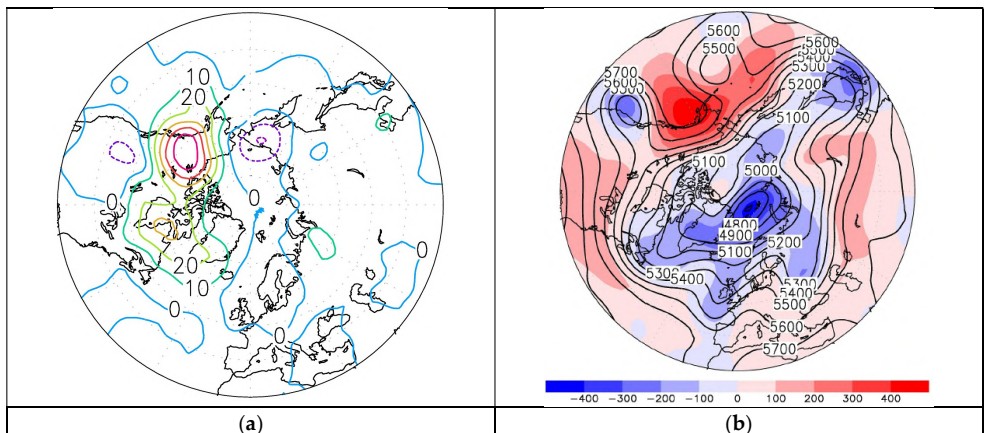

**Figure 5. Vertical component of Plumb fluxes (m²/s² · 10⁻²) at 100 hPa for March 14-16 (a), geopotential height (contours) and its anomalies from climate mean (gpm) (color scale) at 500 hPa on March 16, 2020 (b).**

Propagation of wave activity from region of Gulf of Alaska was analyzed by Plumb fluxes. On March 11-13 altitude-latitudinal diagrams of Plumb fluxes display dominated equatorward propagation of wave activity in the upper troposphere - lower stratosphere (Fig. 6 c). Later on the enhanced upward propagation of wave activity in the troposphere and lower-middle stratosphere is clearly seen on March 14-16, 2020 (Fig. 6 d). Zonal mean zonal wind weakening in lower - middle stratosphere over high latitudes is also observed. Altitude-longitude diagram of Plumb fluxes over the northern high latitudes

(50-70°N) illustrates enhanced upward and eastward propagation from the region of an anticyclone formed in the troposphere nearby 150°W (Fig. 6 e, f). On similar diagram 3 days earlier (March 11-13), such enhanced wave activity propagation and the anticyclone over Gulf of Alaska, were not observed.







**Figure 6. Altitude-latitude diagram of zonal mean Plumb fluxes (Fy, Fz components) and zonal mean zonal wind on March 11-13 (a) and March 14-16 2020 (b). Altitude-longitude diagram of Plumb fluxes (Fx, Fz components) and geopotential height deviation from zonal mean (gpm) over 50-70° N and the same periods (c, d). The vertical component of wave activity flux is multiplied by a factor of 100.**

**3.2 Ozone loss estimation inside the polar vortex**

For Lagrangian estimations of ozone losses thousands of 120-day forward trajectories were initiated in December inside the stratospheric polar vortex at the isentropic levels 550 K and 475 K. For simplicity the trajectories were initiated uniformly distributed on the $85^0$N latitude circle, when it was completely located inside the polar vortex. For the winter 2010-2011 it was on December 9, 2015-2016 – December 16, 2019-2020 – December 17. Trajectories slowly descent due to diabatic

cooling and mostly remained inside the polar vortex in the winters 2010-2011and 2019-2020 up to the end of March. In the winter 2015-2016 trajectories dispersed to middle and low latitudes after the major final SSW event in early March (Manney, Lawrence, 2016). ERA5 ozone mixing ratio data were interpolated to the points along the trajectories and were averaged. The ozone loss was estimated as the difference between the average ozone value at the 1-st date and at the current date. Results of comparisons between winters at the level 475 K are shown on Fig. 7 (2010-2011, 2015-2016, 2019-2020 -

green, blue and red curves respectively). It is seen that ozone losses in the winter 2019-2020 were higher than in the other winters. The oscillations of the blue curve are caused by the large dispersion of the vortex air masses to mid-latitudes after the SSW event in the winter 2015-2016. Comparisons for the different initial vertical levels 550 K and 475 K (not shown) demonstrate that ozone losses in the winter 2019-2020 are much higher in lower altitudes, that is consistent with results of Manney et. al., (2020).






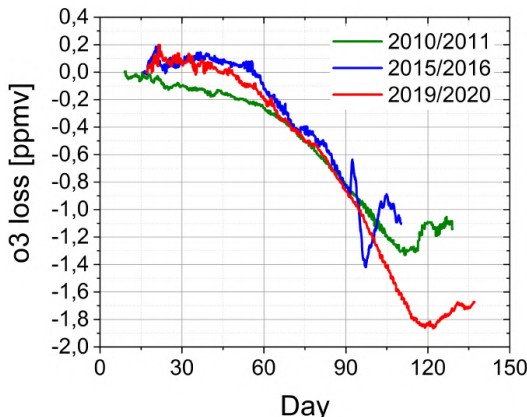

**Figure 7. Ozone loss along trajectories initiated inside the stratospheric polar vortex at 475 K in the winters of 2010-2011 (blue), 2015-2016 (green), 2019-2020 (red). 0 day corresponds to December 1**


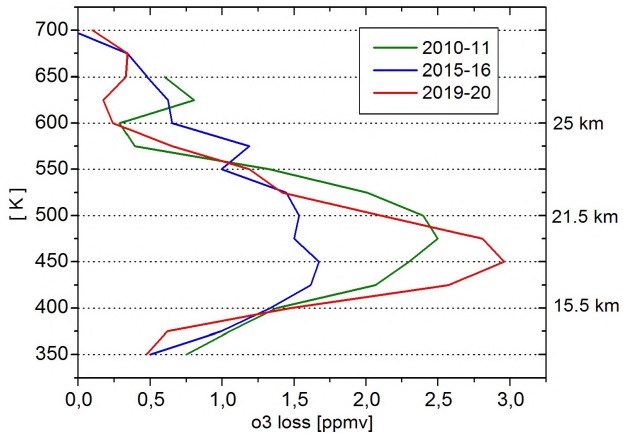

**Figure 8. Vertical distribution of the chemical ozone loss in the stratospheric polar vortex for the winters of 2010-11, 2015-16, and 2019-20.**

It should be noted that our results were obtained using ERA5 data with the horizontal resolution $0.25^0$ x $0.25^0$ and temporal resolution 1 hour. Applying reanalysis data with lower resolution ($0.75^0$ x $0.75^0$ and 6 hours) corresponding to ERA-Interim





data leads to an overestimation of the average vertical descending and as a result to the different ozone loss estimates in all the analyzed winters.

As well to estimate chemical ozone loss inside the polar vortex we used ozonesonde data from four Arctic stations - Alert (82°N, 62°W; Canada), Eureka (79°N, 85°W; Canada), Ny-Ålesund (78°N, 11°E; Norway) and Sodankylä (67°N, 26°E; Finland). About 60 ozonesonde profiles measured at different parts of the vortex from early January to late March 2020 were included in the analysis, most of them were obtained near the vortex centre with high values of potential vorticity (PV). The vortex edge was defined as 42 PVU at 475 K isentropic surface. This criterion was chosen to make sure the measurements

are taken inside the polar vortex.

    Figure 8 demonstrates vertical distribution of vortex-averaged cumulative ozone loss from the beginning of January to the end March 2020 deduced from ozonesonde measurements in comparison to the previous cold winters of 2010-2011 and 2015-2016 with largest ozone losses. Significant ozone loss had been seen from the mid-January till the end of March

between 400-525 K isentropic levels (~15-22 km). In the winter of 2019-2020 ozone losses were the lowest on record but occurred at lower altitudes than in 2010-2011 that is consistent with the results of the described above trajectory analysis and the results of Manney et al., (2020). At the isentropic level of 450 K (~18 km) ozone loss reached a maximum value of ~ 3 ppmv by the end of March. The vortex-averaged decrease in total column ozone over the winter 2019-20 reached $157 \pm 22$ DU, that repeated the record of the winter 2010-2011 ($153 \pm 13$ DU) and exceeded the corresponding value for the winter of

2015-2016 ($135 \pm 40$ DU). This result is very close to that obtained by Goutail et al., (2020). Ozone loss in the partial column between 375 and 550 K isentropic levels shows value $137 \pm 12$ DU similar to Wohltmann et al., (2020).

    Earlier, using the same methodology for the 2002-2003 winter season with the main sudden warming, the ozone loss estimate was ~ 45 DU, and for the much colder 2004-2005 winter: ~ 116 DU (Peters, 2010).

### 4. Chemistry-transport modeling

For a more detailed study of the degree of dynamical and chemical processes influence on the formation of ozone anomalies during the spring of 2020, numerical experiments with a chemical-transport model were carried out, in which the dynamic parameters were set from the MEPRA-2 reanalysis data. The use of temperature, wind speeds, surface pressure and air humidity from the reanalysis data made it possible to simulate the effect of atmospheric circulation on the transport of ozone and associated gases close to reality. Variability of specified dynamical parameters determines the dynamical decrease in

ozone content, as well as the atmospheric temperature govern the rate of chemical reactions, polar stratospheric clouds formation and the rate of heterogeneous reactions on their surface, which determine the chemical destruction of ozone.




**Figure 9. CTM Arctic column ozone (Dobson Units) during 2020 spring: a- March 1, 2020; b- March 15, 2020; c – April 1, 2020; d - April 15, 2020.**

First, a basic numerical experiment was performed taking into account all factors affecting Arctic ozone. The variability of the atmospheric gas composition during the winter of 2019-2020 was calculated in the basic numerical experiment. Figure 9





demonstrates the results of calculations of the total column ozone for March-April 2020, performed using the CTM with the specified dynamical parameters from the MERRA-2 reanalysis. Comparison of the results of numerical modeling to the results of measurements by the OMI instrument (Fig.1) indicates that the model reproduces all the main phases of the

evolution of the ozone anomaly during spring 2020. In particular, its movement in the eastward direction and the minimum values at the beginning of March in the Northern part of the European territory of Russia are reproduced. In mid-March - in the western part of the Arctic, in the area of Greenland, Svalbard and Franz Josef Land in early April and North to the mainland of the ETR in mid-April. In addition, the model results supplement the results of satellite measurements in the absence of measurements in the polar night region in March. The model results demonstrate that the minimum values are

observed at the boundary of the polar night in the part where the Sun has already returned. This confirms the hypothesis of the effect of the chemical destruction of ozone, which intensifies after heterogeneous activation in polar stratospheric clouds and the return of the Sun. However, the results of model calculations reveal that relatively low values of the total column ozone (below 300 DU) are also observed in the polar night zone, where the chemical destruction of ozone is slowed down. This also indicates a significant influence of dynamical factors on the formation of regions of low ozone content.


To better understand the relative role of dynamical and chemical processes in the formation of the Arctic ozone anomaly in spring 2020, Fig. 10 demonstrates the surface area of type 1 polar stratospheric clouds in the lower stratosphere (15-25 km), calculated with the CTM. Basically, the regions of PSCs formation correspond to the region of low temperatures, but the inertia in their melting with increasing temperature is also taken into account. The results of the calculations indicate that the

region of existence of the PSCs, as well as the territory of the ozone anomaly, undergoes a rotation in the eastward direction during March-April 2020 (Fig. 10). At the same time, the area of the PSCs zone is maximum at the beginning of March, and the maximum surface area of the PSCs itself is reached in mid-March. During April, both the PSCs surface area and the area covered by PSCs significantly reduced. In addition, it should be noted that the area where the maximum surface areas of PSC are observed do not completely coincide with the zones of minimum values of the column ozone, which suggests that the

relationship between the formation of PSCs and ozone destruction is not linear and confirms the theory of several stages of the formation of ozone anomalies. In the polar stratosphere, associated, first, with the formation of PSCs, then with halogen activation on their surface, and only then with ozone destruction after the return of the Sun after the polar night.





**Figure 10. Low stratosphere PSCs surface area (mkm$^2$/cm$^3$) estimated with CTM during 2020 Arctic spring: a- March 1, 2020; b- March 15, 2020; c – April 1, 2020; d - April 15, 2020.**




**Figure 11. Low stratospheric coefficient of ozone destruction ($10^8$/s) estimated with CTM during 2020 Arctic spring: a- March 1, 2020; b- March 15, 2020; c – April 1, 2020; d - April 15, 2020.**





Fig. 11 demonstrates the coefficient of chemical ozone destruction in the lower stratosphere calculated using CTM. This coefficient is a factor by which the concentration of ozone should be multiplied in order to obtain the rate of its chemical destruction. The coefficient includes all processes of chemical destruction of ozone, including processes involving chlorine, bromine, nitrogen and hydrogen gases. It should be noted that the rate of ozone destruction in March has its maximum values at the boundary of the polar night in the region of the newly returned Sun. In this case, the maximum rate of ozone

destruction is a necessary, but not sufficient condition for the formation of a zone of low ozone content in the spring. Dynamical factors also play an important role, in particular, for definition of the zone where the polar vortex is located. In particular, in early March, the rate of destruction of the base is maximum over the entire circle of latitude near the boundary of the polar night, and the minimum values of the ozone content are noted only in the eastern hemisphere (Figs. 8 and 9). Also in mid-March and April, areas of high ozone depletion cover a wider zone than areas of minimum total ozone.


For a more detailed study of the influence of dynamical and chemical factors on the local variability of the ozone content Figures 12 - 15 present the simulated with the CTM and measured by the OMI instrument changes in the total ozone content at four stations (two in the Western Hemisphere and two in the Eastern Hemisphere) during six months from the beginning to mid-2020. For comparison with the regular ozone variability at these stations, the same figures contain the average ozone

change for the period 1979-2019 according to SBUV data. In addition, the same figures present the CTM simulation results for two additional scenarios to the baseline (PSC) scenario. The first scenario (noPSCaer) does not take into account the formation of PSC based on stratospheric sulfate aerosol during the winter season (from November 1 to April 30 North to 66° N), and the second scenario (noCHEMall) does not take into account any chemical processes at this region during the same period. Comparison of the baseline scenario with these two additional scenarios allows us to estimate the periods when

chemical destruction of ozone is most effective after heterogeneous activation on the PSC surface, and when gas-phase destruction of ozone in nitrogen catalytic cycles is more significant. In addition, the comparative role of dynamical and chemical processes of ozone reduction can be assessed by comparing these scenarios with each other and to mean climatic values presented at the bottom of these figures.

At Pechora station (65.12°N, 57.1°E), located in the north of European territory of Russia (Fig. 11), one can note the formation of several local minima of the total ozone content with a characteristic time of several days in late January, mid and late February, which are reproduced qualitatively by model simulations for all three scenarios, which indicates the predominant role of dynamical processes in the variability of the ozone content in winter. With the beginning of spring, the influence of chemical factors increases, which maximally affect the ozone depletion in April. In March, when the

stratospheric polar vortex was located, for the most part, in the western hemisphere, at Pechora station, the ozone concentration values correspond to the seasonal maximum, reaching values up to 450 DU. From the end of March, when the polar vortex returns to the Eastern Hemisphere (Figs. 7 and 8), the values of the total ozone content at Pechora station fall by





almost two times, reaching values about 250 DU. Calculations according to the noCHEMall scenario show that, as a result of the influence of dynamic factors without taking into account any chemical destruction of ozone, its decrease from mid-
March to mid-April reaches 150 DU, and up to 80 DU if compare to climatic data. In this case, the chemical destruction of ozone, based on a comparison of the baseline and additional scenarios, amounts to 70 DU. Based on a comparison of the noPSCaer and noCHEMall scenarios, it can be concluded that in the chemical destruction of ozone at Pechora station, the heterogeneous part is about one third (~ 25 DU), and the gas-phase part is ~ 45 DU.

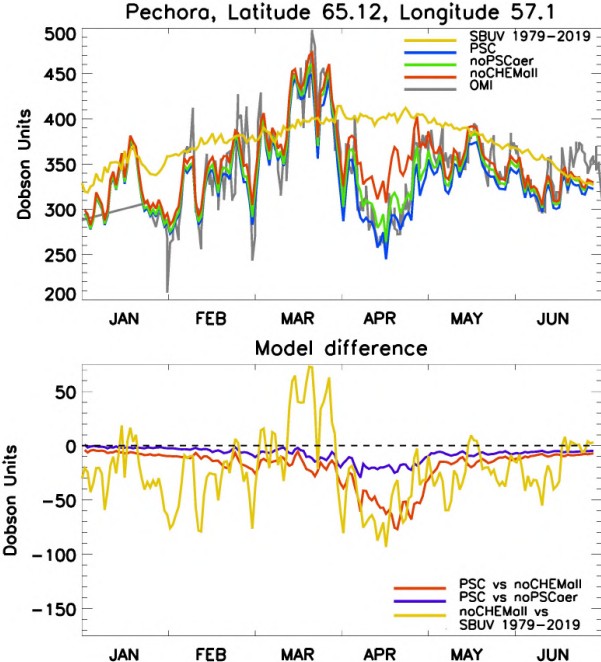

**Figure 12. Column Ozone variability at Pechora Station for the first half of 2020 (top): SBUV mean over 1979-2019 (yellow), OMI data (grey), CTM basic scenario (PSC) (blue), CTM scenario without PSC processes included (noPSCaer) (green), CTM scenario with no chemical ozone destruction included (noCHEMall) (red); and column ozone difference between CTM simulation for several scenarios (bottom): noCHEMall and SBUV 1979-2019 averages (yellow), PSC and noCHEMall (red), PSC and noPSCaer (blue)**


At another station, Tura (64.167°N, 100°E), located eastward to Pechora station, near the zone of minimum values of the total ozone content in early spring (Figs.7a and 8a), the minimum values are reached in the first half of March and the second half of April (Fig. 12). During this time, the stratospheric polar vortex rotates around the pole, due to which the fluctuations

in the total ozone content range from 250 DU in early March to 450 DU in the second half of March and again to 250 DU at

the end of April. Analysis of the noCHEMall scenario reveals that fluctuations in the total ozone content due to dynamical

factors amount to more than 150 DU, and analysis of the model simulation results for different scenarios of switching off the

chemical destruction of ozone demonstrates that the chemical destruction of ozone at Tura station is from 30 to 50 DU, of

which about half (about 25 DU) is the destruction of ozone as a result of halogen activation on the surface of polar

stratospheric clouds. At the same time, results of noCHEMall scenario comparison with the average climatic data shows that,

due to dynamical factors, the decrease in the ozone content at Tura station is on average greater than at the Pechora station

and exceeds 100 DU for some periods.

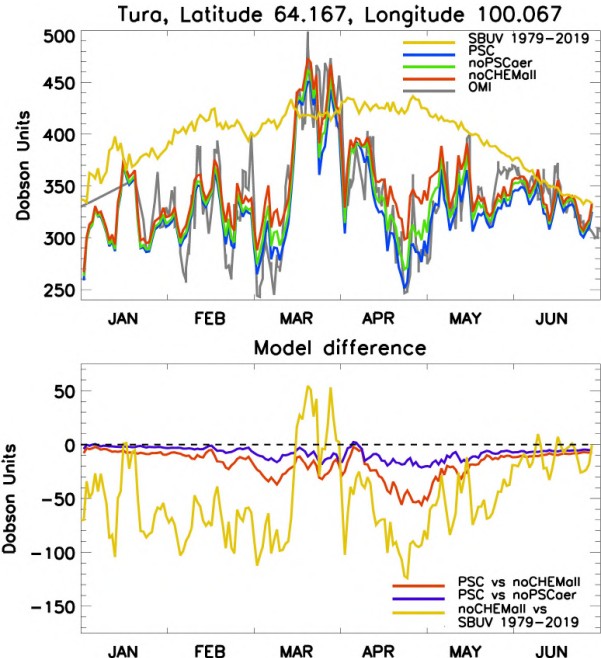

**Figure 13. Column Ozone variability at Tura Station for the first half of 2020 (top): SBUV mean over 1979-2019 (yellow), OMI data (grey), CTM basic scenario (PSC) (blue), CTM scenario without PSC processes included**
**(noPSCaer) (green), CTM scenario with no chemical ozone destruction included (noCHEMall) (red); and column ozone difference between CTM simulation for several scenarios (bottom): noCHEMall and SBUV 1979-2019 averages (yellow), PSC and noCHEMall (red), PSC and noPSCaer (blue)**

In the western part of the Arctic, at stations Resolute (74.72°N, 94.98°W) (Fig. 14) and Eureka (80.04°N, 86.17°W) (Fig.

15), the minimum values of the total ozone concentration (less than 250 DU) were observed in March: mid-March at

Resolute station (Fig. 14) and at the second part of March at Eureka station (Fig. 15). Comparison of the calculation results





for different scenarios indicates that the influence of dynamic factors in the Western Hemisphere is more significant than in the Eastern Hemisphere, since for the noCHEMall scenario comparison to climatic data, the total content fluctuates in the order of 100-150 DU. As can be seen from the comparison of the baseline scenario with additional ones, the chemical destruction of ozone also ranges from 70 to 80 DU, which is slightly higher than in the Eastern Hemisphere. At the same time, in April, chemical factors for ozone destruction prevail over dynamic factors at western stations. It should also be noted that there are two peaks of maximum chemical destruction of ozone: in late March and mid-April. At the same time, chemical destruction in the second half of March is superimposed on a dynamic decrease in its content, which leads to a minimum in the seasonal variation of the total ozone content, while in April, when the chemical destruction of ozone is even greater than in March, the polar vortex is already shifting towards the eastern hemisphere (Fig. 8 and 9), and the total ozone content is higher than in March.

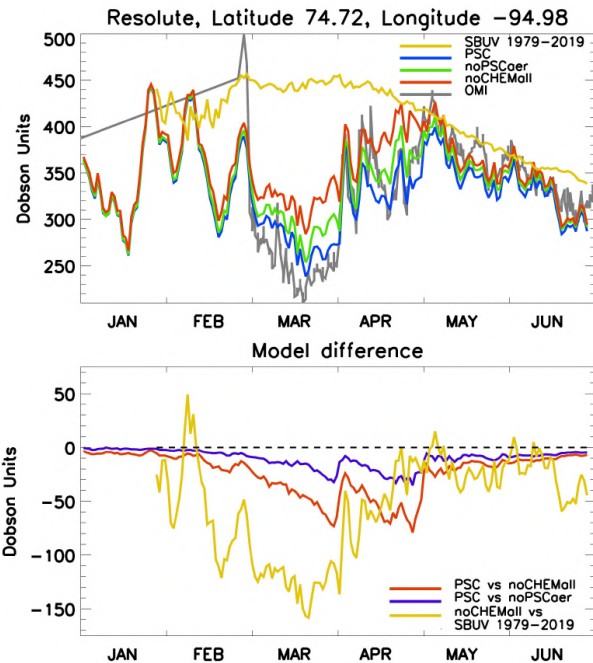

**Figure 14. Column Ozone variability at Resolute Station for the first half of 2020 (top): SBUV mean over 1979-2019 (yellow), OMI data (grey), CTM basic scenario (PSC) (blue), CTM scenario without PSC processes included (noPSCaer) (green), CTM scenario with no chemical ozone destruction included (noCHEMall) (red); and column ozone difference between CTM simulation for several scenarios (bottom): noCHEMall and SBUV 1979-2019 averages (yellow), PSC and noCHEMall (red), PSC and noPSCaer (blue)**



Comparison of calculations for different scenarios of accounting for the chemical destruction of ozone depicts that the destruction of ozone over heterogeneous reactions in the western hemisphere exceeds 30 DU, which is more than in the eastern hemisphere, while the gas-phase destruction of ozone in the Western hemisphere is greater than in the Eastern

Hemisphere. It should also be noted that in the Western Hemisphere, the minimum values of the ozone content according to satellite measurements in March are lower than the values calculated using the model, while in the Eastern Hemisphere the satellite and model results are closer. This result may be due to relatively coarse model resolution to simulate fine local effects in the western hemisphere.

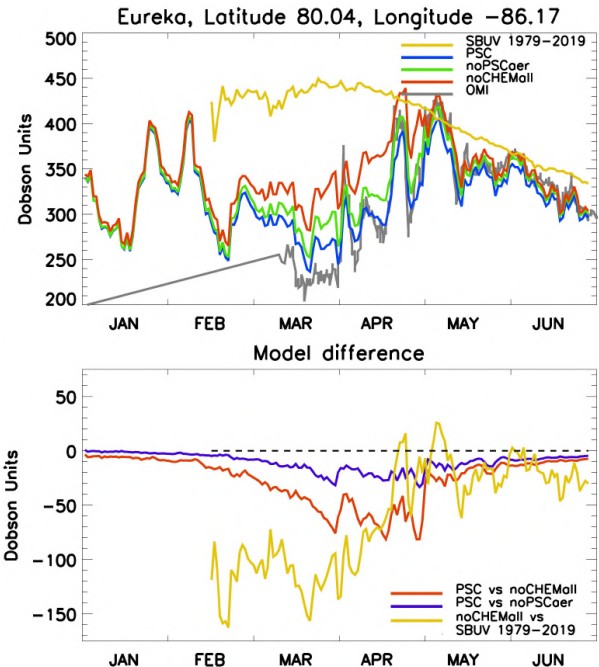

**Figure 15. Column Ozone variability at Eureka Station for the first half of 2020 (top): SBUV mean over 1979-2019 (yellow), OMI data (grey), CTM basic scenario (PSC) (blue), CTM scenario without PSC processes included (noPSCaer) (green), CTM scenario with no chemical ozone destruction included (noCHEMall) (red); and column ozone difference between CTM simulation for several scenarios (bottom): noCHEMall and SBUV 1979-2019 averages (yellow), PSC and noCHEMall (red), PSC and noPSCaer (blue)**


Additional numerical calculations to assess the effect of various catalytic cycles of chemical ozone destruction on a decrease in its content in April-May 2020 revealed that the main increase in the gas-phase ozone destruction occurs in the nitrogen



catalytic cycle, in which the chemical reaction with the participation of nitrogen dioxide and atomic oxygen plays a determining role. In the Arctic stratosphere, in contrast to the Antarctic stratosphere, significant denitrification does not

occur, and therefore a sufficient amount of nitrogen oxides remains in it, which plays a decisive role in the destruction of stratospheric ozone.

## 5. Discussion and conclusions

Motivated by sufficiently rare event in the Arctic: very strong and long-lasting stratospheric polar vortex and near record ozone loss in the winter season 2019-2020 we have analyzed the peculiarities of stratospheric dynamical processes including

those that contributed to the SSW event in the middle of March 2020 and interrupted the period of weakened wave activity propagation and strengthening of the stratospheric polar vortex and ozone layer destruction. Further we have estimated the values of chemical ozone loss by trajectory modeling and using ozonosondes observations and finally we have investigated dynamical and chemical processes in the Arctic stratosphere and their role in ozone destruction.

The of SSW event in the middle of March 2020 which, although it did not satisfy the WMO definition of Major SSW event, prevented an even stronger ozone layer destruction that would have observed if this SSW event had not occurred or occurred later. The revealed enhancement of wave activity propagation over the Gulf of Alaska could be important but not a sole factor responsible for the onset of the SSW event in the middle of March 2020 as supposed only 30% / 60% of SSW events are preceded by extreme wave activity episodes at the lower troposphere / lower stratosphere (White et al., 2019). Therefore

numerical experiments are desirable to verify a role of this enhanced wave activity propagation in the generation of SSW event in the mid-March 2020.The weakened propagation of wave activity from the troposphere to the stratosphere could be caused in addition to the influence of Indian Ocean Dipole suggested by Hardiman et al., (2020) by the positive temperature anomalies in the troposphere over the north-eastern Eurasia. However, the weakening of wave activity propagation to stratosphere could be not only due to changes in troposphere but in the stratosphere too. Therefore this our speculation needs

further research.

Overall conclusions of our study are as follows:

- The 2019-2020 winter season was characterized by an unusually stable and cold stratospheric polar vortex in the Arctic with low temperatures close to the record values of the winter seasons of 1996-1997 and 2010-2011 over the past 40 years. In the lower stratosphere, temperature anomalies were up to -15 K low relative to climatic values in

February-March-April 2020. The reason for such low temperature of the Arctic stratosphere was the weakened propagation of wave activity from the troposphere, especially in February - early March 2020 and the downward reflection of wave activity in the upper stratosphere.

- The enhancement of wave activity propagation in the middle of March with source over the Gulf of Alaska, led to

the SSW event and to an increase in the Arctic stratosphere temperature, decrease of PSC volume and a slowdown





in the destruction of the ozone layer, a weakening and displacement of the stratospheric polar vortex to the north of Canada.

- According to the results of trajectory modeling, the largest ozone losses inside the stratospheric polar vortex in the Arctic were observed in the winter of 2019-2020 compared to the other two winter seasons with the highest ozone depletion in the last years 2010-2011 and 2015-2016.

- The decrease in the total column ozone inside the polar vortex, calculated using ozonesonde observations, reached $157 \pm 22$ DU by the end of March 2020, which repeated the record values of winter 2011 ($153 \pm 13$ DU) and exceeded the corresponding values for winter 2015-2016 ($135 \pm 40$ DU).

- The results of numerical calculations with a chemistry-transport model with dynamical parameters specified from the MERRA-2 reanalysis data, carried out according to several scenarios of accounting for the chemical destruction of ozone, reveal that both dynamical and chemical processes make significant contributions to the decrease in the ozone content inside the polar vortex. In this case, the chemical ozone depletion is determined not only by heterogeneous processes on the surface of polar stratospheric clouds, but by gas-phase destruction in nitrogen catalytic cycles as well.

**Code and data availability.** The data used in this study and IDL code for data reading and plotting can be downloaded from **ra.rshu.ru/files/Smyshlyev_et_al_ACP_2020**. The model codes are available from the authors upon request.

**Author Contributions:** All other authors had valuable contribution in visualization of the results and data analysis including in providing the data of chemistry-transport modeling (S.P.S., M.A.M.), trajectory modeling (A.N.L.); estimation of chemistry ozone destruction (N.D.T.); analysis of Arctic stratosphere dynamics (P.N.V). All authors have read and agreed to the published version of the manuscript.

**Competing interests.** The authors declare that they have no conflict of interest.

**Funding:** Global processes impact on the Arctic dynamics and chemistry was studied under the Russian Scientific Foundation grant #19-17-00198 (S.P.S., P.N.V and M.A.M), trajectory analysis and ozone loss estimation were supported by the Russian Foundation for Basic Research, grant no #19-05-00370 (A.N.L. and N.D.T.). Climate change impact on the Arctic processes was studied under the frame of the State task of The Ministry of Science and High Education of the Russian Federation (project # FSZU-2020-0009).

**Acknowledgments:** The authors thank P. von der Gathen (Alfred Wegener Institute for Polar and Marine Research, Potsdam, Germany), J. Davies, D. Tarasick (Air Quality Research Division, Environment and Climate Change Canada,



Ontario, Canada), and R. Kivi (Space and Earth Observation Centre, Finnish Meteorological Institute, Finland) for providing
the ozone verticals profiles data from Ny-Aalesund, Alert, Eureca, Sodankylä stations and the many observers whose
dedicated and careful work produced the sounding data. ERA5 reanalysis data were provided by the European Centre for
Medium-Range Weather Forecasts (ECMWF), and OMI data and MERRA-2 reanalysis were provided by the National
Aeronautics and Space Administration (NASA). NCEP Reanalysis data provided by Climate Prediction Centre (NOAA).
JRA Reanalysis data were provided by JMA Data Dissemination System (JDDS).

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
