# Peer review of "Dynamical and chemical processes contributing to ozone loss in exceptional Arctic stratosphere winter-spring of 2020"

_Atmospheric Chemistry and Physics, 2021_

## Referee Comment (RC1)

Dear authors,

I would recommend a major revision. This manuscript has to improve significantly before I can recommend it for publication.

I have to admit that I found it difficult to make a decision for this paper. Some parts of this paper make the impression that the authors have a good knowledge of the topic. Most of these parts are scientifically sound, and show knowledge of the literature. The structure and readability of the paper are acceptable and on an average level in these parts. Language could be improved, but I acknowledge that the authors are no native speakers (neither am I). There are also numerous small errors, omissions and inconsistencies, and this manuscript could need more care and attention.

But some other parts make the impression as if the manuscript was written by several different authors, with a different level of scientific background. These parts sometimes show a lack of understanding of the stratospheric processes and fundamental flaws in the methods. In particular, section 4 is in a bad shape.

While the topic is within the scope of the journal and of interest for the audience, another problematic issue is that large parts of the results shown in this paper have already been published elsewhere in similar or identical form. In the end, I am in favor of the authors, since I think that as a reviewer my main responsibility is to make sure that the results are scientifically sound. There is nothing wrong with confirming the results of others with different data sets or models. But some parts of this manuscript make the impression that the authors only reviewed the literature and rephrased the work of others, instead of doing original research. This is ok up to some point, but it should not be taken too far to avoid any negative connotations.

Best regards, Ingo Wohltmann

**Major comments**

1. Result that the contribution of gas phase nitrogen cycles to ozone loss in the polar vortex in spring is significant cannot be correct

The results of the noPSCaer CTM run for the contribution of gas phase nitrogen cycles to ozone loss in the polar vortex cannot be correct (section 4, e.g. your Figures 12-15 and accompanying text). It is basic textbook knowledge (if not to say the most important scientific result of research related to the ozone layer of the last decades) that chemical ozone loss in spring in the polar vortices is dominated by loss from catalytic chlorine and bromine cycles and caused by heterogeneous chemistry. Your results imply that $NO_x$ cycles contribute not only significantly to column ozone loss in the vortex in spring, but even dominate the ozone loss. If that would be true, it would mean that the Montreal Protocol and the ban of CFCs would have been unnecessary and that we should have seen large ozone loss before the rise of CFCs in the 1960s or 1970s.

Unfortunately, there is no sufficient information to figure out what is going wrong here. There is either a bug in your program code (in the CTM or in your scripts that produce the figures), or there are some fundamental flaws in your methods.

There are some candidates:

a) Not considering that $NO_y$ and $NO_x$ are not the same in the noPSCaer run without PSCs and the full run with PSCs (because PSCs change $NO_y$ and $NO_x$, e.g. by denitrification or changes in partitioning).

b) A problem with the initialization of the noCHEMall run that you use as a passive ozone tracer. It may be that you initialize the passive ozone tracer too early and get to much $NO_x$ chemistry from outside of the vortex or in early winter or late autumn into your estimates.

c) A problem with the upper model boundary, which may be too low to obtain meaningful values with the passive ozone tracer.

d) Using a vertical range for column integration which exceeds the range where the passive ozone column method gives reliable results (relates e.g. to c) or the ozone lifetimes getting shorter with increasing height).

I will go into more detail in the specific comments when necessary.

I would recommend either to solve this issue or, if it is not possible to find out what is going wrong here, to remove all discussion of nitrogen cycles and the noPSCaer run, and to delete the corresponding results from the figures.

There is plenty of literature showing how the results should look like. I will give you only one example. Figure 20 of Wohltmann et al. (2017) shows that more than 95% of ozone loss at about 50 hPa in the vortex is caused by the catalytic ClO-ClO and ClO-BrO cycles (for one Arctic and one Antarctic winter). You will find more figures for different altitudes and years in the supplement of this study, showing that this is a general statement.

For specific comments of relevance to this major comment, see e.g. comments to line 24-25, 184, 194-195, 197.

**2. Not sampling the polar vortex homogeneously when determining ozone loss**

In the method to determine ozone loss that you introduce in sections 2.2 and 3.2: For no apparent reason, you start trajectories only on the 85 degrees latitude circle instead of sampling the vortex homogenously (which would have been easy). This introduces a bias, which you could have easily avoided. Please see the specific comment to lines 332-333 for details. Please repeat the runs.

**3. Low novelty value and low scientific significance, redundancy, many repetitions of results from other studies, and the paper needs to be shortened considerably**

You are probably aware that many results which are similar or even identical to what is shown in your manuscript have already been published elsewhere (e.g. Manney et al., 2020, Lawrence et al., 2020, Wohltmann et al., 2020, Dameris et al., 2021). In my opinion, the main responsibilities of a reviewer is to make sure that the results are scientifically sound and that the paper is well structured and readable, and I will concentrate on these aspects in my review. It does not hurt when a paper confirms results of other studies. But I think that this manuscript can be considerably shortened, in particular when you only repeat results from other studies and do no original research on your own.

I find section 3.1 particularly problematic (see also specific comments). There are at least 5 Figures which have already been published in identical or very similar form in other manuscripts (where 4 figures alone are from Lawrence et al., 2020). Large parts of the text only repeat findings from other studies. Please shorten this section considerably and concentrate on your original research (that would be mainly the Plumb fluxes).

Section 4 is equally problematic. It could be shortened considerably. And it suffers from the fundamental flaws that I discuss in major comment 1.

The introduction is almost 3 pages and it is definitely too long. It is a little bit tedious to read. Try to concentrate on what is important for your study. Not all details are important enough to mention them here, e.g. the UV radiation in April in Moscow or the Indian Ocean Dipole. Your introduction would be very fine for a review paper and is a nice review on the literature for the winter 2019/2020, but this is maybe not the right place for it.

**4. Language**

This manuscript would benefit from the help of a native speaker. You have to improve on the use of the English language. Note that I am well aware that the authors are no native speakers and I won't base my judgement on this manuscript on the use of language by the authors, because this would not be fair. But it certainly will help to make the manuscript more readable and also to remove some confusion or misunderstandings. See also similar comment of reviewer #3 in the pre-review.

5. Numerous small errors, omissions and inconsistencies

There are numerous small errors, omissions and inconsistencies in the manuscript. While this can happen to everyone, I think that this manuscript would have needed some more care and attention and this would not have been necessary.

E.g., a figure is missing (Figure 2b), section numbering is not correct in one place (two sections 2.3), at least one paper in the references is not cited (Madrid et al.), there seems to miss a sentence in the author contributions, and so on.

6. Make sure that the title, abstract and conclusions reflect what you say in the main text:

When reading the main text, I discovered that you put quite an emphasis on the troposphere. This is totally fine, but it isn't really reflected in e.g. the title or abstract of the paper. Also, if you would like to put an emphasis on the situation in Russia, this is fine, but then it should be more clearly stated in abstract, title and introduction. Make sure that you don't disappoint the reader who has read the title and abstract and then does not really find what he expects in the main text.

**Specific comments**

Line 9-10:

State in the abstract which CTM you are using (i.e. the RSHU CTM). This will be of interest for many readers.

Lines 21-22:

The phrasing is a little bit misleading, since people often think immediately of "chemical ozone loss" when they read "ozone loss". Suggestion: "…indicated that both dynamical and chemical processes make contributions to ozone changes inside the polar vortex"

Lines 22-23: *"In this case, dynamical processes predominate in the western hemisphere, while in the eastern hemisphere chemical processes make an almost equal contribution with dynamical factors"*

I don't think that this is correct (at least in the way that it is phrased here, it is misleading). The simple reason for this is that the polar vortex is moving. Air masses that are in the western henmisphere at a particular date will be somewhere else a few days later. You have to make sure that you phrase that carefully. Ideally, one would follow the air masses inside the vortex and make statements for the vortex as a whole. That being said, it is of course a valid approach to look at specific locations, but then, for a single location, fast dynamical changes will often dominate. This is probably a question of what you use as a reference frame when you define the chemical and dynamical change.

Lines 24-25: *"the chemical depletion of ozone is determined not only by heterogeneous processes on the surface of the polar stratospheric clouds, but by the gas-phase destruction in nitrogen catalytic cycles as well."*

This would only be correct as a very general statement. But from the context is clear that you refer to ozone loss in spring in the polar vortex here and that you think that the contribution from $NO_x$ cycles is significant. The sentence is misleading and not correct in the end. You have to be careful about the message that you convey here. There is general agreement that chemical depletion in spring in the polar vortices is dominated by heterogeneous processes, see major comment 1. Delete this sentence.

Line 28:

Would be nice if you would not only cite references for the tropospheric influence, but also for the statement *"the circulation of the Arctic stratosphere in the winter-spring season (hereinafter winter season) is characterized by strong interannual and seasonal variability"*. Suggestions: e.g. Tegtmeier et al., 2008, Solomon, 1999.

Line 30:

Pedatella et al. is a news article and not a peer-reviewed paper. Delete the reference.

Line 38-39: *"the largest decrease in the Arctic ozone was observed…"*

This statement needs some references. For the 2019/2020 winter, e.g. Manney et al., 2020, Wohltmann et al., 2020. For 2015/2016, e.g., Khosrawi et al., 2017.

Line 62: *"record low temperatures were observed in the Arctic lower stratosphere, and, as a result, a record volume of Polar Stratospheric Clouds (PSCs) was expected"*

Again, this statement needs some references, e.g. Lawrence et al., 2020 or Wohltmann et al., 2020. And you surely not mean "was expected" but "was observed".

Lines 115-116: *"Signs of recovery in ozone levels began to be noted in the polar regions, in particular, a decrease in the depth of the ozone hole and its size in Antarctica. The 2019 ozone hole in Antarctica was one of the lowest in decades".*

These statements need some references.

Line 138:

The canonical reference for the ERA5 dataset is Hersbach et al., 2020. Your reference is outdated and not peer-reviewed.

Line 136-139:

It makes the impression to me that some of the meteorological analyses that you mention here are never used in the paper. Please do only mention the analyses that are actually used. In addition, please make sure that you give the analysis that you use to calculate quantities or that you discuss in the text at the appropriate places in the paper. This information is missing in several places.

Line 141-142:

State the analysis you used for the temperatures.

Line 159:

Why forward and not backward trajectories? Backward trajectories would probably have the advantage that you would lose less trajectories that leave the vortex. In addition, forward trajectories the disadvantage that at the end of the run, trajectories started at the same potential temperature level will be distributed over a range of potential temperatures. Since you would like to know the ozone loss at the end of the run at a certain level, this is unfortunate.

Section 2.2:

Important information is missing or only given later in section 3.2.

What are the initial locations of the trajectories? What is the start date? These two questions are answered in section 3.2, but you should either move this from section 3.2 to section 2.2. Or vice versa (you could delete this section then)

Did you use vertical winds or heating rates to calculate the vertical motion? What happens with trajectories that leave the vortex? What is your criterion for the vortex edge (some PV value)? These questions are not answered here or in 3.2.

It would also be important to have some more information about the ERA5 ozone product, since your results crucially depend on it. E.g. which measurement data are assimilated, how well does the ERA5 ozone product compare to observations?

Section 2.3:

Important information is missing, that is only given later in section 3.2. Please state the individual stations. What is the time period you are looking at and the starting date you use as the reference? What is your criterion for the vortex edge?

Again, you could move this information from 3.2 to 2.3. Or vice versa (and delete 2.3)

Line 172:

State the name of the radiative transfer model. Just "the radiative transfer model" is no sufficient information.

Line 170-172:

It would be appropriate to go a little bit more into detail here (even though this is an established method). So, you probably first interpolated all the ozone sonde measurements on an isentropic surface. When you fit the linear regression line to the timeseries, you probably don't use all of the ozone data but a time window around some given date to be able to obtain a value for the rate of change for a specific date, I suppose? What is the length of the time window?

Line 175:

I wasn't able to find the reference Tsvetkova et al. (2004) at the home page of the journal. Please replace the reference by something easier to access. I think there are many articles describing the method.

Section 2.4 (mislabeled as a second Section 2.3 in the manuscript):

Again, important information is missing. When did you start the model run? See also comment to lines 194-195.

Line 180:

Smyshlyaev et al., 2017 is only available in Russian language (at least when I access it through the doi). This is very unfortunate, because I can't read it. I don't know what the guidelines of ACP are regarding this, but possibly you have to remove this reference.

Line 183:

5 degrees x 4 degrees is an extremely coarse resolution and is not state-of-the-art anymore (maybe it was 15 years ago). Is it possible to run the model in a higher resolution (say 2 degrees x 2 degrees or 1 degree x 1 degree) to exclude that the coarse resolution has any negative effects on the results?

Line 184:

You should make sure that you don't look at air masses for your "passive ozone tracer" from the noCHEMAll run that were initially (at the start of the model run) at the upper model boundary or above, because this leads to meaningless values for the "passive ozone". From model runs that I performed for 2019/2020 with an upper boundary at 50 km, I estimated that values for the ozone loss above 550 K are not reliable anymore with the "passive ozone tracer" method.

Line 186-187:

It is a rather unusual choice for the PSC scheme that it is based only on STS clouds. Can you elaborate a little bit on the reasons for that (not only in the reply, but also in the manuscript)? Why don't you simulate NAT and ice clouds in addition? Note that I am aware that the addition of NAT and ice clouds would probably only have a small effect on your results, since the heterogeneous reactions are usually sufficiently fast and since the temperature dependence is similar for NAT and STS clouds. But you possibly introduce some uncertainty by this, this should be discussed.

Lines 194-195:

This is not quite clear to me. Do you want to say that you initialize your "passive ozone tracer" from the noCHEMall run for estimating ozone loss on 1 November? But only north of 64 degrees? What are you doing in the following days? Calculating chemistry south of 64 degrees and then switching chemistry off when air masses are transported inside the 64 degree latitude circle?

This is of particular importance for your method to determine ozone loss. You set the reference date here for your determination of ozone loss. Ozone loss is extremely sensitive to the start date for the passive ozone tracer. If you choose a date too early, you will get a significant contribution from ozone loss from outside of the vortex caused by $NO_x$ cycles (since the air masses that are inside the vortex and the end of the model run would have been far outside the vortex at the start of the model run). Or you get loss from $NO_x$ cycles in autumn before the formation of the vortex, when there still is sunlight. And if you set the start date too late, you miss some ozone loss. In my experience, a date between 15 December and 1 January works best for 2019/2020. November 1 is probably much too early.

And please state clearly what you are doing here. I.e., setting the reference date for calculating ozone loss. This is probably not clear to the majority of readers.

If you switch off chemistry only north of 64 degrees, that makes it hard to reason about the results. This way, your passive ozone tracer will not really be "passive", but some mixture of air masses that did experience ozone loss and air masses that did not. This makes it hard to understand what is actually shown in Figs. 12-15. Certainly not just "the" ozone loss.

Maybe this method would have worked if you would have chosen a PV contour at the edge of the vortex instead of 64 degrees. But it still would be a problem that there is probably ozone loss by $NO_x$ cycles in November at high latitudes, since there is still sunlight.

But in any case, this would need much more explanation. The reasoning behind your setup is not at all obvious to the reader. I think I figured it out after some time (you want only to count ozone loss in the vortex and get rid of the loss by $NO_x$ cycles outside of the vortex by switching on chemistry south of 64 degrees), but that does not get clear at all. And probably, 64 degrees as a boundary will not work because this always includes air from outside the vortex.

Line 197: *"Comparison of the baseline scenario with these two additional scenarios makes it possible to estimate the periods when the chemical destruction of ozone is most effective after heterogeneous activation on the PSC surface, and when the gas-phase destruction of ozone in nitrogen catalytic cycles is more significant."*

You are very probably not doing this correctly or as you intended it. Switching off heterogenous chemistry on PSCs (or not forming PSCs at all in the model) will also affect the partitioning and chemistry of $NO_y$ and $NO_x$, and it will affect denitrification. E.g., the heterogenous reaction $N_2O_5+H_2O$ will be important for the partitioning. And effectively switching off denitrification will lead to higher $NO_x$ in the "noPSCaer" run, which could lead to more ozone depletion by $NO_x$ compared to the reference run. This way, you will obtain a different result as if you have kept $NO_x$ constant.

What is not quite clear to me: Do you also switch off chemistry on the binary background aerosol at higher temperatures (which would also lead to unwanted changes)?

It is very likely that you will not obtain the result that you are hoped for: A clean separation of the amount of ozone that is depleted by chlorine and bromine cycles from the amount of ozone that is depleted by $NO_x$ cycles.

Therefore alone, I would recommend to delete all of the discussion on the contribution of the nitrogen cycles to the chemical ozone loss. I don't think that results are reliable.

A clean way to do this correctly would be to keep track of it in the chemistry module of your model, by e.g. looking at the rates of the rate-limiting steps of the different catalytic ozone destruction cycles. A study that shows how to do this correctly and that shows that heterogenous chemistry dominates ozone loss is Wohltmann et al., 2017.

Lines 207-210 *"On the other hand, in the absence of the Sun, the chemical destruction of ozone does not yet reach high values associated with the previous halogen activation on the surface of polar stratospheric clouds, therefore, it can be assumed, that there should not be extremely low values of TCO in the region of absence of observations by the OMI instrument."*

Delete this sentence. This is not correct.

It is well known that ozone values inside the vortex are often relatively homogeneous. It is also well known that the movement of the vortex and the movement of air inside the vortex cause a homogenization. Air is often processed by the PSCs like in a "flow processor", and air masses are transported into the sunlit regions of the vortex and move back into the dark regions again. Of course, there are exceptions, and air masses may sometimes remain in darkness for a long time. But what you are doing here is pure speculation.

And you don't even need to speculate. There are measurements from e.g. the MLS instrument which show the regions that OMI can't measure (and which already have been used for studies of this winter). Why don't you base your discussion on these measurements?

Lines 214-215: *"Again, the minimum values are detected along the border of the region of absence of observations - the zone of polar night."*

See above. Delete this statement.

Lines 202-220:

While I won't judge your paper by relevance, I wonder whether this detailed description of the position and movement of the polar vortex is really necessary. I don't really see the scientific significance of these results. In addition, this can easily be deduced from Figure 1. Furthermore, the development of the vortex has been described elsewhere in studies that already have been published, e.g. some figures in Manney et al., 2020, and more importantly, Dameris et al., 2021. Their figure 1 has a large similarity to your Figure 1.

Line 227: *"The winter season 2019-2020 in the Arctic stratosphere was one of the coldest in the last 40 years."*

Give references for this statement (e.g. Wohltmann et al., 2020, Lawrence et al., 2020)

Line 229:

Would be better to speak of STS and NAT clouds and not of Type I clouds for clarity.

Line 232:

Figure 2b is missing.

Lines 245-246: *"The first period (February 7 - March 7) corresponds to strongest weakening of wave activity propagation in 2020, the second: January - February, the third: January - March.".*

Sorry, but I have no clue what you want to say to me here. Is a part of the sentence missing?

Figure 3:

I would find it helpful to have some kind of colorbar for panels (a) and (c). Or at least to have the unit directly in the plots. It is given in the caption, but it is a little bit hard to bring this together.

Line 258: *"Where the absolute maximum of the average temperature in February was reached."*

Sorry, but again, I can't follow you. Do you mean that the highest temperatures over the course of the year were reached in February in Siberia. Or does the maximum refer to location? Please clarify.

Line 260:

Do you mean "obtained" and not "retained"? Or what do you want to say?

Large part of section 3.1 (Lines 221-end and Figures 2-4):

Without going through Lawrence et al. (2020) in detail, I have the impression that almost all of your section 3.1 only repeats what has already been written in the text and shown in the figures in Lawrence et al. Only for example: Your Figure 2 is Figure 11a from Lawrence et al., your Figure 3a is similar in meaning to Figure 7f from Lawrence et al., your Figure 3c is Figure 6a from Lawrence et al., your Figure 4a is Figure 3a from Lawrence et al. While I think this was probably not intentional, I think this is problematic. You should really think about shortening this section, to delete some of the figures and to refer to Lawrence et al. where appropriate.

Section 3.1: Meteorological data:

What is the reanalysis data that you are using here for the temperatures and the EP fluxes? You only make a very general statement in section 2 that you use ERA5, NCEP, JRA and MERRA, and don't state anything in section 2.1. It makes the impression that you use a particular reanalysis here. Which one?

Line 267: *"and the negative one, on the contrary".*

Sorry, I can't follow you here. What do you want to say? That the negative phase shows opposing changes? I don't think you need to state that. This is obvious and follows from the definition of the AO.

Line 267-270: *"With a positive AO phase, a stronger western zonal transport leads to milder winters, but with more precipitation in Southern Europe. In the negative AO phase, this transfer is weaker; as a result, cold air masses from the Arctic spread more strongly to the territory of Europe."*

It seems to me that this needs a reference.

Line 271: *"AO is the result of interaction between the dynamics of the stratosphere and the troposphere."*

I don't know if I would phrase it like this. I would suggest to write something like "Interaction between the dynamics of the stratosphere and the troposphere can cause changes in the AO" or that the changes in the stratosphere (polar vortex strength) and troposphere (AO) are closely correlated.

Line 273-274: *"which is facilitated by an increase in the temperature gradient between the heated by the sun and shaded parts of the atmosphere"*

I have no idea what you want to say here. Do you mean "associated" again and not "facilitated"? That would make more sense. But why do you mention the sunlit and dark parts of the atmosphere in

conjunction with the temperature gradient? Assuming that you are talking about the zonal temperature gradient in the stratosphere, of course the polar vortex is in the end caused by radiative cooling in the polar night. But I think you are talking about interannual or seasonal changes in the temperature gradient related to polar vortex strength, which are not caused by changes in solar illumination (which is the same in every year).

What do you mean by "zonal mean flux"? Flux of which quantity?

Lines 275-276:

What is cause and effect here? Less wave activity means a stronger and undisturbed vortex, and that in turn means altered conditions for wave propagation. Again, probably better to speak of correlation or association.

Line 284:

Up to here, you talk of the AO. Now, you suddenly start to use the term NAM, which is just another word for the AO. And instead of talking of an AO index value which is just a number like 4 in the paragraph before, you start giving the value as 1.5 sigma. But probably I am not wrong when I suppose that these indices measure the same quantity. And what is sigma? The standard deviation of the AO time series, I suppose?

Line 284-285:

I have no idea what you mean by "spread continuously". Do you mean "propagate downward" or "extend into the troposphere"? Or that there is a clear signal in this time period?

Line 286:

At least in the stratosphere, the plot shows high values of the AO index up to the end of April.

Line 286:

You suddenly talk of a SSW in March which you never have mentioned before. It would be helpful to introduce the warming before you refer to it. It would also be helpful to have a reference or some more explanation. As far as I can see from e.g. PV maps, the vortex was quite stable until the end of April (although there was warming at the end of March).

**(Note: Lines 288-322 are not really my area of expertise. I hope another reviewer can say more to this. I cannot judge whether the results are scientifically sound or not)**

Line 290-292: *"It is known that the main source of wave activity propagation into the stratosphere, characterized by the maximum of the vertical component Fz of Plumb's fluxes, (e.g. Jadin 2011) is located over this region [north-Eastern Eurasia]."*

This is not really my area of expertise and this may be correct. But this is a rather bold statement, and I have never heard of this before. Unfortunately, the only reference that you give is from a predatory journal, and therefore, is no reliable source and I refuse to read it. Please give references from legitimate peer-reviewed sources or delete this statement.

In any case, remove the reference.

Lines 294-295:

Since you already stated this, you should refer to your earlier statements. E.g. "As shown above…"

Figure 4a:

The green contour mentioned in the caption is missing from the plot. The units for the colorbar are not given (neither in the plot nor in the caption).

Figure 4b and c, Figure 5a and b:

Same comment as to Figure 3. Would be helpful to have the units for the contours directly in the plots.

Lines 306-307:

You are again talking about a SSW event which you never have introduced before.

Lines 331-339:

Seems like part of the description of your method is in these lines and the other part is in section 2.2. Can you please describe the method only at a single place? Either, you have to move these lines to the description of the method in section 2.2. Or vice versa (and then delete section 2.2).

Some of the questions I had for section 2.2 are answered here, but others are not (see also following comments and comment to section 2.2)

Lines 332-333: *"For simplicity the trajectories were initiated uniformly distributed on the 85 N latitude circle, when it was completely located inside the polar vortex"*

For no apparent reason, you start trajectories only on the 85 degrees latitude circle instead of sampling the vortex homogenously, which would have been easy. This introduces a bias which you could easily have avoided. Please repeat your calculations with trajectories that sample the vortex homogeneously.

You need to test whether a trajectory is inside the vortex in any case, so that should not introduce any additional effort. Unfortunately, you don't give any information on your criterion for testing whether a trajectories is located inside the polar vortex. Do you use a fixed PV contour (say 36 PVU), equivalent latitude or the edge as defined by Nash?

It should also be easy to sample the vortex homogeneously, e.g. by starting trajectories on more than one latitude circle and starting less trajectories on latitude circles closer to the pole to make sure that every trajectory represents the same area, or by using a random generator to distribute points evenly (the only thing you have to take care of is taking the arcsin of the random number for latitude).

Line 335: *"and mostly remained inside the vortex"*

You give no information what you do with trajectories that leave the vortex. Do you ignore that or do you sort them out? This is potentially important for the results.

Line 332, 339:

You start the trajectories only on two levels, 475 K and 550 K. I think that would have been fine if you would have used backward trajectories, but it is unfortunate with forward trajectories. The quantity that you usually would like to know is the ozone loss at some level in spring at the end of your trajectory run (i.e. in a given well defined air mass). The trajectories which you start at 475 K or 550 K will not only descend, but will also cover a range of potential temperatures (and horizontal locations) at the end of the run, which makes the ozone loss hard to interpret.

The very least you could do is to give the range of potential temperatures that is covered by the trajectories at the end of the run (and the mean value to estimate the diabatic descent).

It would however be much more straightforward to base your method on backward trajectories.

Line 338: *"average ozone value"*

I assume you mean the average over all trajectory locations at this date?

Line 355-356:

I would expect this information earlier in the description of the method.

Line 358: *"overestimation".*

You don't know whether this is an overestimation (compared to "reality") or not. I would say "leads to higher estimates for…"

Line 367-368:

Please give the exact dates. Does beginning of January mean January 1? What is end of March?

Line 369: *"Significant ozone loss had been seen from the mid-January till the end of March between 400-525 K isentropic levels (~15-22 km)."*

Where does the information mid-January until end of March come from? Is this your result from your method? This cannot be deduced from the figure, so you should state that more clearly. E.g. "Analysis of the ozone sonde data shows that significant ozone loss is observed from mid-January to end of March between…"

Lines 377-378:

Are these values your results or the results of Peters (2010)?

Line 382-384: *"The use of temperature, wind speeds, surface pressure and air humidity from the reanalysis data made it possible to simulate the effect of atmospheric circulation on the transport of ozone and associated gases close to reality. Variability of specified dynamical parameters determines the dynamical decrease in ozone content, as well as the atmospheric temperature govern the rate of chemical reactions, polar stratospheric clouds formation and the rate of heterogeneous reactions on their surface, which determine the chemical destruction of ozone.".*

Delete these sentences. They have no information content and are phrased awkwardly. Everybody in the community knows what a CTM is good for. The information given here is much too basic.

Figure 9:

There is a strange zig-zag pattern in some of the contours in the plots. It seems that there is a bug in your plotting or interpolation functions.

Figure 9:

Would be nice again to have the units in the plots.

Figure 9:

Please indicate the vortex edge in the plots.

Line 390: *"First, a basic numerical experiment was performed taking into account all factors affecting Arctic ozone".*

I think what you wanted to say here is: "First, a reference run with full chemistry was performed." Your sentence sounds odd.

Line 390-391: *"The variability of the atmospheric gas composition during the winter of 2019-2020 was calculated in the basic numerical experiment"*

Again, you state the obvious. Delete the sentence.

Line 395-396: *"In particular, its movement in the eastward direction and the minimum values at the beginning of March in the Northern part of the European territory of Russia are reproduced".*

Well, if you would not reproduce these very basic features, you would have a problem anyway. I don't know whether this sentence is really needed.

Line 396-397: *"In mid-March – in the western part of the Arctic, in the area of Greenland, Svalbard and Franz Josef Land in early April and North to the mainland of the ETR in mid-April."*

This is not a complete sentence, and therefore, unintelligible. Delete. And if it would be a complete sentence, I have the feeling that the information given here would be irrelevant. You don't need to state the position of the vortex every few days and every location that it covers. This is not only easily visible in the plots, but also no relevant or scientifically interesting information in my opinion.

Line 399-400: *"The model results demonstrate that the minimum values are observed at the boundary of the polar night in the part where the Sun has already returned"*

I find this statement problematic since the vortex is constantly moving. And I don't think it is correct, see my earlier comment on line 207-210. It also hard to see in the plots because there is no line showing the area of polar night. Delete the sentence.

Lines 400-402: *"This confirms the hypothesis of the effect of the chemical destruction of ozone, which intensifies after heterogeneous activation in polar stratospheric clouds and the return of the Sun."*

This is basic textbook knowledge that everybody who reads this paper is aware of. That would be fine for the introduction, but not for the main text. In addition, it is phrased quite awkwardly, up to the point that it is not quite correct or very hard to understand what you mean. Delete the sentence.

Lines 402-404: *"However, the results of model calculations reveal that relatively low values of the total column ozone (below 300 DU) are also observed in the polar night zone, where the chemical destruction of ozone is slowed down. This also indicates a significant influence of dynamical factors on the formation of regions of low ozone content."*

This is again basic knowledge, misleading and scientifically not relevant. I start to wonder whether the author of this section has a basic lack of understanding of the relevant science.

The total column value is a cumulative quantity, which is not determined by the position of the vortex at a particular date. In addition, the Arctic vortex is more dynamically active than the more circular Antarctic vortex, where it is more likely that air masses stay in darkness for a long time.

I think the only part of interesting information here would be the minimum column values. I also acknowledge that the discussion of dynamical factors can be interesting, but that would include things like interannual variations in diabatic descent or ozone mini-holes. The simple fact that air inside the vortex is moving and that the vortex is moving as a whole does not belong to this. This could be interesting if the authors would have done a detailed trajectory study of the history of air masses, but stating the obvious here is not enough in my opinion.

Line 406-407 *"To better understand the relative role of dynamical and chemical processes in the formation of the Arctic ozone anomaly in spring 2020"*.

I don't understand why it leads to a better understanding of the *relative* roles of dynamics and chemistry to look at the PSC area, i.e. what has the discussion in the following paragraph to do with that?

Line 407:

"Type I cloud" is ambiguous. Please clarify whether you mean STS or NAT clouds.

Line 408-409: *"but the inertia in their melting with increasing temperature is also taken into account."*.

Would be worth noting here that your CTM does not use an equilibrium scheme as some other CTMs and to cite Smyshlyaev et al., 2010.

Lines 414-416: *"which suggests that the relationship between the formation of PSCs and ozone destruction is not linear and confirms the theory of several stages of the formation of ozone anomalies."*

Apart from the fact that this sentence is phrased very awkwardly, this again states textbook knowledge and the obvious. Delete the half-sentence.

Line 416-417: *"In the polar stratosphere, associated, first, with the formation of PSCs, then with halogen activation on their surface, and only then with ozone destruction after the return of the Sun after the polar night."*

Again, this is basic knowledge. Delete the sentence. This would be fine in the introduction, but not as a scientific result in the main text. And it is phrased so awkwardly that it is almost unintelligible.

Figure 10, Figure 11:

Indicate the edge of the polar vortex. It seems to me the area covered by PSCs is much larger than the polar vortex. Is this really correct?

Figure 10, Figure 11:

Again, give the units in the plots.

Figure 11 caption:

"low stratospheric coefficient of ozone destruction" does not give enough information to find out what is shown here.

Line 425: *"the coefficient of chemical ozone destruction in the lower stratosphere"*

This is introduced as it would be a well-known quantity (known by everyone under this name), but I think in fact it will cause confusion for many readers. E.g., what does "lower stratosphere" mean here? There is no altitude range mentioned in the following. This is important information that you need to give here.

Line 425-426: *"This coefficient is a factor by which the concentration of ozone should be multiplied in order to obtain the rate of its chemical destruction."*

This definition seems not to be consistent with the magnitude of the values shown in Figure 11. If I understand you correctly, a value of 1/s would mean that all of the ozone at a particular location would be depleted by chemical processes in 1 second. But the figure shows values on the order of $10^8$ per second. Either I have difficulties to understand your definition or there is something wrong with the magnitude of the values given in the plot.

Lines 425-426:

Just to make sure that nothing is going wrong here. You take into account that there is a fast equilibrium between O and $O_3$ and only look at the net change of $O_3$?

Lines 425-426:

Do you show and discuss instantaneous values at a given point in time (say 12 UTC) or daily averages? The plots make the impression that the latter is the case. But you don't tell us anything about that. This is important. Please clarify.

Lines 425-426:

I have the impression there might be quantities that would be more easy to understand which you could show here, e.g. simply the rates in ppb/day or a similar unit, or the reciprocal of what you show

here (the time scale needed for complete ozone destruction). But maybe my confusion is just caused because I have difficulties to understand your text. A better explanation and definition may help here.

Lines 428-429: *"It should be noted that the rate of ozone destruction in March has its maximum values at the boundary of the polar night in the region of the newly returned Sun."*

This is hard to see, because the boundary of the area of polar night is not shown in the plots. And looking at Figure 11, I doubt that this statement is correct (assuming that you show daily averages). Delete the statement.

Lines 430-434: *"In this case, the maximum rate of ozone destruction is a necessary, but not sufficient condition for the formation of a zone of low ozone content in the spring. Dynamical factors also play an important role, in particular, for definition of the zone where the polar vortex is located. In particular, in early March, the rate of destruction of the base is maximum over the entire circle of latitude near the boundary of the polar night, and the minimum values of the ozone content are noted only in the eastern hemisphere (Figs. 8 and 9). Also in mid-March and April, areas of high ozone depletion cover a wider zone than areas of minimum total ozone."*

Delete all of these statements. First of all, you obviously can't deduce the cumulative chemical destruction of ozone over a longer time period (that causes the low ozone columns) from the chemical rates at a single date, because the values add up. You don't need to argue with dynamical reasons here. This is really a basic flaw in your reasoning here.

And with respect to dynamics: And again, you are stating the obvious here. The polar vortex and the air contained in the vortex are moving. It makes no sense to note that the minimum values of ozone are over the eastern hemisphere in this context. The vortex is moving, and a few days later, this may look totally different.

Lines 436-439: *"For a more detailed study of the influence of dynamical and chemical factors on the local variability of the ozone content Figures 12 - 15 present the simulated with the CTM and measured by the OMI instrument changes in the total ozone content at four stations (two in the Western Hemisphere and two in the Eastern Hemisphere) during six months from the beginning to mid-2020."*.

I have a general comment here: While it is certainly fine to perform case studies like this for single locations, it would be have been so easy here to make more general and scientifically relevant statements by looking at vortex means (which should have been easily possible). I don't really see why the situation at a particular location is so scientifically interesting, but that may be my personal opinion. I think that you wasted a chance here without necessity.

Lines 436-439:

It seems to me that the information given in Figures 12-15 is largely redundant. Figure 12 and 13 (Pechora and Tura) show almost identical results. The same is true for Figure 14 and 15 (Resolute and Eureka). You could easily do with only two figures here. I would suggest to shorten the text in lines 450-526 significantly.

Line 440:

Is there any reason why you use SBUV data here and OMI data earlier in the manuscript?

Line 442:

Earlier in the manuscript, you give a value of 64 degrees N and not 66 degrees N. Only one of these values can be correct. Please correct.

Lines 444-446: *"Comparison of the baseline scenario with these two additional scenarios allows us to estimate the periods when chemical destruction of ozone is most effective after heterogeneous*

*activation on the PSC surface, and when gas-phase destruction of ozone in nitrogen catalytic cycles is more significant"*

See major comment 1. There is something fundamentally going wrong here.

Lines 446-448: *"In addition, the comparative role of dynamical and chemical processes of ozone reduction can be assessed by comparing these scenarios with each other and to mean climatic values presented at the bottom of these figures."*

While I agree that you can disentangle dynamical and chemical changes when looking at these plots, this sentence is easy to misunderstand. E.g. you can't assess the comparative role of dynamical and chemical processes from comparing the "PSC" and "noPSCaer" runs.

Lines 450-526:

I think this part can be shortened considerably, not only for the reason stated above (lines 436-439). It is a little bit tiring that this is basically a description of what you can see in the figures (in very much detail), without so many scientifically interesting results.

Line 461-463: *"Based on a comparison of the noPSCaer and noCHEMall scenarios, it can be concluded that in the chemical destruction of ozone at Pechora station, the heterogeneous part is about one third (~ 25 DU), and the gas-phase part is ~ 45 DU."*

See major comment 1. This can't be correct. Unfortunately, you don't indicate in the plots when the station is located inside the polar vortex.

Figures 12-15:

Indicate when the station is in the polar vortex. This is *very* important to be able to interpret the results correctly.

Figures 12-15:

In case you don't find the problem, remove the noPSCaer run from the plots.

Figure 12-15:

In the b panels, I would have found it more intuitive when the blue line would have been the difference between "noCHEMall" and "noPSCaer".

Lines 471-481:

This is largely redundant with the paragraph about Pechora. I won't comment in detail and suggest to delete this.

Line 495: *"It should also be noted that there are two peaks of maximum chemical destruction of ozone: in late March and mid-April."*

This is only the observation at this location because of the movement of the vortex. If you would look at the same air mass, this would be different.

Line 496-500: *"At the same time, chemical destruction in the second half of March is superimposed on a dynamic decrease in its content, which leads to a minimum in the seasonal variation of the total ozone content, while in April, when the chemical destruction of ozone is even greater than in March, the polar vortex is already shifting towards the eastern hemisphere (Fig. 8 and 9), and the total ozone content is higher than in March."*

I find this sentence very hard to understand and unintelligible. For example, what do you mean by minimum in seasonal variation?

Line 507-509: *"Comparison of calculations for different scenarios of accounting for the chemical destruction of ozone depicts that the destruction of ozone over heterogeneous reactions in the western hemisphere exceeds 30 DU, which is more than in the eastern hemisphere, while the gas-phase destruction of ozone in the Western hemisphere is greater than in the Eastern Hemisphere."*

Delete these sentences. See major comment 1. There seems to be a fundamental flaw in your method.

Lines 510-513: *"It should also be noted that in the Western Hemisphere, the minimum values of the ozone content according to satellite measurements in March are lower than the values calculated using the model, while in the Eastern Hemisphere the satellite and model results are closer. This result may be due to relatively coarse model resolution to simulate fine local effects in the western hemisphere."*

Delete these sentences. You show a fundamental lack of understanding of the processes here. Since the vortex is moving, air masses that are located in the eastern hemisphere will be located somewhere else a few days later. This has nothing to do with the hemispheres.

Line 521-524: *"Additional numerical calculations to assess the effect of various catalytic cycles of chemical ozone destruction on a decrease in its content in April-May 2020 revealed that the main increase in the gas-phase ozone destruction occurs in the nitrogen catalytic cycle, in which the chemical reaction with the participation of nitrogen dioxide and atomic oxygen plays a determining role."*

Since there is a fundamental flaw in your method (major comment 1), these results are very likely not correct. Delete this sentence.

Lines 524-526: *"In the Arctic stratosphere, in contrast to the Antarctic stratosphere, significant denitrification does not occur, and therefore a sufficient amount of nitrogen oxides remains in it, which plays a decisive role in the destruction of stratospheric ozone."*

This statement is not correct, and it would have been easy to see that if you would have looked into the literature (e.g. Manney et al., 2020). Note that this statement is not correct in general, and not only for the winter 2019/2020. There are many Arctic winters which show a significant amount of denitrification, this is basic knowledge. Delete the sentence.

In addition, it seems that you use it here as an (wrong) explanation for your flawed results. I wonder why you did not notice that something must be wrong here.

Lines 528-531:

This sentence is phrased so awkwardly that I have a very hard time to understand what you want to say. It is almost unintelligible. Please rephrase. I won't give a suggestion here, because this is the conclusions and I am not sure what you want to tell us.

Line 535: *"The of SSW event in the middle of March 2020"*

This part of the sentence makes no sense. What do you want to tell us?

Line 535: *"although it did not satisfy the WMO definition of Major SSW event"*

Earlier in the paper, I had some comments that you were referring to a SSW event that you never mentioned before. And now, in the last lines of the paper, you tell me that it actually was no SSW event. What does this mean? That I can forget about everything that I have read about the SSW event? This information should have been given much earlier.

I agree that the warming of the vortex in late March led to a relatively abrupt stop of chemical ozone depletion. Can you rephrase this.

Lines 537-545, 554-557:

This is not my area of expertise. I will skip this part.

Lines 558-560:

You need to be more specific here. You have only results for two potential temperature levels. You don't mention that you refer to vortex means. You don't mention the date you are referring to. And give numbers for the ozone loss.

Line 566-569: *"...reveal that both dynamical and chemical processes make significant contributions to the decrease in the ozone content inside the polar vortex. In this case, the chemical ozone depletion is determined not only by heterogeneous processes on the surface of polar stratospheric clouds, but by gas-phase destruction in nitrogen catalytic cycles as well."*

This is not correct and misleading. See major comment 1. Delete this from the conclusions.

Line 573:

It seems that there is something missing in the "Author contributions". It starts with "All other authors…", implying that a sentence is missing at the start. There is information missing who has written the main text.

**Technical corrections (language etc.)**

Title:

"Dynamical and chemical processes contributing to ozone loss in the exceptional Arctic stratosphere winter-spring of 2020" (added "the")

Line 8:

You can delete "The features". Just start with "Dynamical processes and changes…"

Line 17:

"repeated" is probably not the perfect choice of word. Maybe "which was similar to the depletion in 2010/2011"

Line 33:

Change "the main SSW" to "a main SSW"

Line 38:

Change: "the largest decrease in the Arctic ozone was observed" to "the largest decreases in Arctic ozone were observed"

Line 49:

You certainly mean "statistically" and not "statically"

Line 52:

You probably mean something like "nevertheless" and not "in the meanwhile"

Line 132:

Change "reveal" to something like "estimate" or "determine"

Line 148:

Change "the Lagrangian approach" to "a Lagrangian approach"

Line 161:

Change "were interpolated into the points of each trajectory" to "were interpolated to the positions of each trajectory"

Line 164:

Change to "Ozone sonde data … have been used"

Line 168:

You misspelled the name in the reference. The correct name is in line 175 (Braathen). There is also a superfluous ","

Line 176:

Section 2.4 is mislabeled as Section 2.3

Line 182:

Split the sentence and shorten: "Meteorological fields are specified…"

Lines 184-186:

Awkward phrasing. Change to e.g. "The model includes 74 oxygen, hydrogen, nitrogen, chlorine, bromine, carbon and sulfate species. The chemistry of the species is calculated as described in Smyshlyaev et al. (1998)."

Line 191-192:

Again, phrased awkwardly. Suggestion: "For a more detailed study of the influence of dynamical and chemical factors on the local variability of the ozone content, two additional numerical experiments with the RSHU CTM were performed in addition to the reference run (termed "PSC" here)."

Line 204:

Change "at the early March" to "in early March"

Line 206:

Do you mean "north of Alaska"?

Line 207:

Maybe "which are based on solar radiation" is better English.

Line 215:

Change "north to" to "north of"

Line 217:

Change "values less than 220 DU" to "values of less than 220 DU"

Line 220:

Change "territory" to "area"

Figure 1 caption:

The text speaks of OMI and the caption speaks of AURA. Would be nice to have that consistent.

Line 229:

Split sentence. Write something like: "Temperatures were sufficiently low to allow the formation of NAT and STS clouds".

Line 229:

Can we stick to Kelvin and not to degree Celsius?

Line 230:

Change "Figure 2a" to "Figure 2". There is only one panel here and 2a does not exist.

Line 232:

Do you mean "Two main causes of the cold and stable Arctic polar vortex" and not "Two main causes of so cold and stable Arctic polar vortex"? Or what were you trying to say?

Figure 2 caption:

Change "climate mean" to "climatological mean"

Figure 2 caption:

There is a Russian letter in the caption (probably means "and")

Line 242:

Change to "Furthermore, we compare…"

Caption Figure 3:

Change "Latitudes are from 30 N" to "The map shows only latitudes north of 30 N"

Line 263-264:

"Notably that described positive temperature anomalies were observed not only near surface but at higher levels in troposphere." This is phrased awkwardly. Suggestion: "Positive temperature anomalies were observed not only near the surface but also at higher levels in troposphere."

Line 266:

Change "and increased in" to "and increased pressure in"

Lines 273-274:

You probably mean "between parts of the atmosphere that are heated by the sun and parts that are shaded"? But I think a native speaker probably wouldn't phrase it like this. I would talk of the sunlit part.

Line 294:

Change "In the same time" to "In the same time period"

Line 304:

Change "till" to "until"

Line 305:

Change "This is confirmed by the diagram with…" to "This can be seen in Figure 5a showing …"

Figure 4 caption:

Change "climate mean" to "climatological mean"

Line 316:

Change "display dominated" to "show pronounced"

Line 334:

Change "descent" to "descend"

Figure 7 caption:

Change "0 day…" to "The horizontal axis shows the number of days since December 1."

Line 358:

Change "average vertical descending" to "average vertical descent" (this time the "t" is correct!)

Line 360:

Change "As well to estimate chemical ozone loss" to "As another method to estimate chemical ozone loss"

Line 361:

Change "Ny-Älesund" to "Ny-Ålesund"

Line 367:

Change to "Figure 8 shows the vertical profile of the vortex-averaged cumulative ozone loss…"

Line 369:

"with largest losses": Start a new sentence and write "These winters showed the largest ozone losses previous to the winter 2019/2020."

Line 371:

Split into two sentences. "…than in 2010/2011. That is consistent…"

Line 380:

Awkward phrasing. Change "For a more detailed study of the degree of dynamical and chemical processes influence on the formation of ozone anomalies…" to "For a more detailed study of the dynamical and chemical processes that influence the formation of ozone anomalies…"

Line 381-382:

Awkward phrasing. Split into two sentences. Change "in which the dynamic parameters were set from the MEPRA-2 reanalysis data" to "Meteorological data were obtained from the MERRA-2 reanalysis".

Line 382:

Note the change "MEPRA-2" to "MERRA-2" in the previous comment

Line 391-392:

Change "Figure 9 demonstrates" to "Figure 9 shows"

Line 392:

Awkward phrasing and a lot of repetition of information: Change "the results of calculations of the total column ozone for March-April 2020, performed using the CTM with the specified dynamical parameters from the MERRA-2 reanalysis" to "… shows the total ozone column for March-April 2020 from the CTM".

Line 410:

Replace "territory" by "region" or "area"

Line 411:

Change "the area of the PSCs zone is maximum" to "the area covered by the PSCs is maximum"

Line 413:

Change "the area covered by PSCs significantly reduced" to "the area covered by PSCs are significantly reduced"

Line 425:

Change "Fig. 11 demonstrates" to "Figure 11 shows"

Line 454:

Phrased awkwardly. Change "which maximally affect the ozone depletion in April" to something like "Cumulative ozone depletion shows maximum values in April"

Line 458:

Change "two times" to "by a factor of two"

Line 460:

Change "if compare" to "when compared"

Line 492:

"the total content fluctuates" This is phrased awkwardly.

Line 531:

Change "Further" to "Furthermore"

Line 532:

Change "ozonosondes" to "ozone sondes"

Line 532:

Split into two sentences: "…observations. Finally, …"

Line 537:

Delete "revealed"

Line 562:

 I don't think that "repeat" is the best choice of word here. Maybe "rivalled"

Line 586:

Change "Ny-Aalesund" to "Ny-Ålesund"

Line 607:

Don't abbreviate "QJRMS"

Lines 644-645:

Jadin et al. is an article from a predatory journal. Delete the reference.

Line 686:

It seems to me that the reference Madrid et al. is not cited in the paper. Delete.

Line 703-704:

Pedatella et al. is a news article. Delete the reference.

Line 711:

Change "Lefe`vre" to "Lefèvre"

Line 738-740:

Smyshlyaev et al., 2017 is only available in the Russian language, so I can't read it. See specific comments to line 180.

Line 760-761:

I wasn't able to find this article on the home page of the journal.

**References**

Dameris et al., 2021, doi:10.5194/acp-21-617-2021

Hersbach et al., 2020, QJRMS, 146(730), 1999– 2049, doi:10.1002/qj.3803

Khosrawi et al, 2017, doi:10.5194/acp-17-12893-2017

Lawrence et al., 2020, doi:10.1029/2020JD033271

Manney et al., 2020, doi:10.1029/2020GL089063

Solomon, 1999, Reviews of Geophysics, 37(3), 275– 316

Tegtmeier et al, 2008, doi:10.1029/2008GL034250

Wohltmann et al., 2017, doi:10.5194/acp-17-10535-2017

Wohltmann et al., 2020, doi:10.1029/2020GL089547

---

## Referee Comment (RC2)

Review of "Dynamical and chemical processes contributing to ozone loss in exceptional Arctic stratosphere winter-spring of 2020" by Smyshlyaev et al.

(Gloria L Manney, manney@nwra.com)

This paper aims to analyze and distinguish dynamical and chemical contributions to the evolution of Arctic ozone in the 2019/2020 winter.  Results for dynamical diagnostics from reanalysis data are presented along with an analysis using reanalyses temperatures to estimate chemical ozone loss from ozonesonde data; this is followed by trajectory modeling that is used to diagnose chemical ozone loss from assimilated ozone data from the ERA5 reanalysis. Finally, the authors present analysis of chemistry transport model runs with different scenarios aimed at showing the relative contributions of various chemical and dynamical processes to the evolution of ozone in the 2019/2020 winter.  The paper as it stands is not suitable for publication in ACP, for the following major reasons:

(1) Dynamical results from reanalysis data: The dynamical diagnostics shown from reanalyses and the discussion thereof are almost all things that have already been published in existing papers on the 2019/2020 winter; in addition, the authors are unclear about which reanalyses are used where and why -- in particular, the NCEP / NCAR reanalysis is deprecated for all stratospheric and polar processing studies and should not be used; and it appears that for any given diagnostic or model calculation, one reanalysis (though often it is not stated which) is used -- while comparing multiple reanalyses for each calculation is highly desirable and enhances the robustness of the results, using different individual reanalyses for different calculations does the opposite, since one cannot even evaluate the results as a whole knowing that they are based on the same representation of the atmosphere.  Further, the construction of and/or interpretation of some of the diagnostics is unclear or inconsistent.

(2) Use of ERA5 assimilated ozone for quantitative estimates of ozone loss:  Because ERA5 ozone is an assimilated products based on ingesting several datasets (including different datasets at different times), extensive validation of this product would be needed before using it to derive quantitative estimates of ozone changes, especially on the daily temporal and relatively localized (e.g., where zonal means are inappropriate) spatial scales that are important for polar stratospheric ozone loss.  While doing so is a highly valuable undertaking, I am not aware of any study that has done this already.

(3) Trajectory modeling: The initialization of the trajectory model on a single latitude circle makes all of the results highly suspect, and makes interannual comparison virtually impossible, since any latitude circle will be in different parts of the vortex at different times and especially in different years.  Without relatively uniform sampling (to guarantee which one would have to initialize parcels relatively uniformly throughout the vortex, e.g., a procedure similar to that described in Manney & Lawrence, 2016, ACP), you cannot even compare results on different dates in one year, much less do fair interannual comparisons.

(4) Chemistry-transport modeling:  There is inadequate description of the details (e.g., initialization dates and fields, boundary conditions, etc.) of the set up of the model runs.

Some of the interpretation of the results is unclear or inconsistent. It is not obvious that the model has been well-validated, nor that the agreement with observations shown here is adequate.

Because of these issues, I cannot recommend publication in ACP unless / until the authors focus the paper clearly on the results that are new and clearly acknowledge and describe existing literature where they are not; clearly describe all of the reanalyses and datasets that are used, which ones are used for what diagnostics, and justify why those data products are chosen; resolve the inadequacies in the description and formulation of the model simulations and in the interpretation of the results; and enact substantial improvements to clarify the writing and English usage. The specific comments below give further details on these issues that should provide guidance if the authors choose to revise the paper.

**Clarification issues that are needed throughout:**

You should be careful about using (as you currently do even in the abstract where being precise is especially important) terms like "ozone loss", since that is usually taken to refer to chemical "loss". Also, for the most part, dynamical factors tend to *increase* ozone in the lower stratosphere, so saying they contribute to chemical "loss" (or to ozone decreases to use a term that does not imply chemical loss) can be confusing. Finally, whether and which dynamical processes contribute to decreasing ozone depends on whether you are talking about column or vertically-resolved ozone -- for example, column ozone is lower in cold regions because of the direct impact of lower temperatures on density at a given pressure, and this can be a substantial portion of the appearance of very lower column ozone values in the coldest portions of the vortex; in many places in the paper it is not made clear which you are talking about, and in some places it is not clear how calculations of one relate to the other.

Similarly, you often use the term "ozone anomalies" when you specifically mean low ozone in the winter polar lower stratospheric vortex (or equivalently low column ozone) that is related chemical loss. The are / can be many other kinds of "ozone anomalies", including winter/spring seasons (such as 2015 in the Arctic) with anomalously high ozone, as well as other kinds of low ozone anomalies (such as "mini-holes" in column ozone, which are entirely dynamical in origin and typically appear outside the polar vortex, but often at high latitudes near the vortex in winter). If you are going to use the term anomaly, you should define exactly what it is an anomaly from; however, it appears to me the way you use it means unusually low ozone relative to climatology that arises at least partially from chemical loss -- if that is the case, I would suggest using different terminology that is more precise. (E.g., page 2, line 30, instead of "...significant ozone anomalies are observed in the Arctic less…" it would be clearer to say something like "...extensive chemical ozone depletion occurs less often in the Arctic than in…"; on line 34, for the Antarctic, it makes sense to simply say something like "the Antarctic ozone hole was one of the deepest / most extensive on record…")

**Specific Comments (in order of appearance, not importance):**

(Where I suggest references, I have tried to provide the DOIs if they are not already cited in this manuscript.)

Introduction, overall:  While I'm providing a number of comments below about particular statements and the literature cited for them in the introduction, I question whether this detailed a review of well-known impacts of stratospheric ozone loss is needed or appropriate for this paper.  For example, possible (though as yet still controversial) effects on precipitation or weather seem as best peripheral to this paper.  I believe much of the material that is not directly related to setting the context for interannual variability and interhemispheric variability in stratospheric vortex dynamical and chemical conditions and chemical ozone loss could / should be condensed or deleted.

Page 1, lines 28-29: This is one of many places where there is a very incomplete list of references, some of which are not the most appropriate ones.  In cases like this where it is a general, well-known point, adding a recent review paper (such as Domeisen et al 2019, https://doi.org/10.1029/2019JD030923 in this case) or at least simply adding "e.g.," before or "and references therein" after would convey the information that these are only examples of some of the literature on the subject.  Simply adding "e.g.," beforehand would probably be sufficient in this case.

Page 2, lines 30--31: Smyshlyaev et al (2016) is not a key reference here, I would suggest some earlier papers that were among the first to focus on disentangling chemical and dynamical effects on ozone (e.g., Manney et al, 1995, JAS, https://doi.org/10.1175/1520-0469(1995)052%3C3069:LTCUDP%3E2.0.CO;2; Manney et al 2011, Nature -- especially the SI for details on chemical and dynamical effects on column ozone -- and references in the latter).  WMO reports are always good references, in this case the 2006 one has a particular detailed section on diagnosing chemical and dynamical effects on column ozone.  This is a case where "and references therein" is definitely appropriate.

Page 2, line 33: As you note on line 36, there was also a strong SSW (arguably stronger in terms of abrupt changes than that in 2002) in the SH in 2019;  Wargan et al (2020, JGR) should be cited in both places for that; and it should be mentioned with the 2002 one (reorganizing this paragraph to talk about them together would be helpful.  Solomon et al (2014) is not an appropriate reference for the 2002 SH SSW -- the most appropriate ones would probably be Allen et al (2003, GRL, doi:10.1029/2003GL017117) and/or Hoppel et al (2003, GRL, doi:10.1029/2003GL016899) -- the first peer-reviewed papers on that SSW -- and Shepherd et al (2005, JAS -- the preface to the special issue on that SSW).

Page 2, line 35: More appropriate references for the depth of the 2015 Antarctic ozone hole would be Ivy et al. (2017, GRL, doi:10.1002/2016GL071925), Stone et al. (2017, JGR, https://doi.org/10.1002/2017JD026987), and/or the 2018 WMO report.

Page 2, line 37: Need to specify whether by "largest decrease" (should be "decreases") you mean in vertically-resolved or column ozone.

Page 2, line 41: Should cite Bernhard et al (2013, ACP, https://doi.org/10.5194/acp-13-10573-2013) for anomalously high surface UVI in 2011.

Page 2, line 45: These results of Chubarova et al (2020) are questionable, given that the three methods used in that paper to estimate UV trends resulting from changes in cloudiness and ozone agree very poorly (their Figure 13).

Page 2, line 52: Manney & Lawrence (2016, ACP, cited elsewhere in this manuscript), should be cited here.

Page 2, lines 61-62: Need references for this sentences; Lawrence et al (2020) is good for the temperatures (also several other papers in the JGR/GRL special issue on the 2019/2020 Arctic vortex, including Wohltmann et al, 2020, which you cite elsewhere); DeLand et al (2020) should be cited here (as well as where you do later on) since it discussed observed PSC activity.

Page 3, line 65: Reference should be Dameris et al (2021). Other published papers that discuss the low column ozone and diagnose its chemical origins should be cited here, including Wohltmann et al (2020), Inness et al (2020, JGR), and others from the aforementioned special issue.

Page 3, line 77: The fact that it was exceptionally long-lived, which you don't mention, was also critical (e.g., Manney et al, 2020; others). Because the results in the paragraph this sentence ends are all from published papers, I believe this should be greatly condensed with appropriate references to those papers.

Page 3, line 82: If you are going to discuss "the El Nino-South [sic] Oscillation effect", you need to define what that is. A reference to the review by Domeisen et al (2019, Rev. Geophys, https://doi.org/10.1029/2018RG000596) could be helpful. However, I am not sure that this paragraph contains any information that is necessary / directly relevant to the current manuscript, since you do not analyze any relationships to these SST patterns.

Page 3, lines 86--92: This discussion of the early 2019 major SSW is peripheral to this paper and does not add anything. Further, if it is included, the radiative / dynamical interactions leading to a slow recovery after many strong, early-season SSWs should be discussed (e.g., as in Hitchcock and Shepherd, 2013, JAS, DOI: 10.1175/JAS-D-12-0111.1).

Page 3, line 95: Add "e.g.," before Rex et al reference, since there are numerous papers on this.

Page 4, lines 101--110: This discussion could be condensed to a sentence with appropriate references, or deleted entirely. However, taking this as it is: Saying there were "regular...ozone holes in Antarctica" by "the end of the twentieth century" is misleading given that there were

annual ozone holes by the early 1980s. Even more importantly, there has not been anything that could be unequivocally called an "ozone hole" in the Arctic, even through 2020 (see, e.g., Solomon et al, 2014; Wohltmann et al, 2020; and the online discussion for Dameris et al, 2021). If you are going to talk about the impacts of the Montreal protocol, it would be best to cite some of the several "World Avoided" papers that address this topic in detail (e.g., Newman et a, 2009, ACP, https://doi.org/10.5194/acp-9-2113-2009; Chipperfield et al, 2015, Nature Comms, DOI: 10.1038/ncomms8233).

Page 4, lines 110--114: There are numerous other references so at least add an "e.g.," before Weber et al. Also, if you cite Ball et al (2018) it is also important to cite some following papers that update and / or call those results into question (e.g., Wargan et al, 2018, GRL, https://doi.org/10.1029/2018GL077406; Chipperfield et al, 2018, GRL, https://doi.org/10.1029/2018GL078071; Ball et al, 2019, ACP, https://doi.org/10.5194/acp-19-12731-2019, 2019).

Page 4, lines 115--124: Again, this paragraph could be greatly reduced or deleted. Also: on line 116, saying the 2019 ozone hole was "lowest" is very confusing, I'd suggest "shallowest" or some other such wording; line 120, neither Butler et al 2020 nor Wargan et al 2020 discuss the 2020 Antarctic ozone hole (and Butler et al 2020 is about two NH winters), so neither is an appropriate reference here. Further, the statement on lines 118--120 that the Antarctic is showing increasing interannual variability is entirely speculative and no evidence is given to back it up (two contrasting years does not make a trend).

Page 4, lines 126--128: It would be good here to make a clear statement about what is new in the paper that goes beyond the papers that have already been published.

Page 5, lines 138--139, and Section 2.1: It should be stated which reanalysis or reanalyses are used for each diagnostic shown in the paper. As per the major comments, need to justify using and/or showing different reanalyses for different diagnostics. A very strong justification is needed for using the NCEP/NCAR (aka NCEP-R1, or just NCEP as you call it) reanalysis, which has long been deprecated for any polar processing studies (e.g., Manney et al, 2005, MWR, https://doi.org/10.1175/MWR2926.1; Manney et al, 2005, JGR, doi:10.1029/2004JD005367; Lawrence et al, 2018, ACP, https://doi.org/10.5194/acp-18-13547-2018; and references therein).

Page 5, line 140: Throughout this subsection, need to say which reanalysis or reanalyses was used to calculate each of the diagnostics and why the same one (or, much better, more than one) wasn't used to calculate all of them.

Page 5, lines 141--144: This is not a useful diagnostic since it is neither related to the polar vortex nor expected to capture the actual minimum in high-latitude temperature in all conditions. While you may argue that this region was inside the polar vortex most of the time during 2019/2020, you cannot make that case for all of the winters you focus on, much less all Arctic winters. Even if this region was in the polar vortex, the location of minimum temperatures (which isn't always inside the polar vortex either since the cold region and vortex are often not

concentric in the Arctic) varies a lot both within one season and in between years, so you are almost certainly not comparing the lowest high-latitude temperature at different times in one year or in between years. In the list of years compared, 1996-1997 stands out as being the one that had only modest chemical ozone loss (with the low column ozone in spring 1997 being largely related to dynamical effects including the direct effects of the late period of low temperatures in that winter (e.g., see discussion of and references on 1996-1997 in the supplementary information of Manney et al, 2011, Nature). This distinction is important, particularly when discussing column ozone changes. Finally, the relationship of temperatures in the lower stratospheric vortex in 2019/2020 to the other years with the most ozone depletion, to climatology, and to those in the Antarctic winter, has already been more completely and correctly discussed (accounting for the full region of low temperatures), most completely in Lawrence et al (2020) and Wohltmann et al (2020), so a brief statement citing those papers (as well as DeLand et al, 2020 for the relationship of temperatures to PSC observations) would be quite sufficient and more accurate than including these diagnostics.

Page 5, lines 145: Need to give some references in relation to the effects of wave propagation as diagnosed by the Plumb or other formulations of 3D EP fluxes (e.g., Nishii et al, 2011, J Clim, DOI: 10.1175/JCLI-D-10-05021.1, and references therein.)

Page 5, lines 150-155: It is not clear (here or later) how the discussion of this diagnostic goes beyond that in Lawrence et al (2020). Also need references on the calculation of the NAM index from geopotential height anomalies (e.g., Cohen et al, 2002, Baldwin and Thompson, 2009). If the NAM index is indeed calculated as described here (which description is consistent with the papers mentioned above and with Lawrence et al, 2020) then the range of values in the figure you show later does not appear to make sense (see comment on that figure). (Again, this is a case where it is not clear that the analysis you show of this diagnostic goes beyond or adds anything to that in Lawrence et al, 2020.)

Page 5, line 159--Page 6, line 2: Need to say something about the validity of trajectories as long as 120 days for this purpose. Typically individual trajectories are not considered reliable (even in the lower stratosphere where radiative time scales are long) for more than a couple of weeks, so lengthy trajectories are used only to diagnose large scale motions by using very large ensembles of parcels. It is not at all clear that this purpose -- because you interpolate ozone to individual locations, thus assuming that those locations are relatively precise -- is consistent with that type of usage. Also, as mentioned elsewhere, because the ERA5 ozone is an assimilated product based on combining numerous datasets, one would need to either cite or perform detailed validation before its usage could be considered appropriate for this quantitative usage.

Page 6, Section 2.3: How is "inside" the vortex determined (and which reanalysis is used to do that)? How far inside must the data be for the "well isolated" assumption to be valid? There are many more complete references for effects of descent on ozone in the vortex than Braathen et al (1994; note that you have a typo in that citation), e.g., Tegtmeier et al, 2008, GRL, https://doi.org/10.1029/2008GL034250, as well as numerous other papers cited in the WMO reports (again, the 2006 report has a special focus on distinguishing chemical and dynamical

effect in the Arctic vortex). How are the descent calculations done? To be robust, they would have to follow the motion of the air sampled at the time and location of each ozonesonde measurement, since descent is by no means uniform throughout the vortex. If you are calculating a vortex-averaged descent rate that is used with vortex-averaged ozone, for that to be even roughly an accurate estimate, you would have to demonstrate that you have uniform and consistent coverage of the vortex in both the ozone profiles that go into the average and the diabatic descent (which in the latter case includes demonstrating that the descent rate is a reasonable approximation of all the descent conditions the parcels in the ozone measurements experienced). Why are temperatures from JRA (55 presumably) used with an offline radiation code instead of using diabatic heating rates provides with the reanalyses (ERA5, MERRA-2, and JRA-55 all provide these, and it would seem to make more sense to take those from whichever of these reanalyses you use to determine vortex characteristics for the sonde analysis)?

Page 6, line 184: This seems to be very coarse resolution, especially in the vertical. What is the actual vertical resolution in the lower stratosphere where you are focusing on the results? Is this adequate to capture the expected vertical variations / motion?

Page 6, lines 186--187: I am no expert on this, but I would like to see some justification of why it is appropriate / adequate to treat PSC formation as STS during a winter such as 2019/2020 when temperatures were low enough that larger solid HNO3 containing particles were present (and even at some time ice PSCs).

Page 7, lines 194--195: Why north of 64N? This does not encompass the entire vortex, nor the entire region of lowest temperature, except perhaps on some individual days when the vortex is unusually pole-centered -- thus it does not encompass the full region in which PSCs might be expected.

Page 7, lines 202--210: This has been covered more completely in already published papers including Bernhard et al (2020), Inness et al (2020), and Dameris et al (2021). Simply describing this briefly with citations of those (and potentially other) papers would be more appropriate than presenting this as if it were a new result.

Page 8, lines 226: More like the last approximately 60 years, see Lawrence et al (2020) and Matthias et al (2016, GRL, doi:10.1002/2016GL071676).

Page 9, lines 229--237: As mentioned in relation to the methods section, this has been covered more completely and precisely in Lawrence et al (2020), Wohltmann et al (2020) and others. It would be more appropriate to include a brief statement citing these papers rather than presenting this as if it contained new results.

Page 9, lines 242--243: As mentioned already, 1996-1997 did not have severe ozone loss. Moreover, not only did 1996-1997 not have a strong polar vortex (which is by no means synonymous with a cold one) but rather an exceptionally weak one until spring, but also

2004--2005 was notable for being cold and having substantial ozone loss, but having a rather weak vortex that allowed considerable mixing (e.g., Manney et al, 2006, GRL, doi:10.1029/2005GL024494; Schoeberl et al, 2006, JGR, https://doi.org/10.1029/2006JD007134; Lawrence et al, 2020).

Page 9, lines 246--247:  What are the implications of this?  And what does this add to what has already been shown by Lawrence et al (2020)?

Page 10, lines 256--261: What is new here that goes beyond what was shown by Lawrence et al (2020)?

Page 11, lines 265--286: Again, it is not obvious what this adds to what has already been shown / discussed by Lawrence et al (2020).  Also, Figure 4 shows NAM index values up to about 10, whereas Lawrence et al (2020) show values near 5 at the same time and place (their Figure 4a); the values shown by Lawrence et al are typical of those shown in previous work calculating that index based on GPH anomalies.  Yet your description of your calculation in the methods section sounds like it is the same index used in these previous papers.  Please explain this apparent inconsistency.

Page 12, lines 299--301 (Figure 4 caption):  How significant are the differences in "Plumb" fluxes in (b) from climatology, compared to those during similar length time periods in other individual years or at other times in 2019/2020?   That is, how unusual is this behavior?

Page 12, line 306 to Page 13, line 311:  The "SSW event" you describe was very minor and affected only the upper stratosphere.  Although it could be the case that it resulted directly from the enhanced upward wave propagation, you have shown nothing to demonstrate this.  You have also shown no evidence that a Rossby wave breaking event occurred in the troposphere nor that if it did, it was associated with the enhancement of wave activity.  You have not shown potential vorticity at all, so the reader cannot know if / where it was low.   The situation described by Coy et al (2009) was in relation to a major SSW that affected the entire stratosphere for weeks -- there is no reason to believe that the very brief minor event in the upper stratosphere in 2020 that you describe is analogous in any way.  (In addition, it is not clear in any of the accompanying discussion, why this minor event that showed no evidence of significantly affecting the lower stratosphere is relevant to the analysis in this manuscript.)

Page 15, line 306 (Figure 6 caption):  Why 50-70N?

Page 15, lines 331--344: Please see major comment (3), as well as previous comments on inappropriate initialization locations, need to justify the length of the trajectories for this purpose, and the need to demonstrate (or cite literature that did so) that ERA5 assimilated ozone is appropriate for this purpose.  Also, choosing a different initialization date in each year apparently just because that latitude circle happened to be within the vortex is further degrading the ability to make interannual comparisons and the dependence of the results on details of

vortex shape, position, and evolution.  Again, you do not say how you determine what is inside the vortex.

Page 17, lines 358--359:  Do you mean you ran the model with ERA-Interim, or you ran the model with ERA5 degraded to ERA-Interim-like resolution?  How large are the differences?  If they are large enough so as to make the results highly uncertain, then this points to the need to do something more (typically driving the model with several different reanalyses) to determine whether the results can be considered robust at all.  There are numerous papers (eg, in the S-RIP special issue of ACP/ESSD, https://acp.copernicus.org/articles/special_issue829.html) that show substantial differences in results of trajectory analysis and / or chemistry / transport model results from using different reanalyses to drive them.   (Just because ERA5 is the newest, does not mean you can automatically assume without testing that it is better for all types of analyses.)

Page 17, line 363: How do you determine where the vortex centre is?

Page 17, line 364: How did you select this PV value?  Is this what you use to define the vortex edge previously in the paper, and, if so, what values did you use for the other levels that are shown / discussed?  Is the same value appropriate for each of the reanalyses that you use?

Page 17, line 266: See previous comments regarding how representative an average of measurements at a small number of stations is of the entire vortex.

Page 17, line 370: "ozone losses were the lowest on record" -- you must mean "ozone values were the lowest on record" or "ozone losses were the largest on record".

Page 17, line 372:  There are quite a few other papers in addition to Manney et al (2020) that also show this, including Wohltmann et al (2020).  In addition if on line 370 you were implying that chemical ozone loss was larger in 2020 than in 2011, then it is not really consistent with those papers, since they estimate chemical ozone loss amounts to be very similar in 2020 and 2011 (but indeed peaking at lower altitudes in 2020).

Page 17, lines 377--378:  Why these two winters?  What was 2004-2005 "much colder" than (your wording could be interpreted as saying it is colder than 2019-2020, which obviously is not the case)?  Since there are several studies (e.g., Kuttipurrath et al, 2010, ACP, https://doi.org/10.5194/acp-10-9915-2010; Livesey et al., 2015, ACP, https://doi.org/10.5194/acp-15-9945-2015) that provide chemical ozone loss estimates for a wide range of years in the past decades, why not compare with all of them.  Especially, why not compare with 2011 since it was the previous year that unequivocally had the largest ozone loss?

Page 17, line 380, the terminology "ozone anomalies" is imprecise and potentially misleading, since you are not talking about just any ozone anomaly (which could occur anytime or anywhere for many different reasons), but specifically anomalously low ozone in lower stratospheric vortex

that is related to unusually cold conditions there and thus partly to chemical ozone loss. I recommend not using this wording (here or elsewhere in the paper).

Page 17, line 381: There needs to be more information given on the chemistry transport model and its set up (though this would probably be best in the Methods section). How is the model initialized (especially ozone and other important trace gas fields)? What are the boundary conditions for trace gases? What is(are) the initialization date(s)? Why is MERRA-2 used to drive this model whereas ERA5 is used to drive the trajectory model?

Page 19, 393--395: This doesn't look to me like it agrees very well with the OMI data shown earlier. The minimum values in Figure 9 appear much higher than those in Figure 1 on the first three days shown; there appear to be significant differences in morphology, especially, the OMI data on the second day shown suggest that the lowest column values are actually within the polar night, whereas your model results show them to be well away from there, with no low values immediately surrounding the region OMI cannot observe.

Page 19, lines 398--399: There are, in fact, satellite measurements in 2019/2020 that observe in darkness (e.g., MLS), though you would have to compare profiles rather than column (but profile comparisons are a necessary part of validating model results. But in any case, per my immediately previous comment, since the model and OMI observations appear to agree very poorly (in morphology as well as in values) going into the polar night, I fail to see how you can argue that your model results provide useful information in polar night.

Page 19, lines 399--404: This does not make sense to me. First, the OMI data show minimum values abutting the gap in polar night -- meaning the actual minima are inside the polar night. Second, the results of the chemical reactions are transported throughout the vortex, so there is no reason to expect ozone to be lowest in the sunlit regions. Third, in order to interpret these and understand how direct dynamical effects of the low temperatures may be involved we need to know where the vortex is and where the cold region (which is not necessarily concentric with the vortex) is.

Page 19, lines 408--412: How well do these results agree with the PSC observations described in DeLand et al (2020)? And / or with other PSC observations?

Page 19, lines 413--416: As per two comments above, the region of PSCs and that of chemical ozone loss are not expected to coincide since chemically processing air is rapidly transported throughout the vortex and does not remain only in the region with PSCs.

Page 20, Figure 10: It would be helpful to know where the polar vortex is, and where the cold region is to interpret this figure.

Page 22, lines 427--428: How this is calculated should be explained further (probably in the Methods section), and something should be said about how the processes included compare with those in well-validated models, as well as how the chemistry in this model was validated.

Page 22, lines 428--429:  Why would the maximum rate of ozone destruction be at the edge of polar night?  I would think it would be wherever chlorine is activated (which, when fully activated, is the whole vortex) and there is the most time in sunlight.  Since activated chlorine is expected to be quickly transported throughout the vortex, and since the region just outside the edge of polar night receives only a little sunlight compared to regions farther into daytime, I would think the edge of polar night would be rather low in ozone destruction?

Page 22, lines 429--434: Since all of these processes are going on inside the polar vortex, the main effect of polar vortex position is how much of it experiences sunlight, and that is only a significant factor until early March.  You have not mentioned here descent (which increases ozone in the vortex and is one of the most important dynamical processes changing ozone), nor have you mentioned the direct dynamical effects by which low temperatures are associated with low column ozone.  In addition, none of your statements here have been demonstrated since you never show where the vortex is or where the cold region is.

Page 22, lines 441--449:  These lines appear to be an exact repetition of the statement in the Methods section.  Delete.

Page 23, line 463:  It is my understanding that at the altitudes where ozone contributes most to the column (below ~20--25 km), gas-phase chemistry is very slow (much slower than dynamical time-scales), so I don't understand how it plays such a large role?

Figures 12--15:  Why do you not show OMI / model differences for 2019/2020?  Why do you suddenly bring in SBUV data to show a climatology rather that deriving that from OMI data for 2005 through 2019?  How do OMI and SBUV data compare, are there significant differences?  Comparing to a climatology rather than directly to observations in the same year seems an indirect and potentially inaccurate way of assessing what processes are needed to reproduce observed fields.  Also, it would help with interpretation of the results if you could show a timeseries indicating where each station is with respect to the polar vortex during the period shown.

Page 27, first two paragraphs of conclusions, and line 555:  In the paper, you have not made the case that the very minor SSW event (which affected only the upper stratosphere significantly, hence the absence of any mention of it in the many papers already published on the 2020 lower stratosphere conditions and ozone loss) significantly affected the lower stratosphere.  In fact, temperatures in mid-March 2020 were already rising, but did so at a much slower rate than in the vast majority of years, many of which also had only very minor SSWs during this period. While nothing rules out a small effect of this minor SSW on lower stratospheric temperatures, you haven't demonstrated that there was one either -- given the many variables that affect temperatures and the seasonal cycle, the timing of the temperature increases could have been coincidence.  Thus, it is not justified that this event should feature so prominently in your conclusions.

Page 28, lines 558--560: Per previous comments, this has not been demonstrated because of the inappropriate choices for initialization of the trajectories.

Page 28, lines 561--563: This has been shown for 2020 in previous papers, with which you should compare your results.

Page 28, lines 564--569: Per specific comments, this has not been demonstrated.

**Typos / Minor Points:**

For the most part, I am not including details here of improvements that would be needed in the English usage, as the revisions needed to the scientific content are sufficiently major that much of the structure of the writing will be changed. The following are thus just a few things that happened to catch my eye:

Abstract, line 15, "year" should be "years"

Page 2, line 32, "warm" should be "warmer".

Page 2, line 49, "statically" should be "statistically"

Page 3, line 79, delete comma

Page 3, line 86, "the main SSW" is not appropriate wording, particularly since SSWs are quite common in the Arctic. Perhaps you mean "a major SSW".

Page 3, lines 91-96, this sentence is very long and convoluted and nearly impossible to parse correctly.

Page 6, line 164, "has" should be "have".

Page 9, line 245 "strongest weakening" is extremely confusing, please reword.

Page 15, line 340, saying ozone losses were "higher" could be confusing (if the reader thinks of higher ozone values), I'd suggest a wording more like, e.g., "...more ozone loss occurred…"

---

## Referee Comment (RC3)

Review of the manuscript "**Dynamical and chemical processes contributing to ozone loss in exceptional Arctic stratosphere winter-spring of 2020**" by S.P. Smyshlyaev et al. 2021

This manuscript discusses dynamical and ozone-related chemical aspects of the extreme Arctic stratospheric winter/spring of 2019/2020. It is divided into several loosely connected parts: a discussion of dynamics, calculations of chemical ozone loss by two methods, and an analysis of ozone chemistry using several CTM experiments driven by reanalysis meteorology. The latter includes another calculation of ozone loss. Most of the results broadly confirm what we already know about the 2019/2020 polar winter/spring from other studies. One new and very surprising finding is that a very large contribution to the observed ozone loss came from gas-phase chemistry. The exceptional conditions that occurred in the Arctic stratosphere in 2019/2020 deserve thorough investigation. The several very good studies that have already been published on this subject make it increasingly difficult to say something new. Unfortunately, the present manuscript suffers from several serious shortcomings. The CTM results of Section 4 look extremely problematic, making the assertion about the role of gas-phase chemistry doubtful. Most of the discussion of the vortex dynamics lacks novelty. The ozone loss calculations using a trajectory models are not sufficiently discussed at best and are possibly not representative of the true vortex ozone depletion. The methods used in the paper are not described in sufficient detail. Sometimes it is not even clear what data were used to make a given plot. The different sections of the manuscript are quite disconnected; there doesn't seem to be any overall logic in the presentation. I delineate the main issues that I have with this study in my general comments below, and the details in my specific comments. I also try to suggest some revisions where possible. However, I'm afraid that the amount of work necessary to make this manuscript suitable for ACP greatly exceeds what's normally considered "major revisions". Sadly, I cannot recommend this study for publication.

**General comments**

1. In my opinion, the **CTM results (Section 4)** are extremely problematic. I provide some more detailed discussion in my specific comments below. Here, I will just highlight the main points. The total ozone maps from the CTM shown in Figure 10 are in stark disagreement with the OMI total ozone maps from Figure 1. That alone puts the utility of the CTM experiments into question. Furthermore, the morphology of the ozone loss frequency maps (Figure 11) bears only a vague resemblance to the actual geometry of the polar vortex. The latter is not shown in the paper but it's easy to plot using the same reanalysis that the paper uses, MERRA-2, as I show below. It appears that areas of high loss frequencies from the CTM often fall outside of the vortex boundaries. This doesn't seem right. The way polar ozone loss works is that the most significant depletion occurs within the chemically processed airmass rich in active chlorine, i.e., within the polar vortex, not outside of it. In addition, or perhaps related to the above, the CTM experiments in Section 4 suggest that much, even most of the ozone loss occurred via gas-phase reactions involving NOx. This goes against our established understanding of polar ozone chemistry. That doesn't automatically make it wrong and, yes, if true it would be a major finding – but then it would require a lot stronger evidence than a low-resolution CTM experiment

that fails to reproduce the observed evolution of total ozone distributions! As it is, this result only indicates likely problems with the CTM. See my specific comments for details.

2. **Section 3.1.** Almost everything in this section has already been discussed in detail elsewhere: Evolution of total ozone in Dameris et al., 2020; minimum temperatures as compared to other cold winters and climatology in Innes et al., 2020; heat fluxes, wave activity and geopotential height and related metrics as well as surface impacts in Lawrence et al., 2020. You are clearly aware of that as you cite those other studies in the paper. In principle, it's OK to have a study that confirms previously published results, especially if it uses different methods or data sets. However, I'm not convinced that this is the case here as both papers use reanalysis data and similar diagnostics. If there are any novel or otherwise valuable aspects here, please clearly state what they are and explain how they are distinct from the findings of the existing papers. If not, then I think most of this section should be eliminated. One element that may not have been discussed before (unless I missed something) is the analysis of 3-D Plumb fluxes. Currently, the discussion of Plumb fluxes is somewhat limited in the manuscript. Perhaps one way to salvage this section would be to expand this part while significantly shortening much of the preceding material.

3. **The different parts of the manuscript are quite disconnected from each other**. For example, chemical ozone loss is calculated using three different methods (trajectory analysis, ozonesondes, and CTM) but no attempt is made to cross-check and reconcile the results.

4. **The description of methods and data sets used in this study is insufficient**. It's not always clear which reanalysis is used for what. It's not even clear if the NCEP reanalysis mentioned in line 138 is used at all as it's not talked about anywhere else in the manuscript. I couldn't find any information about which reanalysis was used to generate Figures 2-6. Section 2.2 uses ERA5 to initialize the ozone content of air parcels used in the trajectory analysis, but no evaluation of ERA5 ozone is provided or cited (although I don't think that ERA5 ozone has been thoroughly validated yet). The descriptions of the trajectory model and the CTM also lack detail. For example, how is vertical advection done in the CTM? Why should we think that the very course resolution of the CTM ($5^{\circ} \times 4^{\circ}$) is adequate? It is not clear why different reanalyses are chosen to drive the trajectory model (JRA) and the CTM (MERRA-2). This is not necessarily wrong, but it does require some explanation that is not provided. No justification is given for the choices made in the trajectory calculations, e.g., why the parcels were initialized at those specific locations and times. This list is not exhaustive. See my specific comments for details.

**Specific comments**

**L106**. Note that there's a considerable debate over whether the Arctic depletion events should be called "ozone holes". If you do use that term, please drop the *a*; just "*appearance of large ozone holes*"

**L138**. What was the NCEP reanalysis used for? I couldn't find any mention of it in the rest of the paper.

**Section 2.1**. Are these diagnostics calculated from all three reanalyses mentioned above or just one? Which one?

**L138**. The canonical reference for ERA5 is Hersbach et al (2020).

**L142**. Please explain why this (somewhat narrow) latitude range was chosen.

**L144**. Are the climate averages from reanalyses? Which one, specifically?

**Section 2.2**. It's hard for me to understand from this brief description how ozone loss is calculated. How were the initial parcel locations selected? Were they initialized with ERA5 ozone and then retained their ozone content (a variation of the passive tracer method)? What does "ensemble averaged" mean in this case? Please expand this section significantly. Some of this is explained in Section 3.2, but I think it belongs here. Please, also include or cite validation results of ERA5 ozone during Arctic winter/spring. It's a relatively new reanalysis and it cannot be assumed that its ozone is suitable for science, at least not without some solid evaluation.

**L203**. Please provide a citation for OMI, e.g., Levelt et al. (2018)

**LL209-210**. This is incorrect: while ClOOCl photolysis requires sunlight, air parcels depleted in ozone can get advected out of the illuminated area. Even Figure 1 suggests that this is the case: the ozone minima occur near the terminator, as you say in the next paragraph.

**L226 and below**. What data are used here? One of the three reanalyses mentioned above? Which one?

**LL226-232**. Lower-stratospheric minimum temperatures in different extreme winters were also compared by Innes et al., 2020. What does the present analysis bring to the table that is new?

**LL230-247**. This analysis and most of Fig. 3 repeats the results of Lawrence et al. (2020). For example, see Figs 7 and 9 therein. If there is anything in the present analysis that isn't already in that study (I may have missed something), please indicate clearly what it is. Otherwise, I suggest eliminating this text and Fig 3a and b.

**LL288-290**. I don't think you really mean *assume*. Something like this shouldn't be simply assumed, it needs to be demonstrated.

**Fig. 4 b and c**. It would be better to use the same contour colors in both panels. I also suggest showing a difference plot.

**LL315-317**. I think you mean Fig. 6a, not c and March 10-13 not 11-13.

**L321**. The figure doesn't have panels e or f. I think it should be c and d.

**Section 3.2. The trajectory analysis**. I don't find these results very convincing. Why were these isentropic surfaces and these particular dates chosen? This choice of initial points is very restrictive. To make sure that this calculation represents the average vortex ozone loss, you would have to demonstrate that the trajectories sample the entire lower portion of the vortex (e.g. theta < 550 K) uniformly throughout the chemically processed vortex air and uniformly in time. Without

that it's hard to say what Figure 7 shows other than chemical loss along some arbitrarily chosen trajectories, that could be different if the trajectories were initialized differently. I would imagine that a good strategy might be to select initial points randomly within the vortex and at different times, but I can be convinced otherwise if you can show that your method does provide sufficiently uniform sampling and that the choice to start all the trajectories at the same time doesn't lead to most of them missing layers and times with particularly strong or particularly weak ozone depletion. I'm thinking of a situation when all the parcels initialized at 475 K in December were below 400 K by the end of February before serious depletion (maximized above 400 K, see Manney et al. 2020) even started. Another point: by selecting the initial parcel locations in a more robust way you could estimate ozone loss as a function of altitude / potential temperature and compare the result to that obtained from the ozonesondes analysis, thus providing some cross validation.

**L338**. This is area-weighted average, correct? Please state that clearly if true. Also, I think the few trajectories that did venture out of the vortex should be excluded from the average.

**LL 341-342**. The fact that these rather large oscillations are there indicates that a good number of trajectories left the offspring vortices after the final SSW. It's hard to imagine what chemical mechanism would produce ~0.8 ppmv up and down changes over the course of a few days in the lower stratosphere!

**Figure 7**. Please replace the commas in the y-axis tick labels with decimal points.

**LL364-365**. (1) This logic seems backward: One should select those observations that are inside the vortex, not handcraft the definition so that it accommodates the sonde locations. (2) how is the vortex edge defined at levels other than 475 K? Polar vortexes exhibit complex 3-D geometries with edges at different levels often not lining up. Some of those stations were definitely outside of the polar vortex for some period of time (e.g., Ny-Ålesund and Sodankylä in mid-March). Were these measurements excluded from the regression analysis?

**L376**. The maximum loss of 3 ppmv shown in Fig 8 occurs at 450 K. In Wohltmann et al. (2020) the vortex average (mentally subtracting the lines in their Fig. 4b) is about 2 ppmv at that level. Is it possible that this discrepancy results from different definitions of the vortex edge?

**L383**. If I understand correctly the CTM used water vapor from MERRA-2? If that is true, then it could seriously skew the results. Note that MERRA-2 **stratospheric** humidity is not very much informed by observations. It is, instead, relaxed to a zonally symmetric climatology (3-day relaxation time). It is therefore, especially suspect in extreme situations such as the 2020 winter/spring. Stratospheric water vapor in reanalyses is generally not recommended for scientific use with some exceptions (see Davis et al., 2017).

**LL392-404 and Fig. 9**. I'm sorry but I don't think the model compares well to observations at all. Below I juxtaposed the OMI total ozone map from Fig 1 (left) and the CTM ozone from Fig 9 (right) for March 15. This is a particularly striking example, but things don't look much better on the other days considered, except perhaps on 3 March. One would expect at least the dynamical features to line up as the CTM is driven by reanalysis winds – but they don't (compare the shape

of the total ozone contours and their gradients). Then looking at the ozone values the two figures have almost nothing in common: OMI shows a large complex patch of deeply low values extending between the Hudson Bay and northern Siberia while the CTM has a single weak minimum over the coast of Alaska, where OMI does not show anything noteworthy. Much weaker gradients in the CTM suggest that the model may produce far too much mixing across the vortex edge. The overall positive bias at high latitudes suggests insufficient depletion or too much resupply through descent, or, again, too much horizontal mixing. Overall, based on this comparison against OMI, I see no reason to trust the results from the CTM in this case.

[Figure]

**Figure 10**. What is shown there? The text talks about "PSC surface area". I take it to mean PSC surface area density, but I'm confused about the units, $mkm^2/cm^3$. This would be a dimensionless quantity. Maybe I misunderstood something.

**L425**. What range of isentropic levels or altitudes?

**L428**. "*nitrogen and hydrogen gases*". I think you mean nitrogen oxides and OH.

**L429-432**. This seems to suggest that the location of the vortex is an additional constraint on top of chemical depletion. I don't think this is correct. Rapid depletion is tightly confined to the interior of the polar vortex because that's where all the chemically processed air is. In fact, it is essential that the vortex air mass be isolated from the mid latitudes. Numerous studies demonstrated that chemical composition, particularly the ClOx family in the lower stratosphere exhibits a sharp discontinuity coincident with the vortex edge. Therefore, the position, extent and shape of the polar vortex are already imprinted in the spatial distribution of ozone loss rates. As a side note, that is not to say that dynamical factors don't play a role (resupply through descent, mixing).

This brings me to **Fig. 11**. Since these are snapshots not time-averages, I would expect to see sharp gradients in the loss coefficient maps that would align with the edge of the polar vortex. Instead, the fields vary gradually. Below I plotted 12Z maps of the vortex edge defined as in this paper (black) and via scaled PV contours (see e.g. Manney et al., 2020) at several isentropic levels. I used MERRA-2 so this should be consistent with the CTM. On the right-hand side, I copy/pasted the depletion coefficient fields from Fig. 11. On both days the CTM produces significant depletion outside of the vortex (however defined). For example, on 15 March the CTM shows elevated depletion over Alaska and over Eurasia where it extends almost as far south as Lake Baikal, in both cases far outside of the polar vortex. On 1 April, again, the depletion coefficient map does not bear much resemblance to the vortex. Also, note the similarity between the shape of the vortex on 15 March and the OMI ozone map from the same day.

**15 March 2020**

[Figure]

[Figure]

I see two possibilities: either I grossly misunderstood what is plotted in Fig. 11 or the CTM's chemistry doesn't represent ozone depletion correctly. It may be instructive to look at ClO or ClOx maps in the CTM run. Active chlorine should be confined to the vortex with very limited mixing across the edge. You could compare ClO from the CTM with MLS. Another possible test would be to use the CTM and noCHEMall runs to calculate chemical ozone loss and compare that with the results of section 3.2. In fact, this should be done in order to establish self-consistency of the paper and get a sense of uncertainties.

**L432**. I don't understand this sentence. What does "*destruction of the base*" mean?

**Figures 12-15** and the accompanying discussion add more doubts about the correctness of the CTM chemistry. Comparing the blue and green lines, it looks like heterogeneous chemistry plays a relatively minor role compared to gas-phase chemistry. At Pechora you estimate it to be responsible for only 25 DU of the total 70 DU of chemical loss! This raises a lot of red flags. According to our well-established understanding of polar ozone, heterogeneous chemistry is responsible for most of ozone destruction, particularly during cold winters with no major SSWs, although NOx can be important during weak-vortex winters (Sagi et al., 2017), which is not the case here. If the results presented here are correct, they require much more rigorous analysis and justification than that presented.

**LL521-526**. Again, at best this is a very surprising result that needs to be substantiated and supported by additional comparisons with observations.

**L536**. *Would have **been** observed* or *would have **occurred***. But is this really demonstrated in the paper? If indeed NOx chemistry played the main role (which I find doubtful) then a disruption of the vortex could cause more depletion by bringing more NOx lower down leading to more loss. I

think that this sentence is actually correct (while not substantiated), but it appears at odds with the results of this study (the predominant role of NOx).

**LL548-L553**. This first conclusion repeats one of the results of Lawrence et al., 2020 almost verbatim. There is really nothing new here

**Technical corrections**

There are a large number of grammatical and style issues. Below I list a few. I feel that it's not necessary at this point to mention all of them as the paper is likely to change very substantially if it gets resubmitted in the future.

**L31** *Sudden stratospheric warming* → *sudden stratospheric warming***s**

**L33**. Here and in several other places I think *main SSW* is supposed to be *major SSW*

**L49**. *statically* → *statistically*

**L66**. I suggest replacing *certain* with *selected*.

**L75**. *typical to Antarctica* → *typical for Antarctic conditions*

**L87-89**. Please, revise this sentence. It doesn't read well.

**L99**. *supposed*. I think you mean *suggested*.

**L118**. *have been formed*. I think *have occurred* would sound better.

**LL120-124**. This is a very long sentence. Please consider breaking it up into two.

**L129.** *methodology of applied diagnoses* does not read well. How about something like *diagnostic methods*?

**L136**. This sentence would read better if the word *features* were dropped

**LL135-139**. Please, expand all the acronyms that were not previously defined (even if they are already expanded in the abstract). Also, please provide references for the models here, where they are first introduced.

**L153**. *Following to (Runde et al., 2016)* → *Following Runde et al. (2016)*

**L172**. *radiation transfer model* → ***radiative*** *transfer model*

**L203**. *on the board of* → *onboard*.

**L204**. *at the early March* → *in early March*

**L315** *by Plumb fluxes* → *using Plumb fluxes*

**L316**. *dominated* I think you mean *dominant*

**LL456-458**. I think something like *by the end of March (...) total ozone content at Pechora drops by almost 50 percent* would be more clear and read better.

**L528**. Please rephrase this sentence. It doesn't read well.

**L532**. I suggest changing *values* to *magnitude*.

**Additional references cited in this review**

Davis, S. M., Hegglin, M. I., Fujiwara, M., Dragani, R., Harada, Y., Kobayashi, C. et al. (2017). Assessment of upper tropospheric and stratospheric water vapor and ozone in reanalyses as part of S-RIP,. *Atmos. Chem. Phys.*, 17, 12743-12778, https://doi.org/10.5194/acp-17-12743-2017

Hersbach, H, Bell, B, Berrisford, P, et al. The ERA5 global reanalysis. *Q J R Meteorol Soc*. 2020; 1– 51. https://doi.org/10.1002/qj.3803

Levelt, P. F., Joiner, J., Tamminen, J., Veefkind, J. P., Bhartia, P. K., Stein Zweers, D. C., Duncan, B. N., Streets, D. G., Eskes, H., van der A, R., McLinden, C., Fioletov, V., Carn, S., de Laat, J., DeLand, M., Marchenko, S., McPeters, R., Ziemke, J., Fu, D., Liu, X., Pickering, K., Apituley, A., González Abad, G., Arola, A., Boersma, F., Chan Miller, C., Chance, K., de Graaf, M., Hakkarainen, J., Hassinen, S., Ialongo, I., Kleipool, Q., Krotkov, N., Li, C., Lamsal, L., Newman, P., Nowlan, C., Suleiman, R., Tilstra, L. G., Torres, O., Wang, H., and Wargan, K.: The Ozone Monitoring Instrument: overview of 14 years in space, Atmos. Chem. Phys., 18, 5699–5745, https://doi.org/10.5194/acp-18-5699-2018, 2018.

Sagi, K., Pérot, K., Murtagh, D., and Orsolini, Y.: Two mechanisms of stratospheric ozone loss in the Northern Hemisphere, studied using data assimilation of Odin/SMR atmospheric observations, Atmos. Chem. Phys., 17, 1791–1803, https://doi.org/10.5194/acp-17-1791-2017, 2017.

---

## Author Comment (AC1)

Dear Ingo,

Thank you for your comments on the paper and constructive recommendations. We have tried to follow your suggestions and have taken into account most of them. Following we mention how the manuscript has been changed according to your comments.

**Major Comments:**

1. **Result that the contribution of gas phase nitrogen cycles to ozone loss in the polar vortex in spring is significant cannot be correct**

Reply:

We are familiar with the theory that chemical ozone loss in spring in the polar vortices is dominated by loss from catalytic chlorine and bromine cycles and caused by heterogeneous chemistry. Moreover, our previous published works followed this theory and confirmed it (Smyshlyaev et al, 1998; DeZafra and Smyshlyaev, 2001; Sovde et al, 2008; Smyshlyaev et al., 2010; Smyshlyaev et al., 2016; Timofeyev et al., 2018).

In this work, we used the same model of the formation and evolution of polar stratospheric clouds and heterogeneous processes on their surface as in previous works. Its peculiarity is the consideration of type 1 PSCs as a supercooled ternary solution of $H_2SO_4/HNO_3/H_2O$ following Carslaw et al., 1995, and NAT $HNO_3/H_2O$, type 2 PSCs as frozen drops of type 1 PSCs, taking into account the difference in freezing and melting temperatures of PSCs drops, the temperature history of air masses, gravitational sedimentation of type 1 and 2 PSCs particles (details in Sovde et al., 2008 and Smyshlyaev et al., 2010). Nevertheless, in our opinion, the 2019-2020 ozone anomalies were unusual, including later than usual periods of low ozone, when heterogeneous chemistry no longer plays a dominant role. In addition, an analysis of the altitudinal features of changes in Arctic ozone in winter-spring 2019-2020 (given in the updated version of the article and in the Supplement to it) showed that heterogeneous chemistry still plays a predominant role in the lower stratosphere (up to 25 km), and in the middle stratosphere (25-40 km) there is also a significant decrease in the ozone content, caused by the intensification of nitrogen cycles after the return of the sun to the polar stratosphere.

In the scenario without chemistry, we completely turned off chemistry for all gases, and the initial conditions were the same for all scenarios, so it is unlikely that the initial conditions could affect a significant difference in ozone for the chemistry and no chemistry scenarios. In the updated version of the article, we extended the chemistry shutdown period in the polar regions in the scenario to mid-May to compare the effect of gas-phase chemistry in scenarios without chemistry and with gas-phase chemistry until the end of April, when episodes of low ozone were observed at stations in the eastern hemisphere.

2. **Not sampling the polar vortex homogeneously when determining ozone loss**

Reply:
Trajectories were recalculated in backward and forward directions (reply for lines 332-333 below, text of paper was changed)

3. **Low novelty value and low scientific significance, redundancy, many repetitions of results from other studies, and the paper needs to be shortened considerably**

Reply:  - Abstract was updated.
 - The introduction was shortened from three to two pages.
- The section 3.1 was shortened on 1 page. Two figures were removed to Supplement.
- The section 4 was shortened was shortened as well

**4. Language. This manuscript would benefit from the help of a native speaker. You have to improve on the use of the English language.**

Reply:
We plan to use the English language proofreading service for the final version of the article.

**5. Numerous small errors, omissions and inconsistencies**

Reply:
- fig 2b, reference Madrid et al. were removed, 2nd section number 2.3 was changed on 2.4
- author contributions was added
- time intervals in the titles of Figure 6 (Plumb fluxes) were corrected: 10-13, 13-16 March were substituted on 11-13 and 14-16 March as it indicated in the caption.

**6. Make sure that the title, abstract and conclusions reflect what you say in the main text:**

Reply:
Abstract and conclusions were updated.

**Specific Comments:**

**Answers to specific comments**

| | | |
|---|---|---|
| 1. | Line 9-10:
State in the abstract which CTM you are using (i.e. the RSHU CTM). This will be of interest for many readers. | Done |
| 2. | Lines 21-22:
The phrasing is a little bit misleading, since people often think immediately of "chemical ozone loss" when they read "ozone loss". Suggestion: "…indicated that both dynamical and chemical processes make contributions to ozone changes inside the polar vortex" | Done |
| 3. | Lines 22-23: *"In this case, dynamical processes predominate in the western hemisphere, while in the eastern hemisphere chemical processes make an almost equal contribution with dynamical factors"*
I don't think that this is correct (at least in the way that it is phrased here, it is misleading). The simple reason for this is that the polar vortex is moving. Air masses that are in the western henmisphere at a particular date will be somewhere else a few days later. You have to make sure that you phrase that carefully. Ideally, one would follow the air masses inside the vortex and make statements for the vortex as a whole. That being said, it is of course a valid approach to look at specific locations, but then, for a single location, fast dynamical changes will often dominate. This is probably a question of what you use as a reference frame when you define the chemical and dynamical change. | Comparison of dynamical and chemical factors for different hemispheres is removed from the abstract. Instead, it is noted that, based on comparison with long-term average data, it can be concluded that more than half of the observed unusually large decrease in ozone content is due to the specific dynamic conditions of the winter of 2019-2020. |
| 4. | Lines 24-25: *"the chemical depletion of ozone is determined not only by heterogeneous processes on the surface of the polar stratospheric clouds, but by the gas-phase destruction in nitrogen catalytic cycles as well."* | The final sentence of the abstract has been changed. It is indicated that a comparison of the rates of ozone |

| | | destruction at different heights showed that below 25 km, almost all ozone destruction occurs as a result of chlorine and bromine activation in the PSC, but above 25 km, significant ozone destruction in nitrogen catalytic cycles is also noted. |
|---|---|---|
| | This would only be correct as a very general statement. But from the context is clear that you refer to ozone loss in spring in the polar vortex here and that you think that the contribution from $NO_x$ cycles is significant. The sentence is misleading and not correct in the end. You have to be careful about the message that you convey here. There is general agreement that chemical depletion in spring in the polar vortices is dominated by heterogeneous processes, see major comment 1. Delete this sentence. | |
| 5. | Line 28:
Would be nice if you would not only cite references for the tropospheric influence, but also for the statement *"the circulation of the Arctic stratosphere in the winter-spring season (hereinafter winter season) is characterized by strong interannual and seasonal variability"*. Suggestions: e.g. Tegtmeier et al., 2008, Solomon, 1999. | **Done**

We add

Tegtmeier et al., 2008

Solomon, 1999 |
| 6. | Line 30:
Pedatella et al. is a news article and not a peer-reviewed paper. Delete the reference. | Done

Pedatella et al. removed

Zülicke et al., 2018 added |
| 7. | Line 38-39: *"the largest decrease in the Arctic ozone was observed…"*
This statement needs some references. For the 2019/2020 winter, e.g. Manney et al., 2020, Wohltmann et al., 2020. For 2015/2016, e.g., Khosrawi et al., 2017. | Done

References added |
| 8. | Line 62: *"record low temperatures were observed in the Arctic lower stratosphere, and, as a result, a record volume of Polar Stratospheric Clouds (PSCs) was expected"*
Again, this statement needs some references, e.g. Lawrence et al., 2020 or Wohltmann et al., 2020. And you surely not mean "was expected" but "was observed". | Done

In late February - early March 2020, record low temperatures in the Arctic lower stratosphere resulted in a record volume of Polar Stratospheric Clouds (PSCs) (Lawrence et al., 2020; Wohltmann et al., 2020). |
| 9. | Lines 115-116: *"Signs of recovery in ozone levels began to be noted in the polar regions, in particular, a decrease in the depth of the ozone hole and its size in Antarctica. The 2019 ozone hole in Antarctica was one of the lowest in decades"*.
These statements need some references. | Done

+ Milevsky et al. 2020

& Safieddine et al., 2020

However we remove the chapter on Antarctic ozone anomaly |
| 10. | Line 138:
The canonical reference for the ERA5 dataset is Hersbach et al., 2020. Your reference is outdated and not peer-reviewed. | Done |
| 11. | Line 136-139:
It makes the impression to me that some of the meteorological analyses that you mention here are never used in the paper. Please do only mention the analyses that are actually used. In addition, please make sure that you give the analysis that you use to calculate quantities or that you discuss in the text at the | Done

ERA5 - trajectory analysis (line 163)

 NCEP - Plumb fluxes, tem-re analysis |

| | | |
|---|---|---|
| | appropriate places in the paper. This information is missing in several places. | (line 147)

 *JRA* - Ozonesonde analysis (line 175)

 MERRA2 - Chemistry-transport modeling (line 185) |
| 12. | Line 141-142:
 State the analysis you used for the temperatures. | Done

 We add: "..... using NCEP Reanalysis data" |
| 13. | Line 159:
 Why forward and not backward trajectories? Backward trajectories would probably have the advantage that you would lose less trajectories that leave the vortex. In addition, forward trajectories the disadvantage that at the end of the run, trajectories started at the same potential temperature level will be distributed over a range of potential temperatures. Since you would like to know the ozone loss at the end of the run at a certain level, this is unfortunate. | Changed in section 2.2

 We recalculated both forward and backward trajectories Results are presented in the supplement. The ozone behavior in both directions is almost the same for all winters. In both cases we estimate the ozone loss in the layer rather than at fixed level. In the paper we present the results of backward simulations. |
| 14. | Section 2.2:
 Important information is missing or only given later in section 3.2. What are the initial locations of the trajectories? What is the start date? These two questions are answered in section 3.2, but you should either move this from section 3.2 to section 2.2. Or vice versa (you could delete this section then)
 Did you use vertical winds or heating rates to calculate the vertical motion? What happens with trajectories that leave the vortex? What is your criterion for the vortex edge (some PV value)? These questions are not answered here or in 3.2.
 It would also be important to have some more information about the ERA5 ozone product, since your results crucially depend on it. E.g. which measurement data are assimilated, how well does the ERA5 ozone product compare to observations? | Changed in section 2.2

 The description of trajectory calculations is presented in section 2.2.

 The vortex interior was determined as the region with PV>14 PVU at 400 K, PV>26 PVU at 435 K, PV>36 PVU at 460 K and PV>46 PVU at 500 K, this criterion was used to filter out the initial locations outside the vortex and the trajectories leaving the vortex later on.

 (Hersbach, H, et al., The ERA5 global reanalysis. Q J R Meteorol Soc., 146, 1999– 2049. ttps://doi.org/10.1002/qj.3803, 2020.) discussed some improvements in ERA5 ozone data compared to ERA-Interim. The heterogeneous ozone chemistry was updated, and a number of changes were introduced in the assimilation system. (Added to the text of paper)

 Our comparisons of results of trajectory analysis and ozone losses based on ozonesondes also can be |

| | | considered as ERA5 ozone validation. |
|---|---|---|
| 15. | Section 2.3:
Important information is missing, that is only given later in section 3.2. Please state the individual stations.
What is the time period you are looking at and the starting date you use as the reference? What is your criterion for the vortex edge?
Again, you could move this information from 3.2 to 2.3. Or vice versa (and delete 2.3) | Done
We moved this information from section 3.2 to 2.3. and indicated the time period of our analysis.
It reads now:
"About 60 ozonesonde profiles measured at different parts of the vortex from 3 January to 26 March 2020 were included in the analysis, most of them were obtained near the vortex centre with high values of potential vorticity (PV)." |
| 16. | Line 172:
State the name of the radiative transfer model. Just "the radiative transfer model" is no sufficient information. | **Done**

We changed it. It reads now:

"For diabatic cooling/heating rates calculations we used the solar radiative transfer model CLIRAD developed by Chou and Suares [1999] at the NASA/ Goddard Space Flight Center and daily temperature profiles from JRA reanalysis [Kobayashi et.al., 2015]." |
| 17. | Line 170-172:
It would be appropriate to go a little bit more into detail here (even though this is an established method). So, you probably first interpolated all the ozone sonde measurements on an isentropic surface. When you fit the linear regression line to the time series, you probably don't use all of the ozone data but a time window around some given date to be able to obtain a value for the rate of change for a specific date, I suppose? What is the length of the time window? | Done

As it was recommended, we added the following paragraph to method description:

"All ozone profiles were interpolated on isentropic levels from 350 to 700 K."

And hereafter:

"The whole ozone time series was splitted into four unequal intervals according to different observed ozone decrease rate, with the slowest decrease from 3 January till 5 February and the fastest decrease from 1 March till 10 March. The starting and ending dates of each time interval were chosen to provide the least data biases at the breakpoints. |
| 18. | Line 175:
I wasn't able to find the reference Tsvetkova et al. (2004) at the home page of the journal. Please replace the reference by | Done

 Instead of two references |

| | | |
|---|---|---|
| | something easier to access. I think there are many articles describing the method. | Tsvetkova, et al., 2002 & Tsvetkova, et al., 2004

the following one was included:

Tsvetkova, N.; Yushkov, V.; Lukyanov, A.; Dorokhov, V.; Nakane, H.: Record-Breaking Chemical Destruction of Ozone in the Arctic during the Winter of 2004/2005, Izvestiya. Atmospheric and Oceanic Physics, 43, 592-598. https://doi.org/10.1134/S00014338070 50076, 2007. |
| 19. | Section 2.4 (mislabeled as a second Section 2.3 in the manuscript):
Again, important information is missing. When did you start the model run? See also comment to lines 194-195. | The description of the chemical transport model and the structure of numerical experiments has been expanded. |
| 20. | Line 180:
Smyshlyaev et al., 2017 is only available in Russian language (at least when I access it through the doi). This is very unfortunate, because I can't read it. I don't know what the guidelines of ACP are regarding this, but possibly you have to remove this reference. | Done
English version of this study on Springer
https://link.springer.com/article/10.113 4/S0001433817030148 |
| 21. | Line 183:
5 degrees x 4 degrees is an extremely coarse resolution and is not state-of-the-art anymore (maybe it was 15 years ago). Is it possible to run the model in a higher resolution (say 2 degrees x 2 degrees or 1 degree x 1 degree) to exclude that the coarse resolution has any negative effects on the results? | One of the goals of our work was to demonstrate the possibility of using a model with a rather coarse spatial resolution to analyze general and local features of changes in the ozone content in the Arctic. Comparison with the results of satellite measurements showed that such a model is capable of at least qualitatively reproducing local features. Certain quantitative differences with the observational results can, of course, be associated, among other things, with a coarse spatial resolution, but a pretty good qualitative agreement between the results of modeling and measurements makes it possible to analyze the influence of dynamical and chemical factors on the variability of Arctic ozone. |
| 22. | Line 184:
You should make sure that you don't look at air masses for your "passive ozone tracer" from the noCHEMAll run that were initially (at the start of the model run) at the upper model boundary or above, because this leads to meaningless values for | The upper boundary of the model is at the mesopause level (about 90 km); therefore, the influence of the upper boundary on the transfer of the passive |

the "passive ozone". From model runs that I performed for 2019/2020 with an upper boundary at 50 km, I estimated that values for the ozone loss above 550 K are not reliable anymore with the "passive ozone tracer" method.

tracer, in our opinion, should not affect the ozone variability in the range of 15–40 km. All dynamical conditions (winds, temperature, humidity and surface pressure) were the same for model experiments with and without chemistry. In an experiment without chemistry, chemical production and the destruction of ozone and all other gases simply vanished. The degree of gas-phase destruction of ozone is undoubtedly determined not only by local destruction at certain altitudes, but also by the vertical and horizontal divergence of its fluxes. However, we do not assert that the gas-phase destruction of ozone predominates at the same altitudes as that caused by heterogeneous processes on the PSCs, but we estimate the role of gas-phase chemistry in general. And our estimates show that this role is quite high.

| 23. | Line 186-187:
It is a rather unusual choice for the PSC scheme that it is based only on STS clouds. Can you elaborate a little bit on the reasons for that (not only in the reply, but also in the manuscript)? Why don't you simulate NAT and ice clouds in addition? Note that I am aware that the addition of NAT and ice clouds would probably only have a small effect on your results, since the heterogeneous reactions are usually sufficiently fast and since the temperature dependence is similar for NAT and STS clouds. But you possibly introduce some uncertainty by this, this should be discussed. | Our scheme of the formation and evolution of PSC s takes into account STS, NAT and ice. In the modified version of the article, we have provided a more detailed description of the heterogeneous block of the model. More details are given in our publications (DeZafra and Smyshlyaev, 2001, Sovde et al., 2008 and Smyshlyaev et al., 2010). |
|---|---|---|
| 24. | Lines 194-195:
This is not quite clear to me. Do you want to say that you initialize your "passive ozone tracer" from the noCHEMall run for estimating ozone loss on 1 November? But only north of 64 degrees? What are you doing in the following days? Calculating chemistry south of 64 degrees and then switching chemistry off when air masses are transported inside the 64 degree latitude circle?
This is of particular importance for your method to determine ozone loss. You set the reference date here for your determination of ozone loss. Ozone loss is extremely sensitive to the start date for the passive ozone tracer. If you choose a date too early, you will get a significant contribution from ozone loss from outside of the vortex caused by $NO_x$ cycles (since the air masses that are inside the vortex and the end of the model run would have been far outside the vortex at the start of the model run). Or you get loss from $NO_x$ cycles in autumn before the formation of the | In a model experiment without taking into account chemistry, we simply zero out the chemical production and destruction for all gases from November 1 to May 15 (in the new version of the article) north of 60 degrees (in the new version of the article to reduce the influence of middle latitudes). For other latitudes and on other days, chemical production and destruction are calculated as in the basic model experiment. The results of our model experiments show that the calculation results practically do not differ for the |

| | | |
|---|---|---|
| | vortex, when there still is sunlight. And if you set the start date too late, you miss some ozone loss. In my experience, a date between 15 December and 1 January works best for 2019/2020. November 1 is probably much too early.

And please state clearly what you are doing here. I.e., setting the reference date for calculating ozone loss. This is probably not clear to the majority of readers.

If you switch off chemistry only north of 64 degrees, that makes it hard to reason about the results. This way, your passive ozone tracer will not really be "passive", but some mixture of air masses that did experience ozone loss and air masses that did not. This makes it hard to understand what is actually shown in Figs. 12-15. Certainly not just "the" ozone loss.

Maybe this method would have worked if you would have chosen a PV contour at the edge of the vortex instead of 64 degrees. But it still would be a problem that there is probably ozone loss by $NO_x$ cycles in November at high latitudes, since there is still sunlight. But in any case, this would need much more explanation. The reasoning behind your setup is not at all obvious to the reader. I think I figured it out after some time (you want only to count ozone loss in the vortex and get rid of the loss by $NO_x$ cycles outside of the vortex by switching on chemistry south of 64 degrees), but that does not get clear at all. And probably, 64 degrees as a boundary will not work because this always includes air from outside the vortex. | basic variant and the variant without chemistry until the beginning of February (Figures 12-15). Hence, it seems unlikely that a change in the beginning of the zeroing period for chemistry can affect the calculation results. In June, the calculation results for different scenarios also differ little, which suggests that the sliding initialization of model time steps, which differs for different scenarios, has little effect on subsequent seasonal features. |
| 25. | Line 197: *"Comparison of the baseline scenario with these two additional scenarios makes it possible to estimate the periods when the chemical destruction of ozone is most effective after heterogeneous activation on the PSC surface, and when the gas-phase destruction of ozone in nitrogen catalytic cycles is more significant."*

You are very probably not doing this correctly or as you intended it. Switching off heterogenous chemistry on PSCs (or not forming PSCs at all in the model) will also affect the partitioning and chemistry of $NO_y$ and $NO_x$, and it will affect denitrification. E.g., the heterogenous reaction $N_2O_5+H_2O$ will be important for the partitioning. And effectively switching off denitrification will lead to higher $NO_x$ in the "noPSCaer" run, which could lead to more ozone depletion by $NO_x$ compared to the reference run. This way, you will obtain a different result as if you have kept $NO_x$ constant. What is not quite clear to me: Do you also switch off chemistry on the binary background aerosol at higher temperatures (which would also lead to unwanted changes)?

It is very likely that you will not obtain the result that you are hoped for: A clean separation of the amount of ozone that is depleted by chlorine and bromine cycles from the amount of ozone that is depleted by $NO_x$ cycles.

Therefore alone, I would recommend to delete all of the discussion on the contribution of the nitrogen cycles to the chemical ozone loss. I don't think that results are reliable.

A clean way to do this correctly would be to keep track of it in the chemistry module of your model, by e.g. looking at the rates of the rate-limiting steps of the different catalytic ozone destruction cycles. A study that shows how to do this correctly and that shows that heterogenous chemistry dominates ozone loss is Wohltmann et al., 2017. | In the updated version of the article and in the supplement, we have added an analysis of the vertical features of the ozone content change for different scenarios. This analysis demonstrates that the gas-phase destruction of ozone does not compete with the destruction initiated by heterogeneous processes, but occurs at altitudes above 25 km. Denitrification does not significantly affect the ratio of halogen and nitrogen cycles below 25 km, where chlorine and bromine cycles dominate. |

| | | |
|---|---|---|
| 26. | Lines 207-210 *"On the other hand, in the absence of the Sun, the chemical destruction of ozone does not yet reach high values associated with the previous halogen activation on the surface of polar stratospheric clouds, therefore, it can be assumed, that there should not be extremely low values of TCO in the region of absence of observations by the OMI instrument."* Delete this sentence. This is not correct. It is well known that ozone values inside the vortex are often relatively homogeneous. It is also well known that the movement of the vortex and the movement of air inside the vortex cause a homogenization. Air is often processed by the PSCs like in a "flow processor", and air masses are transported into the sunlit regions of the vortex and move back into the dark regions again. Of course, there are exceptions, and air masses may sometimes remain in darkness for a long time. But what you are doing here is pure speculation. And you don't even need to speculate. There are measurements from e.g. the MLS instrument which show the regions that OMI can't measure (and which already have been used for studies of this winter). Why don't you base your discussion on these measurements? | This sentence is not principle and it is deleted from the manuscript. |
| 27. | Lines 214-215: *"Again, the minimum values are detected along the border of the region of absence of observations - the zone of polar night."* See above. Delete this statement. | Done |
| 28. | Lines 202-220: While I won't judge your paper by relevance, I wonder whether this detailed description of the position and movement of the polar vortex is really necessary. I don't really see the scientific significance of these results. In addition, this can easily be deduced from Figure 1. Furthermore, the development of the vortex has been described elsewhere in studies that already have been published, e.g. some figures in Manney et al., 2020, and more importantly, Dameris et al., 2021. Their figure 1 has a large similarity to your Figure 1. | We believe that the analysis of Fig. 1 is useful for understanding the evolution of the polar vortex during the spring of 2020, as well as for comparison with the simulation results (Fig. 7) and for understanding the differences at stations in the Eastern and Western hemispheres (Fig. 12-15). In the updated version of the article, when describing Fig. 1, we added links to articles by Manney et al., 2020 and Dameris et al., 2021. |
| 29. | Line 227: *"The winter season 2019-2020 in the Arctic stratosphere was one of the coldest in the last 40 years."* Give references for this statement (e.g. Wohltmann et al., 2020, Lawrence et al., 2020) | Done |
| 30. | Line 229: Would be better to speak of STS and NAT clouds and not of Type I clouds for clarity. | Done

We add: (Nitric Acid Trihydrate (NAT) particles)

+ the same on line 412 |
| 31. | Line 232: Figure 2b is missing. | Done |

| | | fig.2b was removed after pre-review |
|---|---|---|
| 32. | Lines 245-246: *"The first period (February 7 - March 7) corresponds to strongest weakening of wave activity propagation in 2020, the second: January - February, the third: January - March.".*
Sorry, but I have no clue what you want to say to me here. Is a part of the sentence missing? | Done

We re-write 3 relevant statement

Furthermore, we compare the daily integrated zonal mean heat flux in the lower stratosphere at 70 hPa over three time periods in the winter of 2019-2020 and the five other winters with strong and cold stratospheric polar vortex and severe ozone destruction: 1995-1996, 1996-1997, 2004-2005, 2010-2011, 2015-2016 (Fig.3 b). These integrated zonal mean values were normalized by the number of days in each period. The first period (February 7 - March 7) is characterized by strongest weakening of upward wave activity propagation in the winter 2019-2020. The second and third periods are January - February and January - March respectively. It is seen that heat fluxes in first and second periods were the lowest in 2020 and in the third period only slightly stronger than in 1997. |
| 33. | Figure 3:
I would find it helpful to have some kind of colorbar for panels (a) and (c). Or at least to have the unit directly in the plots. It is given in the caption, but it is a little bit hard to bring this together. | **Done**

Title and units on Fig.3a & 3c were added. |
| 34. | Line 258: *"Where the absolute maximum of the average temperature in February was reached."*
Sorry, but again, I can't follow you. Do you mean that the highest temperatures over the course of the year were reached in February in Siberia. Or does the maximum refer to location? Please clarify. | **Done**

We split statement in two parts.

The second part is:

The record high February monthly mean temperature was achieved in Siberia |
| 35. | Line 260:
Do you mean "obtained" and not "retained"? Or what do you want to say? | Done

"retained" => obtained" |
| 36. | Large part of section 3.1 (Lines 221-end and Figures 2-4):
Without going through Lawrence et al. (2020) in detail, I have the impression that almost all of your section 3.1 only repeats what has already been written in the text and shown in the figures in Lawrence et al. Only for example:
Your Figure 2 is Figure 11a from Lawrence et al., | **Done**

Figure 2 **was moved in Supplement**

Figure 3a (new No 2a) |

| | your Figure 3a is similar in meaning to Figure 7f from Lawrence et al., your Figure 3c is Figure 6a from Lawrence et al., your Figure 4a is Figure 3a from Lawrence et al. While I think this was probably not intentional, I think this is problematic. You should really think about shortening this section, to delete some of the figures and to refer to Lawrence et al. where appropriate. | Fig. 7f from Lawrence et al. shows the daily time series of standardized **anomalies** in the 40-80 N average upward component of the EP flux.

Our Fig. 3a (new No 2a) shows real values of zonal mean heat flux in Nov 2019- Apr 2020.

The period of strongly reduced upward wave activity propagation is better described by our figure. Therefore we would like to keep it.

Figures 3c (new No 2c)

Our Fig. shows temperature anomaly averaged over the period of strongly reduced planetary wave propagation (7 Feb.- 7 Mar.) whereas Lawrence et al Fig.6a – JFM temperature anomaly. Therefore we would like to keep it.

Figures 4a **(NAM index) was moved in Supplement**

Overall the section 2.1 was shortened on 1 page. |
|---|---|---|
| 37. | Section 3.1: Meteorological data: What is the reanalysis data that you are using here for the temperatures and the EP fluxes? You only make a very general statement in section 2 that you use ERA5, NCEP, JRA and MERRA, and don't state anything in section 2.1. It makes the impression that you use a particular reanalysis here. Which one? | Done

Modified the statement on Line 146-147:

The propagation of wave activity was analyzed by using the zonal mean meridional heat flux and three-dimensional Plumb flux (Plumb 1985) calculated using NCEP reanalysis data |
| 38. | Line 267: *"and the negative one, on the contrary"*. Sorry, I can't follow you here. What do you want to say? That the negative phase shows opposing changes? I don't think you need to state that. This is obvious and follows from the definition of the AO. | Done

"and the negative one, on the contrary" - deleted |
| 39. | Line 267-270: *"With a positive AO phase, a stronger western zonal transport leads to milder winters, but with more precipitation in Southern Europe. In the negative AO phase, this transfer is weaker; as a result, cold air masses from the Arctic spread more strongly to the territory of Europe."* It seems to me that this needs a reference. | Done |

| 40. | Line 271: *"AO is the result of interaction between the dynamics of the stratosphere and the troposphere."* I don't know if I would phrase it like this. I would suggest to write something like "Interaction between the dynamics of the stratosphere and the troposphere can cause changes in the AO" or that the changes in the stratosphere (polar vortex strength) and troposphere (AO) are closely correlated. | Done

Interaction between the dynamics of the stratosphere and the troposphere can cause changes in the AO |
|---|---|---|
| 41. | Line 273-274: *"which is facilitated by an increase in the temperature gradient between the heated by the sun and shaded parts of the atmosphere"* I have no idea what you want to say here. Do you mean "associated" again and not "facilitated"? That would make more sense. But why do you mention the sunlit and dark parts of the atmosphere in conjunction with the temperature gradient? Assuming that you are talking about the zonal temperature gradient in the stratosphere, of course the polar vortex is in the end caused by radiative cooling in the polar night. But I think you are talking about interannual or seasonal changes in the temperature gradient related to polar vortex strength, which are not caused by changes in solar illumination (which is the same in every year). | Done

We agree and delete these two statements to make this section shorter. |
| 42. | Lines 275-276: What is cause and effect here? Less wave activity means a stronger and undisturbed vortex, and that in turn means altered conditions for wave propagation. Again, probably better to speak of correlation or association. | Done

The stratospheric polar vortex becomes less sensitive to the effects of waves, due to their refraction towards the equator.

We delete this statement |
| 43. | Line 284: Up to here, you talk of the AO. Now, you suddenly start to use the term NAM, which is just another word for the AO. And instead of talking of an AO index value which is just a number like 4 in the paragraph before, you start giving the value as 1.5 sigma. But probably I am not wrong when I suppose that these indices measure the same quantity. And what is sigma? The standard deviation of the AO time series, I suppose? | Done

Following two statements were added:

- (new line 289-290): Further to investigate the influence of the circulation of the Arctic stratosphere on the troposphere the changes of NAM index were analyzed.

- (new line 160-161): Notably that zonal-mean eddy heat flux, which is a proxy for the upward wave activity propagation, averaged over prior 40 days, is highly anti-correlated (-0.8) with the NAM index at 10 hPa [Polvani andWaugh 2004].
$\sigma$ – standard deviation of geopotential height anomalies averaged over 60-90°N. |

| | | We modify the statement (new lines 154-156):
The influence of the circulation of the Arctic stratosphere on the troposphere is analyzed through the propagation from the stratosphere to the troposphere of geopotential height anomalies from climatic values in the region of 60-90°N, normalized to the standard deviation (σ).

AO / NAO see below reply on comment # 45 |
|---|---|---|
| 44. | Line 284-285:
I have no idea what you mean by "spread continuously". Do you mean "propagate downward" or "extend into the troposphere"? Or that there is a clear signal in this time period? | Done

spread continuously =>

propagate downward from the middle stratosphere …. |
| 45. | Line 286:
At least in the stratosphere, the plot shows high values of the AO index up to the end of April. | Done

Fig.4a shows NAM index

Generally AO changes could be associated with changes of NAM but not always.

Monthly mean AO index in winter – spring 2019-20:
Dec: 0.4; Jan: 2.4, Feb: 3.4, Mar: 2.6, Apr: 0.9.
https://www.cpc.ncep.noaa.gov/products/precip/CWlink/daily_ao_index/monthly.ao.index.b50.current.ascii.table

The strong stratospheric polar vortex is often accompanied by a positive AO phase (Thompson and Wallace, 1998). But not always |
| 46. | Line 286:
You suddenly talk of a SSW in March which you never have mentioned before. It would be helpful to introduce the warming before you refer to it. It would also be helpful to have a reference or some more explanation. As far as I can see from e.g. PV maps, the vortex was quite stable until the end of April (although there was warming at the end of March). | Done

We include SSW related info (and describe it as observed in late March (instead of mid-March)

Enhanced upward wave activity propagation over about 10 days was observed in the stratosphere in the middle of March (see Fig.3) and was followed by SSW event in late March. |

| | | Strongest temperature increase associated with this SSW was observed in the upper and middle polar stratosphere. The polar cap lower stratosphere temperature increase was less than 10 K (see Supp. Fig.S1) and the vortex in the lower stratosphere was quite stable until the end of April. However this SSW event had a strong impact on lower stratosphere ozone-related chemistry (chemiscal processes). The temperature increase related to this SSW led to abrupt decrease of PSC NAT volume from nearly 55 mln $km^3$ to close to zero values in the last days of March (as it seen from MERRA-2). Sattelite OMPS LP observations aslo show abrupt decrease of PSC area at this time (Fig.6, Deland et al., 2020). Analysis of MLS observations (Manney et al., 2020) shows that ClO values in the lower polar stratosphere dropped from its highest values observed since early March to close to zero values in the last days of March (see their Fig.2). As it could be expected the HCl values showed abrupt increase since the beginning of April. Out trajectory analysis also shows the ozone loss decrease in the lower stratosphere at the end of March in air masses descending from 475 K to 450 K (Fig.7).

 Therefore the SSW event in late March led to a relatively abrupt stop of chemical ozone depletion. That in turn prevented further extension and strengthening of Arctic ozone anomaly. |
|---|---|---|
| 47. | **(Note: Lines 288-322 are not really my area of expertise. I hope another reviewer can say more to this. I cannot judge whether the results are scientifically sound or not)** | |
| 48. | Line 290-292: *"It is known that the main source of wave activity propagation into the stratosphere, characterized by the maximum of the vertical component Fz of Plumb's fluxes, (e.g. Jadin 2011) is located over this region [north-Eastern Eurasia]."*
 This is not really my area of expertise and this may be correct. But this is a rather bold statement, and I have never heard of this | Done

 Removed reference on |

| | | |
|---|---|---|
| | before. Unfortunately, the only reference that you give is from a predatory journal, and therefore, is no reliable source and I refuse to read it. Please give references from legitimate peer-reviewed sources or delete this statement.
In any case, remove the reference. | Jadin 2011 was substituted by

Zyulyaeva and Jadin, 2009 |
| 49. | Lines 294-295:
Since you already stated this, you should refer to your earlier statements. E.g. "As shown above…" | Done |
| 50. | Figure 4a:
The green contour mentioned in the caption is missing from the plot.
 The units for the colorbar are not given (neither in the plot nor in the caption).
Arial 36 | Done

We add green contour, but moved this figure to Supplement |
| 51. | Figure 4b and c, Figure 5a and b:
Same comment as to Figure 3. Would be helpful to have the units for the contours directly in the plots. | Done |
| 52. | Lines 306-307:
You are again talking about a SSW event which you never have introduced before. | Done

See our reply on comment #46 |
| 53. | Lines 331-339:
Seems like part of the description of your method is in these lines and the other part is in section 2.2. Can you please describe the method only at a single place? Either, you have to move these lines to the description of the method in section 2.2. Or vice versa (and then delete section 2.2).
Some of the questions I had for section 2.2 are answered here, but others are not (see also following comments and comment to section 2.2) | - Changed |
| 54. | Lines 332-333: *"For simplicity the trajectories were initiated uniformly distributed on the 85 N latitude circle, when it was completely located inside the polar vortex"*
For no apparent reason, you start trajectories only on the 85 degrees latitude circle instead of sampling the vortex homogenously, which would have been easy. This introduces a bias which you could easily have avoided. Please repeat your calculations with trajectories that sample the vortex homogeneously.
You need to test whether a trajectory is inside the vortex in any case, so that should not introduce any additional effort. Unfortunately, you don't give any information on your criterion for testing whether a trajectories is located inside the polar vortex. Do you use a fixed PV contour (say 36 PVU), equivalent latitude or the edge as defined by Nash?
It should also be easy to sample the vortex homogeneously, e.g. by starting trajectories on more than one latitude circle and starting less trajectories on latitude circles closer to the pole to make sure that every trajectory represents the same area, or by using a random generator to distribute points evenly (the only thing you have to take care of is taking the arcsin of the random | - Changed

At present the backward trajectories were initiated at the 400K, 435K, 460K and 500K levels at the latitude circles between $70^0$ N and $85^0$ N with $0.5^0$ resolution in latitude and $0.5^0$/cos(latitude) spacing in longitude. The dates for the trajectory initiation were chosen when that domain was mainly located inside the polar vortex. For winter 2019-2020 it was April 10, for winter 2010-2011 it was March 26 and for winter 2015-2016 it was February 26 due to the early SSW. The end date of all trajectories is December 1. Trajectories were calculated using ERA5 vertical wind. |

| | | For comparisons with backward simulations at level 460K, the forward trajectories were calculated in the similar way by initialization in December at the level 500K. Results of comparison of backward and forward simulations are presented in Supplement, but not included into the text of paper. |
|---|---|---|
| 55. | Line 335: *"and mostly remained inside the vortex"*
You give no information what you do with trajectories that leave the vortex. Do you ignore that or do you sort them out? This is potentially important for the results. | - Changed |
| 56. | Line 332, 339:
You start the trajectories only on two levels, 475 K and 550 K. I think that would have been fine if you would have used backward trajectories, but it is unfortunate with forward trajectories. The quantity that you usually would like to know is the ozone loss at some level in spring at the end of your trajectory run (i.e. in a given well defined air mass). The trajectories which you start at 475 K or 550 K will not only descend, but will also cover a range of potential temperatures (and horizontal locations) at the end of the run, which makes the ozone loss hard to interpret.
The very least you could do is to give the range of potential temperatures that is covered by the trajectories at the end of the run (and the mean value to estimate the diabatic descent).
It would however be much more straightforward to base your method on backward trajectories. | - Changed

At present the backward trajectories were initiated at the 400K, 435K, 460K and 500K levels, providing ozone loss estimates in the layer 15-22 km. |
| 57. | Line 338: *"average ozone value"*
I assume you mean the average over all trajectory locations at this date? | Changed

average over all trajectories at this date |
| 58. | Line 355-356:
I would expect this information earlier in the description of the method. | - Changed |
| 59. | Line 358: *"overestimation"*.
You don't know whether this is an overestimation (compared to "reality") or not. I would say "leads to higher estimates for…" | Changed

Since we obtained slightly different descending rates in backward and forward simulations this sentence was deleted. |
| 60. | Line 367-368:
Please give the exact dates. Does beginning of January mean January 1? What is end of March? | Done

instead of "from the beginning of January to the end March 2020" it reads now "from 3 January to 26 March 2020" |
| 61. | Line 369: *"Significant ozone loss had been seen from the mid-January till the end of March between 400-525 K isentropic levels* | Done |

| | | |
|---|---|---|
| | *(~15-22 km)."*
Where does the information mid-January until end of March come from? Is this your result from your method? This cannot be deduced from the figure, so you should state that more clearly. E.g. "Analysis of the ozone sonde data shows that significant ozone loss is observed from mid-January to end of March between…" | Changed as it was recommended.

"Analysis of the ozone sonde data shows that significant ozone loss was observed from the mid-January till the end of March between 400-525 K isentropic levels (~15-22 km). |
| 62. | Lines 377-378:
Are these values your results or the results of Peters (2010)? | Done

Right reference is: Peters et al., 2010 and we (N.D.T & P.N.V.) are co-authors of this study |
| 63. | Line 382-384: *"The use of temperature, wind speeds, surface pressure and air humidity from the reanalysis data made it possible to simulate the effect of atmospheric circulation on the transport of ozone and associated gases close to reality. Variability of specified dynamical parameters determines the dynamical decrease in ozone content, as well as the atmospheric temperature govern the rate of chemical reactions, polar stratospheric clouds formation and the rate of heterogeneous reactions on their surface, which determine the chemical destruction of ozone.".*
Delete these sentences. They have no information content and are phrased awkwardly. Everybody in the community knows what a CTM is good for. The information given here is much too basic. | The sentence has been deleted. |
| 64. | Figure 9:
There is a strange zig-zag pattern in some of the contours in the plots. It seems that there is a bug in your plotting or interpolation functions. | We have changed the interpolation method when drawing pictures in the updated version of the article. |
| 65. | Figure 9:
Would be nice again to have the units in the plots. | Done |
| 66. | Figure 9:
Please indicate the vortex edge in the plots. | SPS |
| 67. | Line 390: *"First, a basic numerical experiment was performed taking into account all factors affecting Arctic ozone".*
I think what you wanted to say here is: "First, a reference run with full chemistry was performed." Your sentence sounds odd. | The sentence has been changed. |
| 68. | Line 390-391: *"The variability of the atmospheric gas composition during the winter of 2019-2020 was calculated in the basic numerical experiment"*
Again, you state the obvious. Delete the sentence. | Done |
| 69. | Line 395-396: *"In particular, its movement in the eastward direction and the minimum values at the beginning of March in the Northern part of the European territory of Russia are reproduced".*
Well, if you would not reproduce these very basic features, you would have a problem anyway. I don't know whether this sentence is really needed | Our opinion that this sentence is necessary to compare model results and observations. |

| 70. | Line 396-397: *"In mid-March – in the western part of the Arctic, in the area of Greenland, Svalbard and Franz Josef Land in early April and North to the mainland of the ETR in mid-April."* This is not a complete sentence, and therefore, unintelligible. Delete. And if it would be a complete sentence, I have the feeling that the information given here would be irrelevant. You don't need to state the position of the vortex every few days and every location that it covers. This is not only easily visible in the plots, but also no relevant or scientifically interesting information in my opinion. | This sentence has been deleted. |
|---|---|---|
| 71. | Line 399-400: *"The model results demonstrate that the minimum values are observed at the boundary of the polar night in the part where the Sun has already returned"* I find this statement problematic since the vortex is constantly moving. And I don't think it is correct, see my earlier comment on line 207-210. It also hard to see in the plots because there is no line showing the area of polar night. Delete the sentence. | This sentence has been deleted |
| 72. | Lines 400-402: *"This confirms the hypothesis of the effect of the chemical destruction of ozone, which intensifies after heterogeneous activation in polar stratospheric clouds and the return of the Sun."* This is basic textbook knowledge that everybody who reads this paper is aware of. That would be fine for the introduction, but not for the main text. In addition, it is phrased quite awkwardly, up to the point that it is not quite correct or very hard to understand what you mean. Delete the sentence. | This sentence has been deleted |
| 73. | Lines 402-404: *"However, the results of model calculations reveal that relatively low values of the total column ozone (below 300 DU) are also observed in the polar night zone, where the chemical destruction of ozone is slowed down. This also indicates a significant influence of dynamical factors on the formation of regions of low ozone content."* This is again basic knowledge, misleading and scientifically not relevant. I start to wonder whether the author of this section has a basic lack of understanding of the relevant science. The total column value is a cumulative quantity, which is not determined by the position of the vortex at a particular date. In addition, the Arctic vortex is more dynamically active than the more circular Antarctic vortex, where it is more likely that air masses stay in darkness for a long time. I think the only part of interesting information here would be the minimum column values. I also acknowledge that the discussion of dynamical factors can be interesting, but that would include things like interannual variations in diabatic descent or ozone mini-holes. The simple fact that air inside the vortex is moving and that the vortex is moving as a whole does not belong to this. This could be interesting if the authors would have done a detailed trajectory study of the history of air masses, but stating the obvious here is not enough in my opinion. | This sentence has been modified. Information is given on low ozone values in the area not covered by OMI measurements. |
| 74. | Line 406-407 *"To better understand the relative role of dynamical and chemical processes in the formation of the Arctic ozone anomaly in spring 2020"*. I don't understand why it leads to a better understanding of the | This sentence has been modified. |

| | | |
|---|---|---|
| | *relative* roles of dynamics and chemistry | |
| 75. | Line 407:
"Type I cloud" is ambiguous. Please clarify whether you mean STS or NAT clouds. | It is indicated that PSC 1 is the sum of STS and NAT. |
| 76. | Line 408-409: *"but the inertia in their melting with increasing temperature is also taken into account.".*
Would be worth noting here that your CTM does not use an equilibrium scheme as some other CTMs and to cite Smyshlyaev et al., 2010. | Done. |
| 77. | Lines 414-416: *"which suggests that the relationship between the formation of PSCs and ozone destruction is not linear and confirms the theory of several stages of the formation of ozone anomalies."*
Apart from the fact that this sentence is phrased very awkwardly, this again states textbook knowledge and the obvious. Delete the half-sentence. | This sentence has been reduced. |
| 78. | Line 416-417: *"In the polar stratosphere, associated, first, with the formation of PSCs, then with halogen activation on their surface, and only then with ozone destruction after the return of the Sun after the polar night."*
Again, this is basic knowledge. Delete the sentence. This would be fine in the introduction, but not as a scientific result in the main text. And it is phrased so awkwardly that it is almost unintelligible. | This sentence has been deleted |
| 79. | Figure 10, Figure 11:
Indicate the edge of the polar vortex. It seems to me the area covered by PSCs is much larger than the polar vortex. Is this really correct? | The figures has been corrected. |
| 80. | Figure 10, Figure 11:
Again, give the units in the plots. | Done. |
| 81. | Figure 11 caption:
"low stratospheric coefficient of ozone destruction" does not give enough information to find out what is shown here. | The figure caption has been extended with explanation of coefficient of ozone destruction and altitude range. |
| 82. | Line 425: *"the coefficient of chemical ozone destruction in the lower stratosphere"*
This is introduced as it would be a well-known quantity (known by everyone under this name), but I think in fact it will cause confusion for many readers. E.g., what does "lower stratosphere" mean here? There is no altitude range mentioned in the following. This is important information that you need to give here. | Altitude range has been described. |
| 83. | Line 425-426: *"This coefficient is a factor by which the concentration of ozone should be multiplied in order to obtain the rate of its chemical destruction."*
This definition seems not to be consistent with the magnitude of the values shown in Figure 11. If I understand you correctly, a value of 1/s would mean that all of the ozone at a particular location would be depleted by chemical processes in 1 second. But the figure shows values on the order of $10_8$ per second. Either I have difficulties to understand your definition or there is something wrong with the magnitude of the values given in the | Units corrected |

| | plot. | |
|---|---|---|
| 84. | Lines 425-426:
Just to make sure that nothing is going wrong here. You take into account that there is a fast equilibrium between O and $O_3$ and only look at the net change of $O_3$? | The definition of coefficient has been corrected. It is applied into odd oxygen instead of ozone. However, in the low stratosphere odd oxygen is almost coincide with ozone concentration. |
| 85. | Lines 425-426:
Do you show and discuss instantaneous values at a given point in time (say 12 UTC) or daily averages? The plots make the impression that the latter is the case. But you don't tell us anything about that. This is important. Please clarify. | It is clarified that there are daily averaged values. |
| 86. | Lines 425-426:
I have the impression there might be quantities that would be more easy to understand which you could show here, e.g. simply the rates in ppb/day or a similar unit, or the reciprocal of what you show
here (the time scale needed for complete ozone destruction). But maybe my confusion is just caused because I have difficulties to understand your text. A better explanation and definition may help here. | We use this coefficient instead of odd oxygen rate of destruction because the rate follows ozone concentration, while this coefficient does not depend on ozone concentration. |
| 87. | Lines 428-429: *"It should be noted that the rate of ozone destruction in March has its maximum values at the boundary of the polar night in the region of the newly returned Sun."*
This is hard to see, because the boundary of the area of polar night is not shown in the plots. And looking at Figure 11, I doubt that this statement is correct (assuming that you show daily averages). Delete the statement. | Done |
| 88. | Lines 430-434: *"In this case, the maximum rate of ozone destruction is a necessary, but not sufficient condition for the formation of a zone of low ozone content in the spring. Dynamical factors also play an important role, in particular, for definition of the zone where the polar vortex is located. In particular, in early March, the rate of destruction of the base is maximum over the entire circle of latitude near the boundary of the polar night, and the minimum values of the ozone content are noted only in the eastern hemisphere (Figs. 8 and 9). Also in mid-March and April, areas of high ozone depletion cover a wider zone than areas of minimum total ozone."*
Delete all of these statements. First of all, you obviously can't deduce the cumulative chemical destruction of ozone over a longer time period (that causes the low ozone columns) from the chemical rates at a single date, because the values add up. You don't need to argue with dynamical reasons here. This is really a basic flaw in your reasoning here.
And with respect to dynamics: And again, you are stating the obvious here. The polar vortex and the air contained in the vortex are moving. It makes no sense to note that the minimum values of ozone are over the eastern hemisphere | Done. |
| 89. | Lines 436-439: *"For a more detailed study of the influence of dynamical and chemical factors on the local variability of the ozone content Figures 12 - 15 present the simulated with the CTM* | We are interested in ozone variability at a single point. |

| | | |
|---|---|---|
| | *and measured by the OMI instrument changes in the total ozone content at four stations (two in the Western Hemisphere and two in the Eastern Hemisphere) during six months from the beginning to mid-2020.".*
 I have a general comment here: While it is certainly fine to perform case studies like this for single locations, it would be have been so easy here to make more general and scientifically relevant statements by looking at vortex means (which should have been easily possible). I don't really see why the situation at a particular location is so scientifically interesting, but that may be my personal opinion. I think that you wasted a chance here without necessity. | |
| 90. | Lines 436-439:
 It seems to me that the information given in Figures 12-15 is largely redundant. Figure 12 and 13 (Pechora and Tura) show almost identical results. The same is true for Figure 14 and 15 (Resolute and Eureka). You could easily do with only two figures here. I would suggest to shorten the text in lines 450-526 significantly. | Done. |
| 91. | Line 440:
 Is there any reason why you use SBUV data here and OMI data earlier in the manuscript? | SBUV data cover a longer time interval than OMI data. |
| 92. | Line 442:
 Earlier in the manuscript, you give a value of 64 degrees N and not 66 degrees N. Only one of these values can be correct. Please correct. | Corrected for 60 degrees for the revised version of the manuscript. |
| 93. | Lines 444-446: *"Comparison of the baseline scenario with these two additional scenarios allows us to estimate the periods when chemical destruction of ozone is most effective after heterogeneous activation on the PSC surface, and when gas-phase destruction of ozone in nitrogen catalytic cycles is more significant"*
 See major comment 1. There is something fundamentally going wrong here. | We are talking about different altitude intervals. |
| 94. | Lines 446-448: *"In addition, the comparative role of dynamical and chemical processes of ozone reduction can be assessed by comparing these scenarios with each other and to mean climatic values presented at the bottom of these figures."*
 While I agree that you can disentangle dynamical and chemical changes when looking at these plots, this sentence is easy to misunderstand. E.g. you can't assess the comparative role of dynamical and chemical processes from comparing the "PSC" and "noPSCaer" runs. | ??? |
| 95. | Lines 450-526:
 I think this part can be shortened considerably, not only for the reason stated above (lines 436-439). It is a little bit tiring that this is basically a description of what you can see in the figures (in very much detail), without so many scientifically interesting results. | Done. |
| 96. | Line 461-463: *"Based on a comparison of the noPSCaer and noCHEMall scenarios, it can be concluded that in the chemical* | We are talking about different altitude |

| | | |
|---|---|---|
| | *destruction of ozone at Pechora station, the heterogeneous part is about one third (~ 25 DU), and the gas-phase part is ~ 45 DU."* See major comment 1. This can't be correct. Unfortunately, you don't indicate in the plots when the station is located inside the polar vortex. | ranges. |
| 97. | Figures 12-15: Indicate when the station is in the polar vortex. This is *very* important to be able to interpret the results correctly. | The position of the vortex is not specified in the model. Its position is automatically taken into account in the specified fields of wind speed and temperature from the reanalysis data. |
| 98. | Figures 12-15: In case you don't find the problem, remove the noPSCaer run from the plots. | ??? |
| 99. | Figure 12-15: In the b panels, I would have found it more intuitive when the blue line would have been the difference between "noCHEMall" and "noPSCaer". | ??? |
| 100. | Lines 471-481: This is largely redundant with the paragraph about Pechora. I won't comment in detail and suggest to delete this. | Done. |
| 101. | Line 495: *"It should also be noted that there are two peaks of maximum chemical destruction of ozone: in late March and mid-April."* This is only the observation at this location because of the movement of the vortex. If you would look at the same air mass, this would be different. | We are talking about a specific point at which two peaks are marked. |
| 102. | Line 496-500: *"At the same time, chemical destruction in the second half of March is superimposed on a dynamic decrease in its content, which leads to a minimum in the seasonal variation of the total ozone content, while in April, when the chemical destruction of ozone is even greater than in March, the polar vortex is already shifting towards the eastern hemisphere (Fig. 8 and 9), and the total ozone content is higher than in March."* I find this sentence very hard to understand and unintelligible. For example, what do you mean by minimum in seasonal variation? | |
| 103. | Line 507-509*: "Comparison of calculations for different scenarios of accounting for the chemical destruction of ozone depicts that the destruction of ozone over heterogeneous reactions in the western hemisphere exceeds 30 DU, which is more than in the eastern hemisphere, while the gas-phase destruction of ozone in the Western hemisphere is greater than in the Eastern Hemisphere."* Delete these sentences. See major comment 1. There seems to be a fundamental flaw in your method. | Done |
| 104. | Lines 510-513: *"It should also be noted that in the Western Hemisphere, the minimum values of the ozone content according to satellite measurements in March are lower than the values calculated using the model, while in the Eastern Hemisphere the satellite and model results are closer. This result may be due to* | Done |

| | | |
|---|---|---|
| | *relatively coarse model resolution to simulate fine local effects in the western hemisphere.”*
Delete these sentences. You show a fundamental lack of understanding of the processes here. Since the vortex is moving, air masses that are located in the eastern hemisphere will be located somewhere else a few days later. This has nothing to do with the hemispheres. | |
| 105. | Line 521-524: *“Additional numerical calculations to assess the effect of various catalytic cycles of chemical ozone destruction on a decrease in its content in April-May 2020 revealed that the main increase in the gas-phase ozone destruction occurs in the nitrogen catalytic cycle, in which the chemical reaction with the participation of nitrogen dioxide and atomic oxygen plays a determining role.”*
Since there is a fundamental flaw in your method (major comment 1), these results are very likely not correct. Delete this sentence. | Done |
| 106. | Lines 524-526: *“In the Arctic stratosphere, in contrast to the Antarctic stratosphere, significant denitrification does not occur, and therefore a sufficient amount of nitrogen oxides remains in it, which plays a decisive role in the destruction of stratospheric ozone.”*
This statement is not correct, and it would have been easy to see that if you would have looked into the literature (e.g. Manney et al., 2020). Note that this statement is not correct in general, and not only for the winter 2019/2020. There are many Arctic winters which show a significant amount of denitrification, this is basic knowledge. Delete the sentence.
In addition, it seems that you use it here as an (wrong) explanation for your flawed results. I wonder why you did not notice that something must be wrong here. | Done |
| 107. | Lines 528-531:
This sentence is phrased so awkwardly that I have a very hard time to understand what you want to say. It is almost unintelligible. Please rephrase. I won't give a suggestion here, because this is the conclusions and I am not sure what you want to tell us. | |
| 108. | Line 535: *“The of SSW event in the middle of March 2020”*
This part of the sentence makes no sense. What do you want to tell us? | Done
The SSW event in late March 2020 led to a relatively abrupt stop of chemical ozone depletion and prevented stronger ozone layer destruction.
See plots in the Supplement |
| 109. | Line 535: *“although it did not satisfy the WMO definition of Major SSW event”*
Earlier in the paper, I had some comments that you were referring to a SSW event that you never mentioned before. And now, in the last lines of the paper, you tell me that it actually was no SSW event. What does this mean? That I can forget about everything that I have read about the SSW event? This information should have been given much earlier.
**I agree that the warming of the vortex in late March led to a** | Done

See above |

| | | |
|---|---|---|
| | **relatively abrupt stop of chemical ozone depletion**. Can you rephrase this. | |
| 110. | Lines 537-545, 554-557:
This is not my area of expertise. I will skip this part. | |
| 111. | Lines 558-560:
You need to be more specific here. You have only results for two potential temperature levels. You don't mention that you refer to vortex means. You don't mention the date you are referring to. And give numbers for the ozone loss. | ANL

Changed |
| 112. | Line 566-569: *"...reveal that both dynamical and chemical processes make significant contributions to the decrease in the ozone content inside the polar vortex. In this case, the chemical ozone depletion is determined not only by heterogeneous processes on the surface of polar stratospheric clouds, but by gas-phase destruction in nitrogen catalytic cycles as well."*
This is not correct and misleading. See major comment 1. Delete this from the conclusions. | We are talking about different altitude ranges. |
| 113. | Line 573:
It seems that there is something missing in the "Author contributions". It starts with "All other authors…", implying that a sentence is missing at the start. There is information missing who has written the main text. | **Done**

Statement "The paper was initiated and written by S.P.S. and P.N.V" was added. |

**Answers to Technical corrections (language etc.)**

| | | |
|---|---|---|
| 1. | Title:

"Dynamical and chemical processes contributing to ozone loss in the exceptional Arctic stratosphere winter-spring of 2020" (added "the") | Done |
| 2. | Line 8:

You can delete "The features". Just start with "Dynamical processes and changes…" | Done |
| 3. | Line 17:
"repeated" is probably not the perfect choice of word. Maybe "which was similar to the depletion in 2010/2011" | Done |
| 4. | Line 33:

Change "the main SSW" to "a main SSW" | Done |
| 5. | Line 38:

Change: "the largest decrease in the Arctic ozone was observed" to "the largest decreases in Arctic ozone were | Done |

| | | |
|---|---|---|
| | observed" | |
| 6. | Line 49:

You certainly mean "statistically" and not "statically" | Done |
| 7. | Line 52:

You probably mean something like "nevertheless" and not "in the meanwhile" | Done |
| 8. | Line 132:

Change "reveal" to something like "estimate" or "determine" | Done |
| 9. | Line 148 (158):

Change "the Lagrangian approach" to "a Lagrangian approach" | Done |
| 10. | Line 161:

Change "were interpolated into the points of each trajectory" to "were interpolated to the positions of each trajectory" | Done |
| 11. | Line 164:

Change to "Ozone sonde data … have been used" | Done |
| 12. | Line 168:

You misspelled the name in the reference. The correct name is in line 175 (Braathen). There is also a superfluous "," | Done |
| 13. | Line 176:

Section 2.4 is mislabeled as Section 2.3 | Done |
| 14. | Line 182:

Split the sentence and shorten: "Meteorological fields are specified…" | Done |
| 15. | Lines 184-186:

Awkward phrasing. Change to e.g. "The model includes 74 oxygen, hydrogen, nitrogen, chlorine, bromine, carbon and sulfate species. The chemistry of the species is calculated as described in Smyshlyaev et al. (1998)." | Done |
| 16. | Line 191-192:

Again, phrased awkwardly. Suggestion: "For a more detailed study of the influence of dynamical and chemical factors on the local variability of the ozone content, two additional numerical experiments with the RSHU CTM were performed | Done |

| | | |
|---|---|---|
| | in addition to the reference run (termed "PSC" here)." | |
| 17. | Line 204:
Change "at the early March" to "in early March" | Done |
| 18. | Line 206:
Do you mean "north of Alaska"? | Done |
| 19. | Line 207:
Maybe "which are based on solar radiation" is better English. | Done |
| 20. | Line 215:
Change "north to" to "north of" | Done |
| 21. | Line 217:
Change "values less than 220 DU" to "values of less than 220 DU" | Done |
| 22. | Line 220:
Change "territory" to "area" | Done |
| 23. | Figure 1 caption:
The text speaks of OMI and the caption speaks of AURA. Would be nice to have that consistent. | Done
Figure 1: OMI Arctic column ozone (Dobson Units) during 2020 spring: a- March 1, 2020; b- March 15, 2020; c – April 1, 2020; d - April 15, 2020 |
| 24. | Line 229:
Split sentence. Write something like: "Temperatures were sufficiently low to allow the formation of NAT and STS clouds". | Done |
| 25. | Line 229:
Can we stick to Kelvin and not to degree Celsius? | Done |
| 26. | Line 230:
Change "Figure 2a" to "Figure 2". There is only one panel here and 2a does not exist. | Done |
| 27. | Line 232:
Do you mean "Two main causes of the cold and stable Arctic polar vortex" and not "Two main causes of so cold and stable Arctic polar vortex"? Or what were you trying to say? | Done |
| 28. | Figure 2 caption:
Change "climate mean" to "climatological mean" | Done |
| 29. | Figure 2 caption:
There is a Russian letter in the caption (probably means "and") | Done |
| 30. | Line 242:
Change to "Furthermore, we compare…" | Done |

| 31. | Caption Figure 3:
Change "Latitudes are from 30 N" to "The map shows only latitudes north of 30 N" | PNV Done |
|---|---|---|
| 32. | Line 263-264:
"Notably that described positive temperature anomalies were observed not only near surface but at higher levels in troposphere." This is phrased awkwardly. Suggestion: "Positive temperature anomalies were observed not only near the surface but also at higher levels in troposphere." | Done |
| 33. | Line 266:
Change "and increased in" to "and increased pressure in" | Done |
| 34. | Lines 273-274:
You probably mean "between parts of the atmosphere that are heated by the sun and parts that are shaded"? But I think a native speaker probably wouldn't phrase it like this. I would talk of the sunlit part. | Done. |
| 35. | Line 294:
Change "In the same time" to "In the same time period" | Done |
| 36. | Line 304:
Change "till" to "until" | Done |
| 37. | Line 305:
Change "This is confirmed by the diagram with…" to "This can be seen in Figure 5a showing …" | Done |
| 38. | Figure 4 caption:
Change "climate mean" to "climatological mean" | Done |
| 39. | Line 316:
Change "display dominated" to "show pronounced" | Done |
| 40. | Line 334:
Change "descent" to "descend" | Done |
| 41. | Figure 7 caption:
Change "0 day…" to "The horizontal axis shows the number of days since December 1." | Done |
| 42. | Line 358:
Change "average vertical descending" to "average vertical descent" (this time the "t" is correct!) | Done |
| 43. | Line 360:
Change "As well to estimate chemical ozone loss" to "As another method to estimate chemical ozone loss" | Done |

| 44. | Line 361:
Change "Ny-Älesund" to "Ny-Ålesund" | Done |
|---|---|---|
| 45. | Line 367:
Change to "Figure 8 shows the vertical profile of the vortex-averaged cumulative ozone loss…" | Done |
| 46. | Line 369:
"with largest losses": Start a new sentence and write "These winters showed the largest ozone losses previous to the winter 2019/2020." | Done |
| 47. | Line 371:
Split into two sentences. "…than in 2010/2011. That is consistent…" | Done |
| 48. | Line 380:
Awkward phrasing. Change "For a more detailed study of the degree of dynamical and chemical processes influence on the formation of ozone anomalies…" to "For a more detailed study of the dynamical and chemical processes that influence the formation of ozone anomalies…" | Done |
| 49. | Line 381-382:
Awkward phrasing. Split into two sentences. Change "in which the dynamic parameters were set from the MEPRA-2 reanalysis data" to "Meteorological data were obtained from the MERRA-2 reanalysis". | Done |
| 50. | Line 382:
Note the change "MEPRA-2" to "MERRA-2" in the previous comment | Done |
| 51. | Line 391-392:
Change "Figure 9 demonstrates" to "Figure 9 shows" | Done |
| 52. | Line 392:
Awkward phrasing and a lot of repetition of information: Change "the results of calculations of the total column ozone for March-April 2020, performed using the CTM with the specified dynamical parameters from the MERRA-2 reanalysis" to "… shows the total ozone column for March-April 2020 from the CTM". | SSP |
| 53. | Line 410:
Replace "territory" by "region" or "area" | Done |
| 54. | Line 411:
Change "the area of the PSCs zone is maximum" to "the area covered by the PSCs is maximum" | Done |
| 55. | Line 413:
Change "the area covered by PSCs significantly reduced" to | Done |

| | "the area covered by PSCs are significantly reduced" | |
|---|---|---|
| 56. | Line 425:
Change "Fig. 11 demonstrates" to "Figure 11 shows" | Done |
| 57. | Line 454:
Phrased awkwardly. Change "which maximally affect the ozone depletion in April" to something like "Cumulative ozone depletion shows maximum values in April" | Done |
| 58. | Line 458:
Change "two times" to "by a factor of two" | Done |
| 59. | Line 460:
Change "if compare" to "when compared" | Done |
| 60. | Line 492:
"the total content fluctuates" This is phrased awkwardly. | SSP |
| 61. | Line 531:
Change "Further" to "Furthermore" | Done |
| 62. | Line 532:
Change "ozonosondes" to "ozone sondes" | Done |
| 63. | Line 532:
Split into two sentences: "…observations. Finally, …" | Done |
| 64. | Line 537:
Delete "revealed" | Done |
| 65. | Line 562:
I don't think that "repeat" is the best choice of word here. Maybe "rivalled" | Done |
| 66. | Line 586:
Change "Ny-Aalesund" to "Ny-Ålesund" | Done |
| 67. | Line 607:
Don't abbreviate "QJRMS" | Done
Q. J. R. Meteorol. Soc. |
| 68. | Lines 644-645:
Jadin et al. is an article from a predatory journal. Delete the reference. | Done, the other relevant reference is included:
Zyulyaeva, Y.A.; Jadin, E.A.: Analysis of three-dimensional Eliassen-Palm fluxes in the lower stratosphere, Russian Meteorology and Hydrology, 8, 5-14, https://doi.org/10.3103/S1068373909080019, 2009. |
| 69. | Line 686:
It seems to me that the reference Madrid et al. is not cited in the paper. Delete. | Done |
| 70. | Line 703-704:
Pedatella et al. is a news article. Delete the reference. | Done |

| 71. | Line 711:   Change "Lefe`vre" to "Lefèvre" | Done |
|---|---|---|
| 72. | Line 738-740:
Smyshlyaev et al., 2017 is only available in the Russian language, so I can't read it. See specific comments to line 180. | Done

English version of this study on Springer:

https://link.springer.com/article/10.1134/S0001433817030148 |
| 73. | Line 760-761:
I wasn't able to find this article on the home page of the journal. | Done
Instead of two references
Tsvetkova, et al., 2002 Tsvetkova, et al., 2004 another one was included:

Tsvetkova, N.; Yushkov, V.; Lukyanov, A.; Dorokhov, V.; Nakane, H.: Record-Breaking Chemical Destruction of Ozone in the Arctic during the Winter of 2004/2005, Izvestiya. Atmospheric and Oceanic Physics, 43, 592-598.
https://doi.org/10.1134/S0001433807050076, 2007. |

Thank you again for taking the time to review our manuscript.

With respect,
Sergei P. Smyshlyaev,
Pavel N. Vargin,
Alexander N. Lukyanov,
Natalia D. Tsvetkova,
Maxim A.Motsakov

---

## Author Comment (AC2)

Dear Gloria,

Thank you for your comments on the paper and constructive recommendations. We have tried to follow your suggestions and have taken into account most of them. Following we mention how the manuscript has been changed according to your comments.

**Major Comments:**

(1) Dynamical results from reanalysis data: The dynamical diagnostics shown from reanalyses and the discussion thereof are almost all things that have already been published in existing papers on the 2019/2020 winter; in addition, the authors are unclear about which reanalyses are used where and why -- in particular, the NCEP /NCAR reanalysis is deprecated for all stratospheric and polar processing studies and should not be used; and it appears that for any given diagnostic or model calculation, one reanalysis (though often it is not stated which) is used -- while comparing multiple reanalyses for each calculation is highly desirable and enhances the robustness of the results, using different individual reanalyses for different calculations does the opposite, since one cannot even evaluate the results as a whole knowing that they are based on the same representation of the atmosphere. Further, the construction of and/or interpretation of some of the diagnostics is unclear or inconsistent.

Reply:
Information on used reanalysis data was modified in Chapter 2.

(1) We used NCEP/NCAR reanalysis to analyze large scale dynamical processes as minor SSW and related wave activity enhancement. We supposed that such features are described comparable in most known reanalysis including NCEP/NCAR as it was shown in
+ Ayarzagüena, B.; Palmeiro, F.;  Barriopedro, D.; Calvo N.; Langematz, U.; Shibata, K. On the representation of major stratospheric warmings in reanalyses. Atmos. Chem. Phys., 19, 9469–9484,
https://doi.org/10.5194/acp-19-9469-2019, 2019.

"All datasets reproduce similarly the specific features of wavenumber-1 and wavenumber-2 SSWs. A good agreement among reanalyses is also found for triggering mechanisms, tropospheric precursors, and surface response."
Indicated in this paper limitations related to pre-satellite era: " However, discrepancies are larger in the pre-satellite period compared to afterwards, particularly for the NCEP-NCAR reanalysis"

However we are very grateful for your comments on NCEP/NCAR reanalysis data. We will take into  account your opinion, the papers you mentioned and will use more modern reanalyses.

 (2) Use of ERA5 assimilated ozone for quantitative estimates of ozone loss: Because ERA5ozone is an assimilated products based on ingesting several datasets (including different data sets at different times), extensive validation of this product would be needed before using it to derive quantitative estimates of ozone changes, especially on the daily temporal and relatively localized (e.g., where zonal means are inappropriate) spatial scales that are important for polar stratospheric ozone loss. While doing so is a highly valuable undertaking, I am not aware of any study that has done this already.

Reply:

(3) Trajectory modeling: The initialization of the trajectory model on a single latitude circle makes all of the results highly suspect, and makes interannual comparison virtually impossible, since any latitude circle will be in different parts of the vortex at different times and especially in different years. Without relatively uniform sampling (to guarantee which one would have to initialize parcels relatively uniformly throughout the vortex, e.g., a procedure similar to that described in Manney & Lawrence, 2016, ACP), you cannot even compare results on different dates in one year, much less do fair interannual comparisons.

Reply:

(4) Chemistry-transport modeling: There is inadequate description of the details (e.g., initialization dates and fields, boundary conditions, etc.) of the set up of the model runs. Some of the interpretation of the results is unclear or inconsistent. It is not obvious that the model has been well-validated, nor that the agreement with observations shown here is adequate.

Reply:

The description of the chemical transport model and details of the numerical experiments carried out has been expanded. The interpretation of the results of model experiments has been changed and expanded.

**Clarification issues that are needed throughout:**
You should be careful about using (as you currently do even in the abstract where being precise is especially important) terms like "ozone loss", since that is usually taken to refer to chemical "loss".

Also, for the most part, dynamical factors tend to \*increase\* ozone in the lower stratosphere, so saying they contribute to chemical "loss" (or to ozone decreases to use a term that does not imply chemical loss) can be confusing.

Finally, whether and which dynamical processes contribute to decreasing ozone depends on whether you are talking about column or vertically-resolved ozone -- for example, column ozone is lower in cold regions because of the direct impact of lower temperatures on density at a given pressure, and this can be a substantial portion of the appearance of very lower column ozone values in the coldest portions of the vortex; in many places in the paper it is not made clear which you are talking about, and in some places it is not clear how calculations of one relate to the other.

Similarly, you often use the term "ozone anomalies" when you specifically mean low ozone in the winter polar lower stratospheric vortex (or equivalently low column ozone) that is related chemical loss. The are / can be many other kinds of "ozone anomalies", including winter/spring seasons (such as 2015 in the Arctic) with anomalously high ozone, as well as other kinds of low ozone anomalies (such as "mini-holes" in column ozone, which are entirely dynamical in origin and typically appear outside the polar vortex, but often at high latitudes near the vortex in winter).

If you are going to use the term anomaly, you should define exactly what it is an anomaly from; however, it appears to me the way you use it means unusually low ozone relative to climatology that arises at least partially from chemical loss -- if that is the case,

I would suggest using different terminology that is more precise. (E.g., page 2, line 30, instead of"...significant ozone anomalies are observed in the Arctic less…" it would be clearer to say something like "...extensive chemical ozone depletion occurs less often in the Arctic than in…";
on line 34, for the Antarctic, it makes sense to simply say something like "the Antarctic ozone hole was one of the deepest / most extensive on record…")

**Specific Comments**
"(Where I suggest references, I have tried to provide the DOIs if they are not already cited in this manuscript.)"

| | | Reply |
|---|---|---|
| 1. | **Introduction**, overall: While I'm providing a number of comments below about particular statements and the literature cited for them in the introduction, I question whether this detailed a review of well-known impacts of stratospheric ozone loss is needed or appropriate for this paper. For example, possible (though as yet still controversial) effects on precipitation or weather seem as best peripheral to this paper. I believe much of the material that is not directly related to setting the context for interannual variability and interhemispheric variability in stratospheric vortex dynamical and chemical conditions and chemical ozone loss could / should be condensed or deleted. | According to Review 1 & 2 our Introduction was shortened from three to two pages. |
| 2. | **Page 1, lines 28-29:** This is one of many places where there is a very incomplete list of references, some of which are not the most appropriate ones. In cases like this where it is a general, well-known point, adding a recent review paper (such as Domeisen et al 2019, https://doi.org/10.1029/2019JD030923 in this case) or at least simply adding "e.g.," before or "and references therein" after would convey the information that these are only examples of some of the literature on the subject. Simply adding "e.g.," beforehand would probably be sufficient in this case. | + Domeisen, D.; Butler, A.; Charlton-Perez, A.; Ayarzagena, B.; Baldwin, M.; Dunn-Sigouin, E., Furtado, J.; Garfinkel, C.; Hitchcock, P.; Karpechko, A.; Kim, H.; Knight, J.; Lang, A.; Lim, E-P.; Marshall, A.; Roff, G.; Schwartz, C.; Simpson, I.; Son, S-W.; Taguchi, M. The role of the stratosphere in subseasonal to seasonal prediction: 1. Predictability of the stratosphere. J. Geop. Res., Atm., 125, e2019JD030920. https://doi.org/10.1029/2019JD030920, 2020.

+ Domeisen, D.; Butler, A.; Charlton-Perez, A.; Ayarzagüena B., Baldwin, M.;Dunn-Sigouin, E.; Furtado, J.; Garfinkel, C.; Hitchcock, P.; Karpechko, A.; Kim, H.; Knight, J.; Lang, A.; Lim, E-P.; Marshall, A.; Roff, G.; Schwartz, C.; Simpson, I.; Son, S-W.; Taguchi, M. The Role of the Stratosphere in Subseasonal to Seasonal Prediction: 2. Predictability Arising From Stratosphere -Troposphere Coupling. J. Geophys. Res., Atm., 125, https://doi.org/10.1029/2019JD030923, 2020. |

| | | |
|---|---|---|
| 3. | **Page 2, lines 30-31**: Smyshlyaev et al (2016) is not a key reference here, I would suggest some earlier papers that were among the first to focus on disentangling chemical and dynamical effects on ozone (e.g., Manney et al, 1995, JAS, https://doi.org/10.1175/1520-0469(1995)052%3C3069:LTCUDP%3E2.0.CO;2; Manney et al 2011, Nature -- especially the SI for details on chemical and dynamical effects on column ozone -- and references in the latter). WMO reports are always good references, in this case the 2006 one has a particular detailed section on diagnosing chemical and dynamical effects on column ozone. This is a case where "and references therein" is definitely appropriate. | References have been added.
Manney, G. L. et al. Lagrangian transport calculations using UARS data. Part II: ozone. J. Atmos. Sci., 52, 3069–3081, 1995.
Manney, G.; Santee, M.; Rex, M.;Livesey, N.; Pitts, M.;Veefkind, P.; Nash, E.;Wohltmann, I.; Lehmann, R.;Froidevaux, L.; Poole, L.;Schoeberl, M.;Haffner, D.; Davies, J.;Dorokhov, V.;Gernandt, H.; Johnson, B.;Kivi, R.;Kyrö, E.; Larsen, N.;Levelt, P.;Makshtas, A.; McElroy, C.; Nakajima, H.;Parrondo, M.;Tarasick, D.; von der Gathen, P.; Walker, K.; Zinoviev, N.Unprecedented Arctic ozone loss in 2011. — Nature, 478, pp. 469–475, https://doi.org/10.1038/nature10556, 2011 |
| 4. | **Page 2, line 33**: As you note on line 36, there was also a strong SSW (arguably stronger in terms of abrupt changes than that in 2002) in the SH in 2019; Wargan et al (2020, JGR) should be cited in both places for that; and it should be mentioned with the 2002 one (reorganizing this paragraph to talk about them together would be helpful. Solomon et al (2014) is not an
appropriate reference for the 2002 SH SSW -- the most appropriate ones would probably be Allen et al (2003, GRL, doi:10.1029/2003GL017117) and/or Hoppel et al (2003, GRL, doi:10.1029/2003GL016899) -- the first peer-reviewed papers on that SSW - and Shepherd et al (2005, JAS - the preface to the special issue on that SSW). | Take into account obtained early Review 1 we delete everything on Antarctic stratosphere and SSW in 2002 |
| 5. | **Page 2, line 35:** More appropriate references for the depth of the 2015 Antarctic ozone hole would be Ivy et al. (2017, GRL, doi:10.1002/2016GL071925), Stone et al. (2017, JGR, https://doi.org/10.1002/2017JD026987), and/or the 2018 WMO report. | Take into account obtained early Review 1 we delete everything on Antarctic stratosphere and SSW in 2002 |
| 6. | **Page 2, line 37**: Need to specify whether by "largest decrease" (should be "decreases") you mean in vertically-resolved or column ozone. | This paragraph was significantly reduced. |
| 7. | **Page 2, line 41:** Should cite Bernhard et al (2013, ACP, https://doi.org/10.5194/acp-13-10573-2013) for anomalously high surface UVI in 2011. | Done:
Bernhard, G.; Dahlback, A.; Fioletov, V.; Heikkilä, A.; Johnsen, B.; T. Koskela, T.; Lakkala, K.; Svendby, T. High levels of ultraviolet radiation observed by ground-based instruments below the 2011 Arctic ozone hole. Atmos. Chem. Phys., 13, 10573–10590, 2013
https://doi.org/10.5194/acp-13-10573-2013, 2013. |
| 8. | **Page 2, line 45**: These results of Chubarova et al (2020) are questionable, given that the three methods used in that paper to estimate UV trends resulting from changes in cloudiness and ozone agree very poorly (their Figure 13). | We delete this reference |
| 9. | **Page 2, line 52:**Manney& Lawrence (2016, ACP, cited elsewhere in this manuscript), should be cited here. | Done |
| 10. | **Page 2, lines 61-62**: Need references for this sentences; Lawrence et al (2020) is good for the temperatures (also several other papers in the JGR/GRL special issue on the 2019/2020 Arctic vortex, including Wohltmann et al, 2020, which you cite elsewhere); DeLand et al (2020) should be cited here (as well as where you do later on) since it discussed observed PSC activity. | Done |
| 11. | **Page 3, line 65**: Reference should be Dameris et al (2021). Other published papers that discuss the low column ozone and diagnose its | Dameris et al (2021) - done
Wohltmann et al (2020) and Inness et al (2020) are |

| | | |
|---|---|---|
| | chemical origins should be cited here, including Wohltmann et al (2020), Inness et al (2020, JGR), and others from the aforementioned special issue. | also cited. |
| 12. | **Page 3, line 77**: The fact that it was exceptionally long-lived, which you don't mention, was also critical (e.g., Manney et al, 2020; others). Because the results in the paragraph this sentence ends are all from published papers, I believe this should be greatly condensed with appropriate references to those papers. | We include this point "... exceptionally long-live stratospheric polar vortex (e.g., Manney et al, 2020)" |
| 13. | **Page 3, line 82:** If you are going to discuss "the El Nino-South [sic] Oscillation effect", you need to define what that is. A reference to the review by Domeisen et al (2019, Rev. Geophys, https://doi.org/10.1029/2018RG000596) could be helpful. However, I am not sure that this paragraph contains any information that is necessary / directly relevant to the current manuscript, since you do not analyze any relationships to these SST patterns. | The reference Domeisen et al (2019, Rev. Geophys,) was included. Paragraph aims to describe published results on possible cause of wave activity propagation to stratosphere weakening. However, we are ready to delete it if necessary. |
| 14. | **Page 3, lines 86-92:** This discussion of the early 2019 major SSW is peripheral to this paper and does not add anything. Further, if it is included, the radiative / dynamical interactions leading to a slow recovery after many strong, early-season SSWs should be discussed (e.g., as in Hitchcock and Shepherd, 2013, JAS, DOI: 10.1175/JAS-D-12-0111.1). | We delete two sentences on Arctic SSW 2019 |
| 15. | **Page 3, line 95:** Add "e.g.," before Rex et al reference, since there are numerous papers on this. | Done |
| 16. | **Page 4, lines 101-110:** This discussion could be condensed to a sentence with appropriate references, or deleted entirely. However, taking this as it is: Saying there were "regular...ozone holes in Antarctica" by "the end of the twentieth century" is misleading given that there were annual ozone holes by the early 1980s. Even more importantly, there has not been anything that could be unequivocally called an "ozone hole" in the Arctic, even through 2020 (see, e.g.,Solomon et al, 2014; Wohltmann et al, 2020; and the online discussion for Dameris et al, 2021). If you are going to talk about the impacts of the Montreal protocol, it would be best to cite some of the several "World Avoided" papers that address this topic in detail (e.g., Newman et a, 2009, ACP, https://doi.org/10.5194/acp-9-2113-2009; Chipperfield et al, 2015, Nature Comms, DOI:10.1038/ncomms8233). | We delete everything on Antarctica ozone anomaly and add the reference on Newman et a, 2009, ACP, with "e.g." |
| 17. | **Page 4, lines 110--114:** There are numerous other references so at least add an "e.g.," before Weber et al. Also, if you cite Ball et al (2018) it is also important to cite some following papers that update and / or call those results into question (e.g., Wargan et al, 2018, GRL, https://doi.org/10.1029/2018GL077406; Chipperfield et al, 2018, GRL, https://doi.org/10.1029/2018GL078071; Ball et al, 2019, ACP,https://doi.org/10.5194/acp-19-12731-2019, 2019). | e.g.," before Weber et al. - Done Take into account obtained from Reviewers suggestions to reduce Introduction we remove the sentence on Ball et al., 2018 |
| 18. | **Page 4, lines 115-124:** Again, this paragraph could be greatly reduced or deleted. Also: on line 116, saying the 2019 ozone hole was "lowest" is very confusing, I'd suggest "shallowest" or some other such wording; line 120, neither Butler et al 2020 nor Wargan et al 2020 discuss the 2020 Antarctic ozone hole (and Butler et al 2020 is about two NH winters), so neither is an appropriate reference here. Further, the statement on lines 118-120 that the Antarctic is showing increasing interannual variability is entirely speculative and no evidence is given to back it up (two contrasting years does not make a trend). | We delete everything on Antarctica ozone anomaly |
| 19. | **Page 4, lines 126--128:** It would be good here to make a clear statement about what is new in the paper that goes beyond the papers that have already been published. | Our investigation is focused on following interesting issues of exceptional Arctic winter season 2019-2020 using several approaches: |

| | | - Numerical calculations with the chemistry transport model with dynamical parameters specified from the MERRA-2 reanalysis data, carried out according to several scenarios of accounting for the chemical destruction of ozone
- Comparison of ozone loss estimates inside the polar vortex based on trajectories analysis using ERA5 ozone data and based on ozonesonde data.
- Dynamical processes contributed to onset of the minor SSW event in late March 2020
As for our knowledge these issues were not discussed early. |
|---|---|---|
| 20. | **Page 5, lines 138--139, and Section 2.1**: It should be stated which reanalysis or reanalyses are used for each diagnostic shown in the paper. As per the major comments, need to justify using and/or showing different reanalyses for different diagnostics. A very strong justification is needed for using the NCEP/NCAR (aka NCEP-R1, or just NCEP as you call it) reanalysis, which has long been deprecated for any polar processing studies (e.g., Manney et al, 2005, MWR, https://doi.org/10.1175/MWR2926.1; Manney et al, 2005, JGR, doi:10.1029/2004JD005367; Lawrence et al, 2018, ACP, https://doi.org/10.5194/acp-18-13547-2018; and references therein). | Information on used reanalysis data was modified in Chapter 2.
We used NCEP/NCAR reanalysis to analyze large scale dynamical processes as minor SSW and related wave activity enhancement.
We supposed that such features are described comparable in most known reanalysis data including NCEP/NCAR as it was shown in
+ Ayarzagüena, B.; Palmeiro, F.;  Barriopedro, D.; Calvo N.; Langematz, U.; Shibata, K. On the representation of major stratospheric warmings in reanalyses. Atmos. Chem. Phys., 19, 9469–9484, https://doi.org/10.5194/acp-19-9469-2019, 2019.

"All datasets reproduce similarly the specific features of wavenumber-1 and wavenumber-2 SSWs. A good agreement among reanalyses is also found for triggering mechanisms, tropospheric precursors, and surface response."

Indicated in this paper limitations related  to pre-satellite era: " However, discrepancies are larger in the pre-satellite period compared to afterwards, particularly for the NCEP-NCAR reanalysis"

However we are very grateful for your comments on NCEP/NCAR reanalysis data. We will take into account your opinion, the papers you mentioned and will use more modern reanalyses. |
| 21. | **Page 5, line 140**: Throughout this subsection, need to say which reanalysis or reanalyses was used to calculate each of the diagnostics and why the same one (or, much better, more than one) wasn't used to calculate all of them. | We indicate the aims of using all used reanalysis data. |
| 22. | **Page 5, lines 141--144:** This is not a useful diagnostic since it is neither related to the polar vortex nor expected to capture the actual minimum in high-latitude temperature in all conditions.

While you may argue that this region was inside the polar vortex most of the time during 2019/2020, you cannot make that case for all of the winters you focus on, much less all Arctic winters. Even if this region was in the polar vortex, the location of minimum temperatures (which isn't always inside the polar vortex either since the cold region and vortex are often not concentric in the Arctic) varies a lot both within one season and in between years, so you are almost certainly not comparing the lowest high-latitude temperature at different times in one year or in between years. In the list of years compared, 1996-1997 stands out as being the one that had only modest chemical ozone loss (with the low column ozone in spring 1997 being largely related to dynamical effects including the direct effects of the late period of low temperatures in that winter (e.g., see | We remove these sentences and related plot |

| | | |
|---|---|---|
| | discussion of and references on 1996-1997 in the supplementary information of Manney et al, 2011, Nature). This distinction is important, particularly when discussing column ozone changes. Finally, the relationship of temperatures in the lower stratospheric vortex in 2019/2020 to the other years with the most ozone depletion, to climatology, and to those in the Antarctic winter, has already been more completely and correctly discussed (accounting for the full region of low temperatures), most completely in Lawrence et al (2020) and Wohltmann et al (2020), so a brief statement citing those papers (as well as DeLand et al, 2020 for the relationship of temperatures to PSC observations) would be quite sufficient and more accurate than including these diagnostics. | |
| 23. | **Page 5, lines 145**: Need to give some references in relation to the effects of wave propagation as diagnosed by the Plumb or other formulations of 3D EP fluxes (e.g., Nishii et al, 2011, J Clim, DOI: 10.1175/JCLI-D-10-05021.1, and references therein.) | + Nishii, K.; Nakamura, H.; Orsolini, Y. Influence on the Stratospheric Variability through Enhancement and Suppression of Upward Planetary-Wave Propagation. J. Climate, 24, DOI:https://doi.org/10.1175/JCLI-D-10-05021.1, 2011.
 + Peters et al., 2010 |
| 24. | **Page 5, lines 150-155:** It is not clear (here or later) how the discussion of this diagnostic goes beyond that in Lawrence et al (2020). Also need references on the calculation of the NAM index from geopotential height anomalies (e.g., Cohen et al, 2002, Baldwin and Thompson, 2009). If the NAM index is indeed calculated as described here (which description is consistent with the papers mentioned above and with Lawrence et al, 2020) then the range of values in the figure you show later does not appear to make sense (see comment on that figure). (Again, this is a case where it is not clear that the analysis you show of this diagnostic goes beyond or addsanything to that in Lawrence et al, 2020.) | Figure with recalculated NAM index is placed in Supplement |
| 25. | **Page 5, line 159--Page 6, line 2**: Need to say something about the validity of trajectories as long as 120 days for this purpose. Typically individual trajectories are not considered reliable (even in the lower stratosphere where radiative time scales are long) for more than a couple of weeks, so lengthy trajectories are used only to diagnose large scale motions by using very large ensembles of parcels. It is not at all clear that this purpose -- because you interpolate ozone to individual locations, thus assuming that those locations are relatively precise -- is consistent with that type of usage. Also, as mentioned elsewhere, because the ERA5 ozone is an assimilated product based on combining numerous datasets, one would need to either cite or perform detailed validation before its usage could be considered appropriate for this quantitative usage. | Not individual but ensembles of trajectories are used. We recalculated trajectories in backward and forward direction to validate the results.
 At present the backward trajectories were initiated at the 400K, 435K, 460K and 500K levels at the latitude circles between $70^0$ N and $85^0$ N with $0.5^0$ resolution in latitude and $0.5^0$/cos(latitude) spacing in longitude. The dates for the trajectory initiation were chosen when that domain was mainly located inside the polar vortex. For winter 2019-2020 it was April 10, for winter 2010-2011 it was March 26 and for winter 2015-2016 it was February 26 due to the early SSW. The end date of all trajectories is December 1. Trajectories were calculated using ERA5 vertical wind.
 The vortex interior was determined as the region with PV>14 PVU at 400 K, PV>26 PVU at 435 K, PV>36 PVU at 460 K and PV>46 PVU at 500 K, this criterion was used to filter out the initial locations outside the vortex and the trajectories leaving the vortex later on.
 Forward trajectories were initiated in the similar way in December. Results of comparison of backward (460K) and forward (500K) simulations are presented in Supplement, but not included into the text of paper. The ozone behavior is almost the same (as well as temperature) and also not sensitive to the date of trajectories initialization.
 The validity and advantages of using ERA5 data for Lagrangian studies are presented in (Hoffmann, L., Günther, G., Li, D., Stein, O., Wu, X., Griessbach, S., Heng, Y., Konopka, P., Müller, R., Vogel, B., and |

| | | Wright, J. S.: From ERA-Interim to ERA5: the considerable impact of ECMWF's next-generation reanalysis on Lagrangian transport simulations, Atmos. Chem. Phys., 19, 3097–3124, https://doi.org/10.5194/acp-19-3097-2019, 2019) Regarding ERA5 ozone (Hersbach, H, et al., The ERA5 global reanalysis. Q J R Meteorol Soc., 146, 1999– 2049. ttps://doi.org/10.1002/qj.3803, 2020.) discussed some improvements in ERA5 ozone data compared to ERA-Interim. The heterogeneous ozone chemistry was updated, and a number of changes were introduced in the assimilation system. (Added to the text of paper) Our comparisons of results of trajectory analysis and ozone losses based on ozonesondes also can be considered as ERA5 ozone validation. |
|---|---|---|
| 26. | **Page 6, Section 2.3:** How is "inside" the vortex determined (and which reanalysis is used to do that)? How far inside must the data be for the "well isolated" assumption to be valid? There are many more complete references for effects of descent on ozone in the vortex than Braathen et al (1994; note that you have a typo in that citation), e.g., Tegtmeier et al, 2008, GRL, https://doi.org/10.1029/2008GL034250, as well as numerous other papers cited in the WMO reports (again, the 2006 report has a special focus on distinguishing chemical and dynamical effect in the Arctic vortex). How are the descent calculations done? To be robust, they would have to follow the motion of the air sampled at the time and location of each ozonesondemeasurement, since descent is by no means uniform throughout the vortex. If you are calculating a vortex-averaged descent rate that is used with vortex-averaged ozone, for that to be even roughly an accurate estimate, you would have to demonstrate that you have uniform and consistent coverage of the vortex in both the ozone profiles that go into the average and the diabatic descent (which in the latter case includes demonstrating that the descent rate is a reasonable approximation of all the descent conditions the parcels in the ozone measurements experienced). Why are temperatures from JRA (55 presumably) used with an offline radiation code instead of using diabatic heating rates provides with the reanalyses (ERA5, MERRA-2, and JRA-55 all provide these, and it would seem to make more sense to take those from whichever of these reanalyses you use to determine vortex characteristics for the sonde analysis)? | The vortex interior was determined as the region with PV>42 PVU at 475 K vertical level using JRA reanalysis, as it was indicated in section 3.2. Now this information is moved in section 2.3.. "Well isolated" is now changed to "isolated" and it assumes negligible transfer across the vortex edge (that is shown by the results of trajectory calculations). The reference Braathen et al (1994) concern the description of calculation method as a whole, not the effects of descent on ozone in the vortex only. Detailed justification of the method is given in Lucic et al (1999). We included this work in references. We used vortex-averaged values for calculations. Concerning diabatic descent: uniform and consistent coverage of the vortex area is provided since the descent rates inside the vortex were calculated daily at grid points with resolution 2.5° × 2.5° and than averaged over the vortex area. Worse with ozone profiles, but we use 7-days running average values to improve uniformity (during this time period the influence of diabatic descent on ozone profiles is negligible). Besides, calculations of both forward and backward trajectories shows that the ozone behavior in both directions is the same, proving that the vortex air masses are well mixed since the backward and forward trajectories have the different pathways. Unfortunately, we didn't find diabatic heating rates in public available reanalyses JRA, ERA5 and MERRA. |
| 27. | **Page 6, line 184**: This seems to be very coarse resolution, especially in the vertical. What is the actual vertical resolution in the lower stratosphere where you are focusing on the results? Is this adequate to capture the expected vertical variations / motion? | One of the goals of our work was to demonstrate the possibility of using a model with a rather coarse spatial resolution to analyze general and local features of changes in the ozone content in the Arctic. A pretty good qualitative agreement between the results of modeling and measurements makes it possible to analyze the influence of dynamical and chemical factors on the variability of Arctic ozone. |
| 28. | **Page 6, lines 186--187**: I am no expert on this, but I would like to see some justification of why it is appropriate / adequate to treat PSC formation as STS during a winter such as 2019/2020 when temperatures were low enough that larger solid HNO3 containing particles were present (and even at some time ice PSCs). | Our scheme of the formation and evolution of PSC s takes into account STS, NAT and ice. In the modified version of the article, we have provided a more detailed description of the heterogeneous block of the model. More details are given in our publications (DeZafra and Smyshlyaev, 2001, Sovde et al., 2008 and Smyshlyaev et al., 2010). |

| | | |
|---|---|---|
| 29. | **Page 7, lines 194--195: Why north of 64N?** This does not encompass the entire vortex, nor the entire region of lowest temperature, except perhaps on some individual days when the vortex is unusually pole-centered -- thus it does not encompass the full region in which PSCs might be expected. | We don't resolve vortex position in the CTM. Its edges are automatically defined in the reanalysis data. The idea of the noCHEMall model run was to switch off chemistry in the entire Arctic region for the winter-spring period. In the updated version of the model we switched of chemistry north to 60 degrees for the period from November 1 till May 15. |
| 30. | **Page 7, lines 202--210**: This has been covered more completely in already published papers including Bernhard et al (2020), Inness et al (2020), and Dameris et al (2021). Simply describing this briefly with citations of those (and potentially other) papers would be more appropriate than presenting this as if it were a new result. | References to already published papers have been included. |
| 31. | **Page 8, lines 226:** More like the last approximately 60 years, see Lawrence et al (2020) and Matthias et al (2016, GRL, doi:10.1002/2016GL071676). | Modified |
| 32. | **Page 9, lines 229-237:** As mentioned in relation to the methods section, this has been covered more completely and precisely in Lawrence et al (2020), Wohltmann et al (2020) and others. It would be more appropriate to include a brief statement citing these papers rather than presenting this as if it contained new results. | References on Wohltmann et al., 2020, and Lawrence et al., 2020 were added. |
| 33. | **Page 9, lines 242--243:** As mentioned already, 1996-1997 did not have severe ozone loss.
Moreover, not only did 1996-1997 not have a strong polar vortex (which is by no means synonymous with a cold one) but rather an exceptionally weak one until spring, but also 2004--2005 was notable for being cold and having substantial ozone loss, but having a rather weak vortex that allowed considerable mixing (e.g., Manney et al, 2006, GRL, doi:10.1029/2005GL024494; Schoeberl et al, 2006, JGR, https://doi.org/10.1029/2006JD007134; Lawrence et al, 2020). | After 1st Review we moved this plot to Supplement, |
| 34. | **Page 9, lines 246--247:** What are the implications of this? And what does this add to what has already been shown by Lawrence et al (2020)? | We want to highlight the period with strongly reduced upward wave propagation from approximately February 7 to March 7 that was followed by enhanced upward propagation (related to the breaking event over Gulf of Alaska and minor SSW in late March |
| 35. | **Page 10, lines 256--261**: What is new here that goes beyond what was shown by Lawrence et al (2020)? | Lawrence et al (2020) showed a positive surface temperature anomalies over Northern Eurasia as Jan-March mean (their Fig.6a). However they did not mention it as a possible cause of reduced upward wave activity propagation.
We speculate on possible link of positive temperature anomalies over North-Eastern Eurasia and reduced upward wave activity propagation. We found a negative correlation between near surface temperature anomalies over for zones: 1) 40-60° N & 90-120° E; 2) 50-70° N & 90-120° E; 3) 50-70° N & 80-120° E; 4) 50-80° N & 90-120° Eand 50-80N mean heat flux on 70 hPa smoothed by 7-day running mean over the period from November 1, 2019 to the SSW event onset in the middle of March 2020. Following correlation coefficients were obtained for mentioned above zones respectively: -0.49, -0.54, -0.57, -0.58. We did not include it in our paper as it is beyond the main focus of our study and needs additional investigation. |
| 36. | **Page 11, lines 265--286**: Again, it is not obvious what this adds to what has already been shown / discussed by Lawrence et al (2020). Also, Figure 4 shows NAM index values up to about 10, whereas | Figure with recalculated NAM index was placed in Supplement |

| | | |
|---|---|---|
| | Lawrence et al (2020) show values near 5 at the same time and place (their Figure 4a); the values shown by Lawrence et al are typical of those shown in previous work calculating that index based on GPH anomalies. Yet your description of your calculation in the methods section sounds like it is the same index used in these previous papers. Please explain this apparent inconsistency. | |
| 37. | **Page 12, lines 299--301 (Figure 4 caption):** How significant are the differences in "Plumb" fluxes in (b) from climatology, compared to those during similar length time periods in other individual years or at other times in 2019/2020? That is, how unusual is this behavior? | Plots with "Plumb" fluxes were moved in Supplement as we did not find significant difference between selected period of 2020 and climatology. |
| 38. | **Page 12, line 306 to Page 13, line 311**: The "SSW event" you describe was very minor and affected only the upper stratosphere. Although it could be the case that it resulted directly from the enhanced upward wave propagation, you have shown nothing to demonstrate this. You have also shown no evidence that a Rossby wave breaking event occurred in the troposphere nor that if it did, it was associated with the enhancement of wave activity. You have not shown potential vorticity at all, so the reader cannot know if / where it was low. The situation described by Coy et al (2009) was in relation to a major SSW that affected the entire stratosphere for weeks -- there is no reason to believe that the very brief minor event in the upper stratosphere in 2020 that you describe is analogous in any way. (In addition, it is not clear in any of the accompanying discussion, why this minor event that showed no evidence of significantly affecting the lower stratosphere is relevant to the analysis in this manuscript.) | 1) We agree that this SSW event was minor. We think that influence of this SSW event (as well as previous one in the beginning of February) was not negligible for the Arctic lower stratosphere temperature because:
- the lower stratosphere minimum temperature over 50-90 N increased up to the threshold of PSC NAT formation in the second half of March https://ozonewatch.gsfc.nasa.gov/meteorology/figures/merra2/temperature/tminn_70_2019_merra2.pdf (our former and removed Fig.2 with Tmin on 70hPa inside 70-90N shows it too)
- PSC NAT volume reduced from ~50 mln $km^3$ to nearly zero in the second half of March https://ozonewatch.gsfc.nasa.gov/meteorology/figures/merra2/temperature/natvn_2019_merra2.pdf
- According to MLS data ClO dropped to nearly zero values in the second half of March as it shown on Fig.2f in Manney et al., (2020).
Certainly the other important factor (and possibly dominant) was an influence of increasing radiative heating due to seasonal cycle.

2) Rossby wave breaking event (RWB)
We suppose that this RWB event was related to enhancement of upward wave activity propagation and following minor SSW because:
- Upward enhanced propagation of wave activity illustrated by Plumb vertical component Fz during the onset of the minor SSW event in the middle of March was observed only over the region of Gulf of Alaska - north-western Canada (Fig.4a)
- Related to RWB event anticyclone nearby 160W enhanced and reached the lower stratosphere (Fig.5d)
- 3) PV plots on 100 hPa using ERA5 reanalysis data on pressure levels for 00Z (Fig. S4) are included in Supplement.
The lower PV incursion over the Gulf of Alaska - north-western Northern America is seen in the middle of March. |
| 39. | **Page 15, line 306 (Figure 6 caption):** Why 50-70N? | As we focus on influence of upward wave activity propagation over the region of Gulf of Alaska - noth-western Canada the 50-70 N cross-sections were presented. They illustrate investigated features better than 40-60 N ones. |
| 40. | **Page 15, lines 331--344:** Please see major comment (3), as well as previous comments on inappropriate initialization locations, need to justify the length of the trajectories for this purpose, and the need to demonstrate (or cite literature that did so) that ERA5 assimilated ozone is appropriate for this purpose. Also, choosing a different initialization date in each year apparently just because that latitude circle happened to be within the vortex is further degrading the ability | See reply for Page 5, line 159--Page 6, line 2
The text of the paper was changed |

| | | |
|---|---|---|
| | to make interannual comparisons and the dependence of the results on details of vortex shape, position, and evolution. Again, you do not say how you determine what is inside the vortex. | |
| 41. | **Page 17, lines 358--359**: Do you mean you ran the model with ERA-Interim, or you ran the model with ERA5 degraded to ERA-Interim-like resolution? How large are the differences? If they are large enough so as to make the results highly uncertain,then this points to the need to do something more (typically driving the model with several different reanalyses) to determine whether the results can be considered robust at all. There are numerous papers (eg, in the S-RIP special issue of ACP/ESSD, https://acp.copernicus.org/articles/special_issue829.html) that show substantial differences in results of trajectory analysis and / or chemistry / transort model results from using different reanalyses to drive them. (Just because ERA5 is the newest, does not mean you can automatically assume without testing that it is better for all types of analyses.) | We removed these sentences since the backward and forward simulations gave the different descending rates. |
| 42. | **Page 17, line 363:** How do you determine where the vortex centre is? | In this case we mean region with high PV values (50-60 PVU at 475 K level). |
| 43. | **Page 17, line 364:** How did you select this PV value? Is this what you use to define the vortex edge previously in the paper, and, if so, what values did you use for the other levels that are shown / discussed? Is the same value appropriate for each of the reanalyses that you use? | The PV values to define the vortex edge on different levels were taken from source code *pvpick*(available in /nadir/scr/nongraph/meteorol). For ozone loss estimation based on ozone sonde measurements we used only JRA reanalyses. |
| 44. | **Page 17, line 3(2)66:** See previous comments regarding how representative an average of measurements at a small number of stations is of the entire vortex. | See reply to page 6 above |
| 45. | **Page 17, line 370**: "ozone losses were the lowest on record" -- you must mean "ozone values were the lowest on record" or "ozone losses were the largest on record". | We changed the whole paragraph. |
| 46. | **Page 17, line 372:** There are quite a few other papers in addition to Manney et al (2020) that also show this, including Wohltmann et al (2020). In addition if on line 370 you were implying that chemical ozone loss was larger in 2020 than in 2011, then it is not really consistent with those papers, since they estimate chemical ozone loss amounts to be very similar in 2020 and 2011 (but indeed peaking at lower altitudes in 2020). | We mean that largest ozone loss occurred in 2020 at lower altitudes than in 2011 and the values of cumulative chemical ozone loss at 450 K level consistent with Manney et al. But we obtained smaller values in 2010-11 winter. We corrected this statement in the paper. |
| 47. | **Page 17, lines 377--378**: Why these two winters? What was 2004-2005 "much colder" than (your wording could be interpreted as saying it is colder than 2019-2020, which obviously is not the case)? Since there are several studies (e.g., Kuttipurrath et al, 2010, ACP, https://doi.org/10.5194/acp-10-9915-2010; Livesey et al., 2015, ACP, https://doi.org/10.5194/acp-15-9945-2015) that provide chemical ozone loss estimates for a wide range of years in the past decades, why not compare with all of them. Especially, why not compare with 2011 since it was the previous year that unequivocally had the largest ozone loss? | Here we mean that winter 2004-2005 was much colder compared with winter 2002-2003. It's not a comparison with previous years. It's just a reference on our previous published results obtained by the described method. In present paper we compare only three the most cold winters: 2010-11, 2015-16, 2019-20 with largest ozone loss. We removed this statement from the paper |
| 48. | **Page 17, line 380**, the terminology "ozone anomalies" is imprecise and potentially misleading, since you are not talking about just any ozone anomaly (which could occur anytime or anywhere for many different reasons), but specifically anomalously low ozone in lower stratospheric vortex that is related to unusually cold conditions there and thus partly to chemical ozone loss. I recommend not using this wording (here or elsewhere in the paper). | Terminology "ozone anomalies" has been replaced by "ozone loss", "ozone decrease" and "low ozone" . |
| 49. | **Page 17, line 381:** There needs to be more information given on the | CTM description has been extended. Details of |

| | chemistry transport model and its set up (though this would probably be best in the Methods section). How is the model initialized (especially ozone and other important trace gas fields)? What are the boundary conditions for trace gases? What is(are) the initialization date(s)? Why is MERRA-2 used to drive this model whereas ERA5 is used to drive the trajectory model? | numerical experiments are defined. |
|---|---|---|
| 50. | Page 19, 393--395: This doesn't look to me like it agrees very well with the OMI data shown earlier. The minimum values in Figure 9 appear much higher than those in Figure 1 on the first three days shown; there appear to be significant differences in morphology, especially, the OMI data on the second day shown suggest that the lowest column values are actually within the polar night, whereas your model results show them to be well away from there, with no low values immediately surrounding the region OMI cannot observe. | Results of model experiments were updated in the revised version of the manuscript. |
| 51. | **Page 19, lines 398--399:** There are, in fact, satellite measurements in 2019/2020 that observe in darkness (e.g., MLS), though you would have to compare profiles rather than column (but profile comparisons are a necessary part of validating model results. But in any case, per my immediately previous comment, since the model and OMI observations appear to agree very poorly (in morphology as well as in values) going into the polar night, I fail to see how you can argue that your model results provide useful information in polar night. | In this article, we used OMI data for winter 2019-2020 and SBUV data for climatology 1979-2019. For the development of this study, we plan to use the data of the MLS, as well as the Russian Infrared Fourier Spectrometer (IKFS-2) on the Meteor-M satellite board. |
| 52. | **Page 19, lines 399--404:** This does not make sense to me. First, the OMI data show minimum values abutting the gap in polar night -- meaning the actual minima are inside the polar night. Second, the results of the chemical reactions are transported throughout the vortex, so there is no reason to expect ozone to be lowest in the sunlit regions. Third, in order to interpret these and understand how direct dynamical effects of the low temperatures may be involved we need to know where the vortex is and where the cold region (which is not necessarily concentric with the vortex) is. | This statement has been deleted from the manuscript. |
| 53. | **Page 19, lines 408--412:** How well do these results agree with the PSC observations described in DeLand et al (2020)? And / or with other PSC observations? | We plan to compare the calculated and measured characteristics of the PSO in future work. |
| 54. | **Page 19, lines 413--416**: As per two comments above, the region of PSCs and that of chemical ozone loss are not expected to coincide since chemically processing air is rapidly transported throughout the vortex and does not remain only in the region with PSCs. | Corrected. |
| 55. | **Page 20, Figure 10:** It would be helpful to know where the polar vortex is, and where the cold region is to interpret this figure. | We have provided maps of potential vorticity and temperature in the Supplement to the updated version of the manuscript. |
| 56. | **Page 22, lines 427--428**: How this is calculated should be explained further (probably in the Methods section), and something should be said about how the processes included compare with those in well-validated models, as well as how the chemistry in this model was validated. | Explanation of the chemistry validation has been added to the section Methods. |
| 57. | **Page 22, lines 428--429:** Why would the maximum rate of ozone destruction be at the edge of polar night? I would think it would be wherever chlorine is activated (which, when fully activated, is the whole vortex) and there is the most time in sunlight. Since activated chlorine is expected to be quickly transported throughout the vortex, and since the region just outside the edge of polar night receives only a little sunlight compared to regions farther into daytime, I would think the edge of polar night would be rather low in ozone destruction? | Deleted. |
| 58. | **Page 22, lines 429--434:** Since all of these processes are going on | Deleted. |

| | | |
|---|---|---|
| | inside the polar vortex, the main effect of polar vortex position is how much of it experiences sunlight, and that is only a significant factor until early March. You have not mentioned here descent (which increases ozone in the vortex and is one of the most important dynamical processes changing ozone), nor have you mentioned the direct dynamical effects by which low temperatures are associated with low column ozone. In addition, none of your statements here have been demonstrated since you never show where the vortex is or where the cold region is. | |
| 59. | **Page 22, lines 441--449:** These lines appear to be an exact repetition of the statement in the Methods section. Delete. | Done. |
| 60. | **Page 23, line 463**: It is my understanding that at the altitudes where ozone contributes most to the column (below ~20--25 km), gas-phase chemistry is very slow (much slower than dynamical time-scales), so I don't understand how it plays such a large role? | We have added analysis for two altitude regions: below 25 and up to 25 and revealed that gas-phase chemistry is important for the upper region. |
| 61. | **Figures 12--15:** Why do you not show OMI / model differences for 2019/2020? Why do you suddenly bring in SBUV data to show a climatology rather that deriving that from OMI data for 2005 through 2019? How do OMI and SBUV data compare, are there significant differences? Comparing to a climatology rather than directly to observations in the same year seems an indirect and potentially inaccurate way of assessing what processes are needed to reproduce observed fields. Also, it would help with interpretation of the results if you could show a timeseries indicating where each station is with respect to the polar vortex during the period shown. | SBUV data cover a longer time interval than OMI data. That is why we preferred to use it for climatology. |
| 62. | **Page 27, first two paragraphs of conclusions, and line 555:** In the paper, you have not made the case that the very minor SSW event (which affected only the upper stratosphere significantly, hence the absence of any mention of it in the many papers already published on the 2020 lower stratosphere conditions and ozone loss) significantly affected the lower stratosphere. In fact, temperatures in mid-March 2020 were already rising, but did so at a much slower rate than in the vast majority of years, many of which also had only very minor SSWs during this period. While nothing rules out a small effect of this minor SSW on lower stratospheric temperatures, you haven't demonstrated that there was one either -- given the many variables that affect temperatures and the seasonal cycle, the timing of the temperature increases could have been coincidence. Thus, it is not justified that this event should feature so prominently in your conclusions. | 1$^{st}$ conclusion was modified: The enhancement of wave activity propagation over the Gulf of Alaska in the middle of March, contributed to the onset of the minor SSW event and to an increase in the Arctic stratosphere temperature. |
| 63. | **Page 28, lines 558--560:** Per previous comments, this has not been demonstrated because of the inappropriate choices for initialization of the trajectories. | Changed |
| 64. | **Page 28, lines 561--563**: This has been shown for 2020 in previous papers, with which you should compare your results. | Done |
| 65. | **Page 28, lines 564--569:** Per specific comments, this has not been demonstrated. | Done |

**Typos / Minor Points:**
For the most part, I am not including details here of improvements that would be needed in the English usage, as the revisions needed to the scientific content are sufficiently major that much of the structure of the writing will be changed. The following are thus just a few things that happened to catch my eye:

Abstract, line 15, "year" should be "years"
This sentence was modified
Page 2, line 32, "warm" should be "warmer".

Done
Page 2, line 49, "statically" should be "statistically"
Done
Page 3, line 79, delete comma
Done
Page 3, line 86, "the main SSW" is not appropriate wording, particularly since SSWs are quite common in the Arctic. Perhaps you mean "a major SSW".
Done
Page 3, lines 91-96, this sentence is very long and convoluted and nearly impossible to parse correctly.
This sentence was reduced and edited.
Page 6, line 164, "has" should be "have".
Done
Page 9, line 245 "strongest weakening" is extremely confusing, please reword.
Modified: ... by largest and lasting a month weakening of upward wave activity propagation
Page 15, line 340, saying ozone losses were "higher" could be confusing (if the reader thinks of higher ozone values), I'd suggest a wording more like, e.g., "...more ozone loss occurred…"
Done

Thank you again for taking the time to review our manuscript.

With respect,
Sergei P. Smyshlyaev,

Pavel N. Vargin,

Alexander N. Lukyanov,

Natalia D. Tsvetkova,

Maxim A. Motsakov

---

## Author Comment (AC3)

Dear Referee,

Thank you for your comments on the paper and constructive recommendations. We have tried to follow your suggestions and have taken into account most of them. Following we mention how the manuscript has been changed according to your comments.

**General comments**
1. In my opinion, the **CTM results (Section 4)** are extremely problematic. I provide some more detailed discussion in my specific comments below. Here, I will just highlight the main points. The total ozone maps from the CTM shown in Figure 10 are in stark disagreement with the OMI total ozone maps from Figure 1. That alone puts the utility of the CTM experiments into question. Furthermore, the morphology of the ozone loss frequency maps (Figure 11) bears only a vague resemblance to the actual geometry of the polar vortex. The latter is not shown in the paper but it's easy to plot using the same reanalysis that the paper uses, MERRA-2, as I show below. It appears that areas of high loss frequencies from the CTM often fall outside of the vortex boundaries. This doesn't seem right. The way polar ozone loss works is that the most significant depletion occurs within the chemically processed airmass rich in active chlorine, i.e., within the polar vortex, not outside of it. In addition, or perhaps related to the above, the CTM experiments in Section 4 suggest that much, even most of the ozone loss occurred via gas-phase reactions involving NOx. This goes against our established understanding of polar ozone chemistry. That doesn't automatically make it wrong and, yes, if true it would be a major finding – but then it would require a lot stronger evidence than a low-resolution CTM experiment that fails to reproduce the observed evolution of total ozone distributions! As it is, this result only indicates likely problems with the CTM. See my specific comments for details.
**Reply:**
We removed the total ozone maps from the article, as well as other maps, and transferred them to the Supplement. Instead of these figures, we have added to the article figures of the vertical distribution of ozone and associated gases, as well as the destruction of ozone in chlorine and nitrogen catalytic cycles. It can be seen from these figures that at altitudes of 15-25 km, the main role in the destruction of ozone is played by its destruction in halogen cycles, and the role of nitrogen cycles is negligible. However, at altitudes of 25-40 km, significant ozone depletion is noted, in which nitrogen catalytic cycles play the main role.

2. **Section 3.1.** Almost everything in this section has already been discussed in detail elsewhere: Evolution of total ozone in Dameris et al., 2020; minimum temperatures as compared to other cold winters and climatology in Innes et al., 2020; heat fluxes, wave activity and geopotential height and related metrics as well as surface impacts in Lawrence et al., 2020. You are clearly aware of that as you cite those other studies in the paper. In principle, it's OK to have a study that confirms previously published results, especially if it uses different methods or data sets. However, I'm not convinced that this is the case here as both papers use reanalysis data and similar diagnostics. If there are any novel or otherwise valuable aspects here, please clearly state what they are and explain how they are distinct from the findings of the existing papers. If not, then I think most of this section should be eliminated. One element that may not have been discussed before (unless I missed something) is the analysis of 3-D Plumb fluxes. Currently, the discussion of Plumb fluxes is somewhat limited in the manuscript. Perhaps one way to salvage this section would be to expand this part while significantly shortening much of the preceding material.
**Reply:**
Section 3.1 was reduced: former Fig.2a (Tmin mean 70-90N at 70 hPa) was moved to the Supplement, former Fig.4a (NAM index) was removed, former Figures 4b-4c (with estimation of significance of difference between Fz (2020) and Fz(climate mean)) were moved to the Supplement. Also several plots with Plumb fluxes for periods March 18-20, March 20-22, and March 22-24 2020 were added to Supplement.

3. **The different parts of the manuscript are quite disconnected from each other**. For example, chemical ozone loss is calculated using three different methods (trajectory analysis, ozonesondes, and CTM) but no attempt is made to cross-check and reconcile the results.
**Reply:**
We made the backward trajectory calculations for other vertical levels (400 K - 500 K) in winter 2019-2020 and obtained the results showing the maximum of ozone loss at levels 435 K-460 K that correspond to estimates based on ozonesondes. The probable reason of quantitative difference is also discussed in revised text.

4. **The description of methods and data sets used in this study is insufficient**. It's not always clear which reanalysis is used for what. It's not even clear if the NCEP reanalysis mentioned in line 138 is used at all as it's not talked about anywhere else in the manuscript. I couldn't find any information about which reanalysis was used to generate Figures 2-6. Section 2.2 uses ERA5 to initialize the ozone content of air parcels used in the trajectory analysis, but no evaluation of ERA5 ozone is provided or cited (although I don't think that ERA5 ozone has been thoroughly validated yet). The

descriptions of the trajectory model and the CTM also lack detail. For example, how is vertical advection done in the CTM?

Why should we think that the very course resolution of the CTM (5° x 4°) is adequate? It is not clear why different reanalyses are chosen to drive the trajectory model (JRA) and the CTM (MERRA-2). This is not necessarily wrong, but it does require some explanation that is not provided. No justification is given for the choices made in the trajectory calculations, e.g., why the parcels were initialized at those specific locations and times. This list is not exhaustive. See my specific comments for details.

**Reply:**

In the updated version of the article, we described the methods used in more detail. In particular, we have described in detail the chemical scheme of the model and the structure of the performed numerical experiments. With regard to the rather coarse resolution of the model, we were interested in whether it was possible, using such a model, to describe the morphology of the formation of ozone mini-holes in winter-spring 2019-2020. The results of a comparison of the calculated and measured variability of the total ozone content at ozonometric stations demonstrated that this is qualitatively possible, although the quantitative differences can be quite significant.

|   | **Specific comments** | **Reply** |
|---|---|---|
| 1. | **L106**. Note that there's a considerable debate over whether the Arctic depletion events should be called "ozone holes". If you do use that term, please drop the *a*; just "*appearance of large ozone holes*" | Modified |
| 2. | **L138**. What was the NCEP reanalysis used for? I couldn't find any mention of it in the rest of the paper. | We include following in Section 2.1: The propagation of wave activity was analyzed by using the zonal mean meridional heat flux and three-dimensional Plumb flux (Plumb 1985) calculated using NCEP reanalysis data. |
| 3. | **Section 2.1**. Are these diagnostics calculated from all three reanalyses mentioned above or just one? Which one? | NCEP-R |
| 4. | **L138**. The canonical reference for ERA5 is Hersbach et al (2020). | Modified |
| 5. | **L142**. Please explain why this (somewhat narrow) latitude range was chosen. | This plot was moved to Supplement. The latitudinal range 70-90°N is rather appropriate for winter 2019-20 because vortex most of the winter was relatively undisturbed with center near the Pole. However for some winters with sever ozone reduction the polar vortex was more disturbed and shifted from the Pole. |
| 6. | **L144**. Are the climate averages from reanalyses? Which one, specifically? | NCEP-R |
| 7. | **Section 2.2**. It's hard for me to understand from this brief description how ozone loss is calculated. How were the initial parcel locations selected? Were they initialized with ERA5 ozone and then retained their ozone content (a variation of the passive tracer method)? What does "ensemble averaged" mean in this case? Please expand this section significantly. Some of this is explained in Section 3.2, but I think it belongs here. Please, also include or cite validation results of ERA5 ozone during Arctic winter/spring. It's a relatively new reanalysis and it cannot be assumed that its ozone is suitable for science, at least not without some solid evaluation. | Section 2.2 was changed |

| | | |
|---|---|---|
| 8. | **L203**. Please provide a citation for OMI, e.g., Levelt et al. (2018) | Done |
| 9. | **LL209-210**. This is incorrect: while ClOOCl photolysis requires sunlight, air parcels depleted in ozone can get advected out of the illuminated area. Even Figure 1 suggests that this is the case: the ozone minima occur near the terminator, as you say in the next paragraph. | Thus statement has been deleted from the manuscript. |
| 10. | **L226 and below**. What data are used here? One of the three reanalyses mentioned above? Which one? | NCEP-R |
| 11. | **LL226-232**. Lower-stratospheric minimum temperatures in different extreme winters were also compared by Innes et al., 2020. What does the present analysis bring to the table that is new? | This plot was moved to Supplement. |
| 12. | **LL230-247**. This analysis and most of Fig. 3 repeats the results of Lawrence et al. (2020). For example, see Figs 7 and 9 therein. If there is anything in the present analysis that isn't already in that study (I may have missed something), please indicate clearly what it is. Otherwise, I suggest eliminating this text and Fig 3a and b. | This part of the text was reduced, Fig.2a (with Tmin) was moved to the Supplement. Figure 2a displays the temporal evolution of heat flux (upward wave activity propagation) in the winter season 2019-2020. This figure is necessary to highlight the period of strongly reduced wave activity propagation from early February till early March 2020. The aim of the Figure 2b is to show the daily integrated values of zonal mean meridional heat flux at 70 hPa averaged over 45-75° N for three periods selected periods (normalized by the number of days in each period) of 2020 and the other five winters with strong polar vortex and ozone loss. |
| 13. | **LL288-290**. I don't think you really mean *assume*. Something like this shouldn't be simply assumed, it needs to be demonstrated. | Fig2c (former Fig.3c) illustrates strong positive temperature anomalies at 925 hPa. |
| 14. | **Fig. 4 b and c**. It would be better to use the same contour colors in both panels. I also suggest showing a difference plot. | These plots with estimation of difference significance were moved to Supplement |
| 15. | **LL315-317**. I think you mean Fig. 6a, not c and March 10-13 not 11-13. | Modified. |
| 16. | **L321**. The figure doesn't have panels e or f. I think it should be c and d. | Modified. |
| 17. | **Section 3.2. The trajectory analysis**. I don't find these results very convincing. Why were these isentropic surfaces and these particular dates chosen? This choice of initial points is very restrictive. To make sure that this calculation represents the average vortex ozone loss, you would have to demonstrate that the trajectories sample the entire lower portion of the vortex (e.g. theta < 550 K) uniformly throughout the chemically processed vortex air and uniformly in time. Without that it's hard to say what Figure 7 shows other than chemical loss along some arbitrarily chosen trajectories, that could be different if the trajectories were initialized differently. I would imagine that a good strategy | Section 3.2 and figure 5 were changed. In revised version the backward trajectories were initiated at the 400K, 435K, 460K and 500K levels at the latitude circles between $70^0$ N and $85^0$ N with $0.5^0$ resolution in latitude and $0.5^0$/cos(latitude) spacing in longitude. The dates for the |

| | | |
|---|---|---|
| | might be to select initial points randomly within the vortex and at different times, but I can be convinced otherwise if you can show that your method does provide sufficiently uniform sampling and that the choice to start all the trajectories at the same time doesn't lead to most of them missing layers and times with particularly strong or particularly weak ozone depletion. I'm thinking of a situation when all the parcels initialized at 475 K in December were below 400 K by the end of February before serious depletion (maximized above 400 K, see
Manney et al. 2020) even started. Another point: by selecting the initial parcel locations in a more robust way you could estimate ozone loss as a function of altitude / potential temperature and compare the result to that obtained from the ozonesondes analysis, thus providing some cross validation. | trajectory initiation were chosen when that domain was mainly located inside the polar vortex. Also we recalculated both forward and backward trajectories. Results are presented in the supplement. The ozone behavior in both directions is almost the same for all winters, proving that the vortex air is well mixed. |
| 18. | **L338**. This is area-weighted average, correct? Please state that clearly if true. Also, I think the few trajectories that did venture out of the vortex should be excluded from the average. | In revised version the vortex interior was determined as the region with PV>14 PVU at 400 K, PV>26 PVU at 435 K, PV>36 PVU at 460 K and PV>46 PVU at 500 K, this criterion was used to filter out the initial locations outside the vortex and the trajectories leaving the vortex later on. |
| 19. | **LL 341-342**. The fact that these rather large oscillations are there indicates that a good number of trajectories left the offspring vortices after the final SSW. It's hard to imagine what chemical mechanism would produce ~0.8 ppmv up and down changes over the course of a few days in the lower stratosphere! | In revised version we initiated the backward trajectories before SSW. The figure was changed. |
| 20. | **Figure 7**. Please replace the commas in the y-axis tick labels with decimal points. | Done |
| 21. | **LL364-365**. (1) This logic seems backward: One should select those observations that are inside the vortex, not handcraft the definition so that it accommodates the sonde locations. (2) how is the vortex edge defined at levels other than 475 K? Polar vortexes exhibit complex 3-D geometries with edges at different levels often not lining up. Some of those stations were definitely outside of the polar vortex for some period of time (e.g., Ny-Ålesund and Sodankylä in mid-March). Were these measurements excluded from the regression analysis? | We excluded from analysis ozone data for the dates when this or that station was outside the vortex. We used only ozone observations inside the vortex. The vortex edge was defined as 42 PVU at 475 K isentropic surface. The PV limits for other levels were taken from source code *pvpick* (available in /nadir/scr/nongraph/meteorol. |
| 22. | **L376**. The maximum loss of 3 ppmv shown in Fig 8 occurs at 450 K. In Wohltmann et al. (2020) the vortex average (mentally subtracting the lines in their Fig. 4b) is about 2 ppmv at that level. Is it possible that this discrepancy results from different definitions of the vortex edge? | Certainly most of ozone data we used were obtained in the most depleted parts of the vortex. Unfortunately many sonde flights near the vortex edge terminated at quite low height. But on the other hand our results are consistent well with results of (Manney et al., 2020) -2.8 ppmv cumulative chemical ozone loss on 460 K level.
Manney, G.; Livesey, N.; Santee, |

| | | M.; Lawrence, Z.; Lambert A.; Millan, L.; Fuller, R. Record low Arctic stratospheric ozone in 2020: MLS polar processing observations compared with 2016 and 2011. *Geop. Res. Lett.* 2020 |
|---|---|---|
| 23. | **L383**. If I understand correctly the CTM used water vapor from MERRA-2? If that is true, then it could seriously skew the results. Note that MERRA-2 **stratospheric** humidity is not very much informed by observations. It is, instead, relaxed to a zonally symmetric climatology (3-day relaxation time). It is therefore, especially suspect in extreme situations such as the 2020 winter/spring. Stratospheric water vapor in reanalyses is generally not recommended for scientific use with some exceptions (see Davis et al., 2017). | The water vapor content was specified according to the reanalysis data only for the troposphere, while in the stratosphere it was calculated taking into account chemical reactions and transport. In the updated version of this article, this is explained in the Methods section. |
| 24. | **LL392-404 and Fig. 9**. I'm sorry but I don't think the model compares well to observations at all. Below I juxtaposed the OMI total ozone map from Fig 1 (left) and the CTM ozone from Fig 9 (right) for March 15. This is a particularly striking example, but things don't look much better on the other days considered, except perhaps on 3 March. One would expect at least the dynamical features to line up as the CTM is driven by reanalysis winds – but they don't (compare the shape of the total ozone contours and their gradients). Then looking at the ozone values the two figures have almost nothing in common: OMI shows a large complex patch of deeply low values extending between the Hudson Bay and northern Siberia while the CTM has a single weak minimum over the coast of Alaska, where OMI does not show anything noteworthy. Much weaker gradients in the CTM suggest that the model may produce far too much mixing across the vortex edge. The overall positive bias at high latitudes suggests insufficient depletion or too much resupply through descent, or, again, too much horizontal mixing. Overall, based on this comparison against OMI, I see no reason to trust the results from the CTM in this case. | Unfortunately, these maps used an unsuccessful interpolation scheme. In the updated version of the article, the maps are corrected and moved to the Supplement. |
| 25. | **Figure 10**. What is shown there? The text talks about "PSC surface area". I take it to mean PSC surface area density, but I'm confused about the units, *mkm2/cm3*. This would be a dimensionless quantity. Maybe I misunderstood something. | To estimate the rates of heterogeneous reactions, the surface area of aerosol particles and particles of polar stratospheric clouds per unit volume is used. The dimension of this parameter is mkm2 / cm3. In the Methods section of the updated version of this article, we have clarified this. |
| 26. | **L425**. What range of isentropic levels or altitudes? | The vertical range is indicated in the updated version of the article. |
| 27. | **L428**. "*nitrogen and hydrogen gases*". I think you mean nitrogen oxides and OH. | This refers to all catalytic cycles of ozone destruction, including nitrogen and hydrogen cycles. |
| 28. | **L429-432**. This seems to suggest that the location of the vortex is an additional constraint on top of chemical depletion. I don't think this is correct. Rapid depletion is tightly confined to the interior of the polar vortex because that's where all the chemically processed air is. In fact, it is essential that the vortex air mass be isolated from the mid latitudes. | These map have been removed from the article. |

Numerous studies demonstrated that chemical composition, particularly the ClOx family in the lower stratosphere exhibits a sharp discontinuity coincident with the vortex edge. Therefore, the position, extent and shape of the polar vortex are already imprinted in the spatial distribution of ozone loss rates. As a side note, that is not to say that dynamical factors don't play a role (resupply through descent, mixing).

This brings me to **Fig. 11**. Since these are snapshots not time-averages, I would expect to see sharp gradients in the loss coefficient maps that would align with the edge of the polar vortex. Instead, the fields vary gradually. Below I plotted 12Z maps of the vortex edge defined as in this paper (black) and via scaled PV contours (see e.g. Manney et al., 2020) at several isentropic levels. I used MERRA-2 so this should be consistent with the CTM. On the right-hand side, I copy/pasted the depletion coefficient fields from Fig. 11. On both days the CTM produces significant depletion outside of the vortex (however defined). For example, on 15 March the CTM shows elevated depletion over Alaska and over Eurasia where it extends almost as far south as Lake Baikal, in both cases far outside of the polar vortex. On 1 April, again, the depletion coefficient map does not bear much resemblance to the vortex. Also, note the similarity between the shape of the vortex on 15 March and the OMI ozone map from the same day.

I see two possibilities: either I grossly misunderstood what is plotted in Fig. 11 or the CTM's chemistry doesn't represent ozone depletion correctly. It may be instructive to look at ClO or ClOx maps in the CTM run. Active chlorine should be confined to the vortex with very limited mixing across the edge. You could compare ClO from the CTM with MLS. Another possible test would be to use the CTM and noCHEMall runs to calculate chemical ozone loss and compare that with the results of section 3.2. In fact, this should be done in order to establish self-consistency of the paper and get a sense of uncertainties.

| 29. | **L432**. I don't understand this sentence. What does "*destruction of the base*" mean? | Corrected. |
|---|---|---|
| 30. | **Figures 12-15** and the accompanying discussion add more doubts about the correctness of the CTM chemistry. Comparing the blue and green lines, it looks like heterogeneous chemistry plays a relatively minor role compared to gas-phase chemistry. At Pechora you estimate it to be responsible for only 25 DU of the total 70 DU of chemical loss! This raises a lot of red flags. According to our well-established understanding of polar ozone, heterogeneous chemistry is responsible for most of ozone destruction, particularly during cold winters with no major SSWs, although NOx can be important during weak-vortex winters (Sagi et al., 2017), which is not the case here. If the results presented here are correct, they require much more rigorous analysis and justification than that presented. | We added vertical analysis into updated version of the manuscript. It can be seen from these figures that at altitudes of 15-25 km, the main role in the destruction of ozone is played by its destruction in halogen cycles, and the role of nitrogen cycles is negligible. However, at altitudes of 25-40 km, significant ozone depletion is noted, in which nitrogen catalytic cycles play the main role. |
| 31. | **LL521-526**. Again, at best this is a very surprising result that needs to be substantiated and supported by additional comparisons with observations. | Additional discussion has been added. |
| 32. | **L536**. *Would have **been** observed* or *would have **occurred***. But is this really demonstrated in the paper? If indeed NOx chemistry played the main role (which I find doubtful) then a disruption of the vortex could cause more depletion by bringing more NOx lower down leading to more loss. I think that this sentence is actually correct (while not substantiated), but it appears at odds with the results of this study (the predominant role of NOx). | We are talking about altitudes 25-40 km related to NOx catalytic cylcles. |
| 33. | **LL548-L553**. This first conclusion repeats one of the results of Lawrence et al., 2020 almost verbatim. There is really nothing new here | Three sentences (first conclusion) were moved upward in the beginning of our short discussion (chapter 5). |

**Technical corrections**

There are a large number of grammatical and style issues. Below I list a few. I feel that it's not necessary at this point to mention all of them as the paper is likely to change very substantially if it gets resubmitted in the future.

**L31** *Sudden stratospheric warming* a *sudden stratospheric warming***s**

Modified

**L33**. Here and in several other places I think *main SSW* is supposed to be *major SSW*

Modified

**L49**. *statically* a *statistically*

Modified

**L66**. I suggest replacing *certain* with *selected*.

Modified

**L75**. *typical to Antarctica* a *typical for Antarctic conditions*

Modified

**L87-89**. Please, revise this sentence. It doesn't read well.

This sentence with comparison of winter 2019-2020 with previous one was removed according to Review 1.

**L99**. *supposed*. I think you mean *suggested*.

Modified

**L118**. *have been formed*. I think *have occurred* would sound better.

We remove several sentences on ozone loss in Antarctica according Review #2

**LL120-124**. This is a very long sentence. Please consider breaking it up into two.

This sentence was removed.

**L129**. *methodology of applied diagnoses* does not read well. How about something like *diagnostic methods*?

Modified

**L136**. This sentence would read better if the word *features* were dropped

Modified

**LL135-139**. Please, expand all the acronyms that were not previously defined (even if they are already expanded in the abstract). Also, please provide references for the models here, where they are first introduced.

Modified

**L153**. *Following to (Runde et al., 2016)* a *Following Runde et al. (2016)*

Modified

**L172**. *radiation transfer model* a **radiative** *transfer model*

Modified

**L203**. *on the board of* a *onboard*.

Modified

**L204**. *at the early March* a *in early March*

Modified

**L315** *by Plumb fluxes* a *using Plumb fluxes*

Modified

**L316**. *dominated* I think you mean *dominant*

According to Review 1 "display dominated" was changed to "show pronounced"

**LL456-458**. I think something like *by the end of March (...) total ozone content at Pechora drops by almost 50 percent* would be more clear and read better.

Modified

**L528**. Please rephrase this sentence. It doesn't read well.

This sentence was removed

**L532**. I suggest changing *values* to *magnitude*.

We remove this sentence.

Thank you again for taking the time to review our manuscript.

With respect,
Sergei P. Smyshlyaev,
Pavel N. Vargin,
Alexander N. Lukyanov,
Natalia D. Tsvetkova,
Maxim A.Motsakov